# UniMedVL: Unifying Medical Multimodal Understanding and Generation through Observation-Knowledge-Analysis

**Junzhi Ning** [* 1]  **Wei Li** [* 1 3]  **Cheng Tang** [* 1 4]  **Jiashi Lin** [1]  **Chenglong Ma** [2 5]  **Chaoyang Zhang** [2]  **Jiyao Liu** [1 5]
**Ying Chen** [1]  **Shujian Gao** [1 5]  **Yuandong Pu** [1 3]  **Huihui Xu** [1 11]  **Chenhui Gou** [7]  **Ziyan Huang** [1]  **Yi Xin** [1 2]  **Qi Qin** [1]
**Diping Song** [1]  **Bin Fu** [1]  **Guang Yang** [9]  **Yuanfeng Ji** [10]  **Tianbin Li** [1]  **Yanzhou Su** [† 8 12]  **Jin Ye** [1 7]  **Shixiang Tang** [13]
**Zhongying Deng** [6]  **Lihao Liu** [1]  **Ming Hu** [1 7]  **Junjun He** [† 1 2]

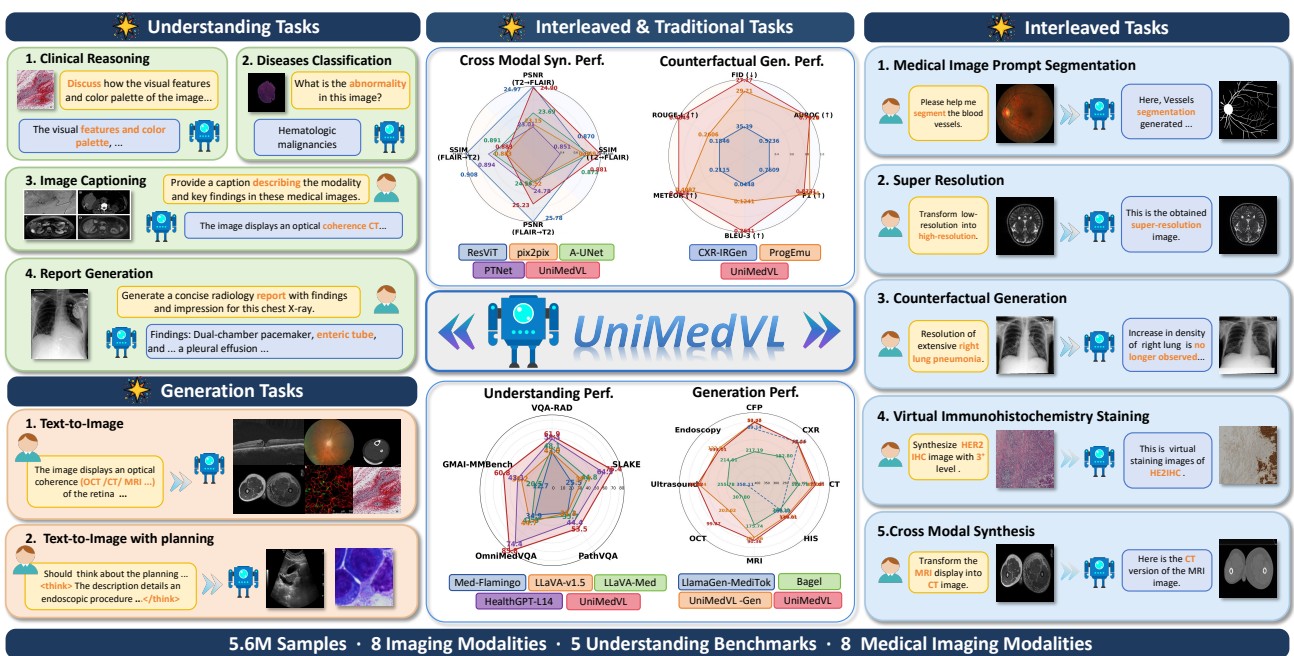

*Figure 1.* **Overview of UniMedVL capabilities and benchmark coverage.**

## Abstract

Medical workflows routinely combine reading images with producing visual and textual outputs, making both image understanding and generation central to medical AI. Most existing systems, however, address these abilities in isolated models, los-

ing the shared knowledge that a unified architecture could exploit. To bridge this gap, we present **UniMedVL**, the first unified medical model that seamlessly integrates multimodal understanding and generation capabilities within a single model without switching weights. We achieve this via a tailored progressive training pipeline where understanding and generation mutually reinforce each other. To effectively train **UniMedVL**, we curate **UniMedVL-5M**, the first large-scale medical dataset comprising over 5.6M instances across 8 medical imaging modalities, tailored for multimodal input-output tasks in unified medical understanding and generation. Experimental results demonstrate that UniMedVL achieves competitive performance on five medical understanding benchmarks. Crucially, UniMedVL natively sup-

---

[*]Equal contribution  [1]Shanghai Artificial Intelligence Laboratory [2]Shanghai Innovation Institute [3]Shanghai Jiao Tong University [4]Shanghai Institute of Optics and Fine Mechanics [5]Fudan University [6]University of Cambridge [7]Monash University [8]DAMO Academy, Alibaba Group [9]Imperial College London [10]The University of Hong Kong [11]The Hong Kong University of Science and Technology [12]Hupan Lab [13]The Chinese University of Hong Kong. Correspondence to: Yanzhou Su <yanzhou.su@outlook.com>, Junjun He <junjun.he@pjlab.org.cn>.

*Proceedings of the 43rd International Conference on Machine Learning*, Seoul, South Korea. PMLR 306, 2026. Copyright 2026 by the author(s).

ports diverse interleaved generation tasks, e.g., virtual staining, super-resolution, cross-modal synthesis, essential for complex medical workflows. Our code and dataset are publicly available.

# 1. Introduction

Medical diagnosis fundamentally follows a structured multilevel reasoning pipeline that is inherently multimodal in both inputs and outputs. Physicians systematically **observe** multimodal raw data, *e.g.*, imaging patterns, patient histories, and symptom descriptions (Huang et al., 2020; Xing et al., 2024), integrate this with medical **knowledge**, *e.g.*, medical literature, domain expertise, and cross-modal associations (Khader et al., 2023; Fang et al., 2025), and **analyse** to produce diverse diagnostic outputs: textual reports explaining findings, visual annotations localizing abnormalities, and comparative imagery for treatment planning (Nguyen et al., 2023; Tanida et al., 2023; Gao et al., 2026; Xu et al., 2025a).

Consider a radiologist examining suspected lung pathology: they process chest X-rays, prior CT scans, and patient history to generate multiple complementary outputs: detailed reports describing findings, visual annotations highlighting specific regions, and comparative visualizations for surgical planning. This exemplifies how a typical medical diagnosis requires unified processing of multimodal inputs to generate diverse multimodal outputs, where neither textual reports alone nor visual annotations alone suffice. However, existing medical AI systems can only understand medical images or produce visual annotations using separate models (Li et al., 2026; Ning et al., 2025b; Lin et al., 2025; Zhang et al., 2025b; Su et al., 2025), rather than unifying them within a single model, as shown in Fig. 2. The limitation primarily originates from three critical levels: (i) **Data**: Medical datasets remain predominantly single-modal despite clear evidence that multimodal integration substantially improves diagnostic accuracy (Warner et al., 2024; Huang et al., 2023; 2024). (ii) **Learning Paradigm**: Although existing medical multimodal models accept multimodal inputs, current approaches essentially apply naive two-stage training strategies, i.e., continual pre-training and instruction tuning, and thus lack a systematic progressive learning paradigm that can effectively capture deep cross-modal relationships between visual and textual information. Recent progress in large language models shows that a well-designed learning paradigm can elicit emergent cross-task generalization at scale, yet such a paradigm remains absent in medical multimodal modeling. (iii) **Capability Convergence**: While general-domain models have made progress toward unified architectures, the medical domain still lacks truly unified models. For instance, although HealthGPT (Lin et al., 2025) demonstrates both understanding and generation capabilities for medical tasks, it requires loading different model

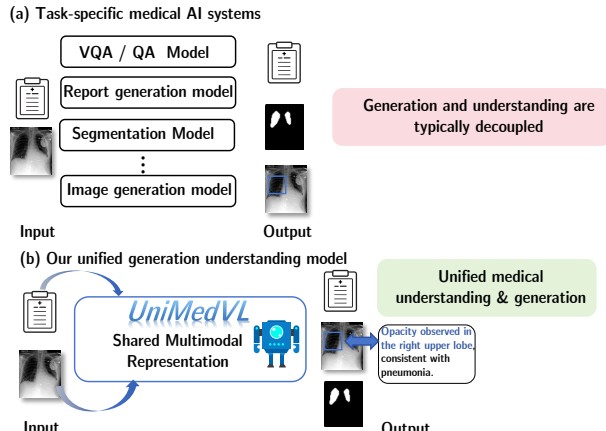

*Figure 2.* **Motivation and Overview of UniMedVL.** (a) Conventional medical AI deploys task-specific models for visual question answering, report generation, segmentation, and image generation, each producing a distinct output from a shared input. (b) UniMedVL replaces the four task-specific models with a single model that shares one multimodal representation across all tasks.

checkpoints depending on the task type. However, the design compels domain experts to manually select different checkpoints for different tasks, hindering seamless multitask operation in medical workflows. Hence, this raises a central question for medical multimodal modeling: can image understanding and image generation share a single medical model and mutually reinforce each other, or does joint training inevitably compromise either capability?

To bridge this gap, we cast unified medical multimodal modeling as a three-level alignment problem through the Observation-Knowledge-Analysis (OKA) paradigm, directly mapping it to the three bottlenecks above: data, learning paradigm, and capability convergence. At the Observation Level, we reformat existing single-modal medical datasets across various tasks into multimodal input-output pairs and construct UniMedVL-5M, comprising over 5.6M samples; this directly exposes the model to large-scale cross-modal pairings instead of disjoint single-modal data and address the data silo that bottlenecks current medical multimodal learning. At the Knowledge Integration Level, we design Progressive Curriculum Learning with three stages: foundation training for basic medical vision–language alignment, instruction tuning for task-following on high-quality data, and a final unified multimodal training stage that jointly optimizes understanding and generation with interleaved inputs and outputs. At the Analysis Level, we introduce **UniMedVL**, a unified medical model that performs both understanding and generation with a single set of parameters trained jointly under the curriculum above.

Experiments provide evidence that, in medical multimodal modeling, aligned multimodal supervision and Progressive Curriculum Learning enable bidirectional transfer between

image understanding and generation within a single model. Our experiments suggest two insights: (1) unified medical training captures cross-modal correlations that transfer across medical tasks and benefit both understanding and generation; and (2) the learned unified representation can be effectively adapted to downstream tasks via task-specific fine-tuning. Our contributions are as follows, as illustrated in Fig. 1:

- **Medical Multimodal Dataset.** We construct and release **UniMedVL-5M**, a 5.6M-sample multimodal medical dataset covering 8 imaging modalities. It provides a standardized training scheme that jointly supports medical image understanding and generation.
- **Progressive Multimodal Learning Paradigm.** We propose a three-stage training pipeline with stage-specific data enhancements and a final unified multimodal training stage, offering a reusable recipe for unified medical model training.
- **Unified Medical Vision-Language Model.** We introduce **UniMedVL**, a medical vision-language model with a single set of parameters for both medical image understanding and generation. UniMedVL unifies inference across task types and demonstrates competitive performance.

## 2. Related Work

### 2.1. Medical Multimodal Large Language Models

Early medical MLLMs largely adopted adapter-style fusion between medical vision encoders and general LLMs (Thawakar et al., 2024; Li et al., 2023), enabling visual question answering and report generation, but often generalizing poorly to real-world medical tasks. A data-centric effort, HuatuoGPT-Vision, converts PubMed papers into large-scale medical image-caption and VQA data (Chen et al., 2024a); however, it remains primarily comprehension-oriented with limited medical reasoning. Reasoning-focused systems including BioMedGPT and Med-PaLM 2 improved biomedical reasoning (?Singhal et al., 2025), and a recent study further systematizes medical output evaluation (Ren et al., 2026); however, these works do not unify image-level generation with textual reasoning. Recently, Lin et al. (2025) proposed HealthGPT as the first medical MLLM for unified multimodal inputs and outputs. It introduces MoE LoRA to mitigate task interference and support diverse tasks. Its unification relies on multiple task-specific models during inference. Thus, these capabilities are not consolidated into a single end-to-end model that performs all tasks in a unified manner.

### 2.2. Unified Image Understanding and Generation

Outside the medical domain, unified multimodal research spans several paradigms. Autoregressive models (Team,

2024a; Wang et al., 2026; Lu et al., 2024) treat images as discrete tokens in decoder-only Transformers, providing architectural unity but constraining high-resolution synthesis due to long sequences and discrete reconstruction. Dual-encoder designs (Wu et al., 2025a; Ma et al., 2025c) mitigate the granularity conflict between semantic understanding and pixel-level generation via separate visual pathways, improving task performance at higher inference cost. Hybrid objectives jointly optimize complementary generative losses, as in Transfusion (Zhou et al., 2025) and Show-O (Xie et al., 2025), while modular approaches (Wu et al., 2025d; Deng et al., 2025; Wu et al., 2024) connect frozen MLLMs to diffusion models via learnable connectors, trading end-to-end differentiability for flexibility. Representation advances narrow the semantics–fidelity gap via multi-codebook quantization (Ma et al., 2026), contrastive-aligned tokenization (Wu et al., 2025e), and unified CLIP semantic spaces (Chen et al., 2025). Recent autoregressive methods (Liao et al., 2025; Zhang et al., 2025a) further improve interleaved generation through deeper fusion. Despite these advances, balancing semantic understanding with pixel-level image generation remains challenging in the medical field, where supervision is fragmented and objectives are heterogeneous.

## 3. Methodology

We present UniMedVL through an observation–knowledge–analysis framework, which first curates large-scale medical multimodal observations, then builds task capability through progressive curriculum learning, and finally unifies medical understanding and generation within a single model.

### 3.1. Observation Level

At the observation level, we address the data bottleneck by reformulating fragmented medical datasets into aligned multimodal input-output pairs.

**Data Source and Modality Coverage.** UniMedVL-5M is built through systematic re-curation and re-creation of heterogeneous medical resources into a unified multimodal learning corpus. It spans 8 primary medical imaging modalities and supports understanding, generation, and interleaved generation tasks through standardized multimodal input-output instances. Detailed dataset statistics are provided in Appendix A.2.

**Quality Control Pipeline.** The data curation process is formalized as sequential quality filters:

$$\mathcal{D}_{\mathrm{qc}} = \mathcal{F}_{\mathrm{align}} \circ \mathcal{F}_{\mathrm{coarse}}(\mathcal{D}_{\mathrm{raw}}) \tag{1}$$

where $\mathcal{F}_{\mathrm{coarse}}$ and $\mathcal{F}_{\mathrm{align}}$ retain image-text pairs $(x, t) \in \mathcal{X} \times \mathcal{T}$ satisfying progressively stricter criteria. Expert validation is performed on a stratified subset of $\mathcal{D}_{\mathrm{qc}}$ as an audit.

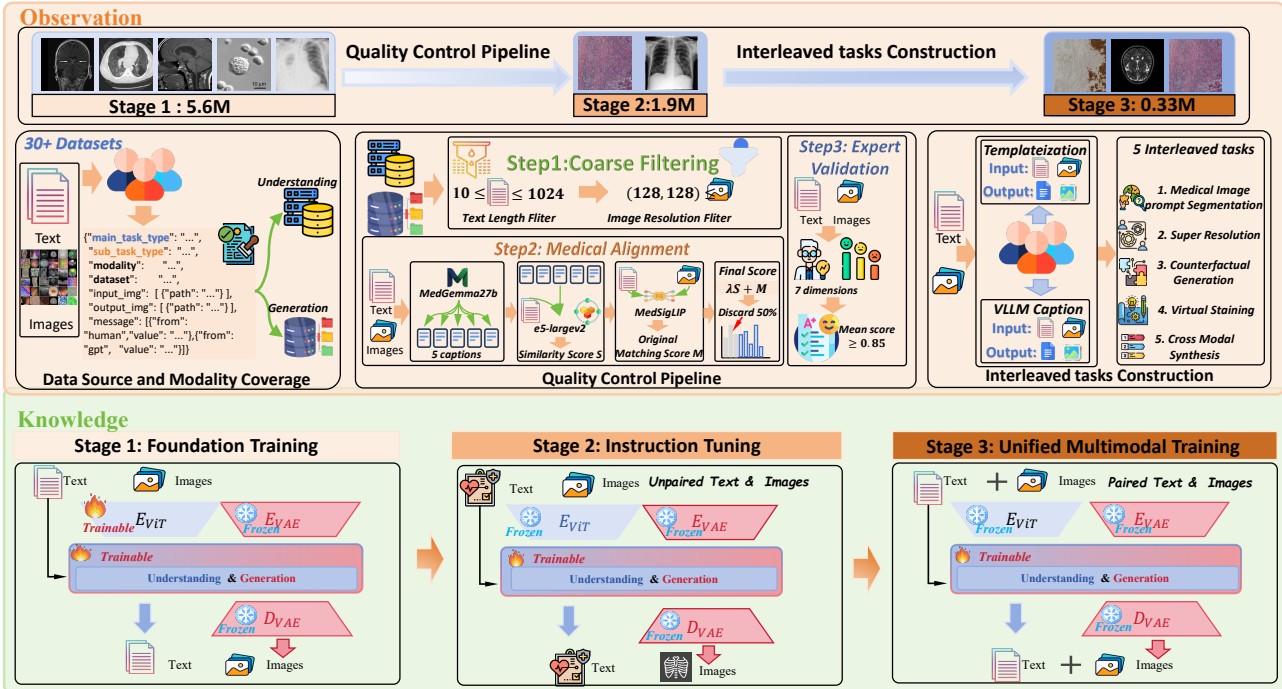

*Figure 3.* **Observation–Knowledge framework. Observation** (top): Data curation pipeline (sources, quality control, interleaved-task construction). **Knowledge** (bottom): Three-stage progressive curriculum (Foundation Training, Instruction Tuning, Unified Multimodal Training).

- **Coarse Filtering** ($\mathcal{F}_{\mathrm{coarse}}$). We apply standard modality-specific preprocessing and light text cleaning/tokenization and retain samples based on the quality indicator: $\phi(x, t) = \mathbf{1}[\min\{H(x), W(x)\} \geq 128] \cdot \mathbf{1}[\tau_{\min} \leq \ell(t) \leq \tau_{\max}]$, where $H(x)$ and $W(x)$ denote image height and width, and $\ell(t)$ is the text length in characters with bounds $\tau_{\min} = 10$, $\tau_{\max} = 1024$.

- **Medical Alignment** ($\mathcal{F}_{\mathrm{align}}$). To preserve pathological findings, we compute a composite alignment score. MedGemma-27b (Sellergren et al., 2025) generates $K = 5$ candidate captions $\mathcal{C}(x) = \{c_k\}_{k=1}^{K}$ per image. The alignment score combines semantic similarity and medical-specific matching:

$$S(x, t) = \max_{c \in \mathcal{C}(x)} \cos(E(t), E(c)),$$
$$M(x, t) = S_{\mathrm{MedSigLIP}}(x, t), \quad (2)$$
$$S_{\mathrm{final}}(x, t) = \lambda \cdot S(x, t) + M(x, t)$$

where $E(\cdot)$ denotes E5-large-v2 embeddings (Wang et al., 2022), $S_{\mathrm{MedSigLIP}}$ is obtained via MedSigLIP (Sellergren et al., 2025), and $\lambda = 0.5$. We normalize both $S(x, t)$ and $M(x, t)$ to $[0, 1]$ before combination. We retain the top 50% of pairs ranked by $S_{\mathrm{final}}$ to obtain a higher-quality subset of training data.

- **Expert Validation.** Five medical experts independently assess a stratified subset spanning medical modalities, with sampling proportional to each modality's frequency.

Each sample is rated along seven medically grounded dimensions. We aggregate the seven scores into a mean quality metric, $\bar{Q} = \frac{1}{7} \sum_{j=1}^{7} Q_j$. We note that MedGemma-27b is used only for alignment scoring during filtering, not to generate the retained training targets; the retained pairs remain original image–text pairs from human-validated open datasets. Expert validation serves as a quality audit on a representative subset, but not a guarantee of zero error across the full dataset.

**Interleaved Tasks Construction.** We curate data from 5 interleaved tasks, including medical image prompt segmentation, super-resolution, counterfactual generation, virtual immunohistochemistry staining, and cross-modal synthesis. Since the original datasets contain only images, we develop a two-stage construction pipeline to formulate them into structured multimodal input-output pairs: Templatization and MLLM Refinement. First, we instantiate structured multimodal input-output pairs by sampling textual prompts and responses from a task-specific template library. Then, we leverage MedGemma-27b to refine the linguistic diversity and medical expressiveness of the input prompts and output explanations. This two-stage design lets us scale interleaved supervision from image-only sources while keeping the textual prompts and explanations clinically faithful. The resulting instances provide aligned multimodal targets that directly support the curriculum described next.

## 3.2. Knowledge Level: Progressive Curriculum Learning

At the knowledge level, we address heterogeneous objectives in medical multimodal modeling with Progressive Curriculum Learning. The curriculum first learns medical vision–language alignment, then strengthens instruction following, and couples understanding and generation through interleaved multimodal tasks.

- **Stage 1: Foundation Training.** We conduct foundation training on the UniMedVL-5M dataset to equip the model with basic medical image understanding and generation capabilities. At this stage, training emphasizes learning general medical vision-language alignments without task-specific constraints.

- **Stage 2: Instruction Tuning.** In this stage, we conduct instruction tuning on our curated high-quality instruction data to improve the model's instruction-following capability for solving complex medical tasks. The instruction-formatted medical tasks follow the format $(q, x_v, k) \rightarrow (a_t, a_v)$ where query $q$, visual input $x_v$, and knowledge context $k$, e.g., patient history, imaging modality metadata, or medical guidelines relevant to the query, generate textual $a_t$ and visual $a_v$ responses. We implement enhancement strategies for distinct task types: For medical understanding tasks such as medical VQAs, we augment standard responses with existing Distilled Chain of Thought (DCOT) data that explicitly articulate the reasoning pathway from visual observation to medical conclusions. For generation tasks, we employ Caption Augmented Generation (CAG) pipeline to enhance caption quality, which incorporates structured planning steps that guide the visual synthesis process. The details are provided in Appendix A.4.

- **Stage 3: Unified Multimodal Training.** We further adapt the stage-2 model using our curated interleaved tasks, where the model jointly processes image inputs and textual prompts to produce corresponding visual content and explanatory text. This stage is designed to establish interleaved multimodal reasoning, which requires our model to integrate understanding and generation within unified sequences. The model consolidates previously learned capabilities while unlocking the ability to generate visual and textual contents simultaneously.

## 3.3. Analysis Level: UniMedVL

At the analysis level, UniMedVL uses a single set of parameters to perform both medical understanding and generation.

**Task Organization.** Model training is systematically organized into three primary task families that reflect the fundamental capabilities required for unified medical multimodal systems: *(i) Understanding tasks*, which emphasize medical image comprehension, visual question answering, diagnostic reasoning, image captioning, and medical report generation; *(ii) Generation tasks*, which focus on conditional medical image synthesis and planning-guided text-to-image generation; and *(iii) Interleaved tasks*, which involve coupled visual-textual inputs and outputs. Unlike single-direction understanding or generation tasks, interleaved tasks require the model to perform bidirectional multimodal grounding, maintain cross-modal semantic consistency and combine both of textual and visual responses within a unified sequence.

**Model Architecture Overview.** Following Deng et al. (2025), we adopt a unified architecture with dual visual encoders and a Transformer backbone. The understanding-oriented encoder $E_{\text{VIT}}$ extracts semantic tokens $z_{\text{VIT}} = E_{\text{VIT}}(x_v)$ for multimodal comprehension tasks, while the generation-oriented encoder $E_{\text{VAE}}$ produces latent representations $z_{\text{VAE}} = E_{\text{VAE}}(x_v)$ for visual synthesis tasks. The transformer backbone consists of specialized FFN layers for the understanding and generation tasks, with shared self-attention layers. The VIT tokens $z_{\text{VIT}}$ and VAE tokens $z_{\text{VAE}}$ are first processed by the specialized FFN layers and then concatenated with text tokens $x_{\text{text}}$ to form a single token sequence. This sequence is fed into shared self-attention layers to predict the next text tokens and the velocity on the VAE latent features. For generation outputs, the decoder $D_{\text{VAE}}$ reconstructs the VAE latent features to images.

**Training Objectives.** We train a single model for multimodal understanding and conditional image generation. Given an image-text pair $(X, T)$ with $T = (T_{\text{in}}, T_{\text{out}})$, we extract two latent features $z_{\text{VIT}} = E_{\text{VIT}}(X)$ and $z_{\text{VAE}} = E_{\text{VAE}}(X)$, and define the unified condition $C = (T_{\text{in}}, z_{\text{VIT}})$. To motivate the unified training objective, we observe a standard information-theoretic identity: jointly modeling the output distribution $(T_{\text{out}}, z_{\text{VAE}})$ can exploit cross-task correlations that conditionally factorized (independent) training cannot. Specifically, the joint conditional entropy is bounded by the sum of individual entropies, with the gap quantifying the redundancy reduction:

$$
\begin{aligned}
&[H(T_{\text{out}} \mid C) + H(z_{\text{VAE}} \mid C)] - H(T_{\text{out}}, z_{\text{VAE}} \mid C) \\
&= I(T_{\text{out}}; z_{\text{VAE}} \mid C) \geq 0.
\end{aligned}
\tag{3}
$$

This inequality implies that a unified model can exploit cross-modal correlations ($I > 0$) to achieve a lower Bayes-optimal uncertainty than separate models. This perspective suggests that joint training can outperform factorized training when the two outputs share non-trivial conditional dependencies. Full derivations are provided in Appendix A.9. Motivated by this theoretical lower bound, we train the model with a combined objective of next-token prediction (NTP) $\mathcal{L}_{\text{NTP}}$ and rectified flow matching $\mathcal{L}_{\text{flow}}$:

$$
\mathcal{L}(\theta) = \mathcal{L}_{\text{NTP}} + \mathcal{L}_{\text{flow}},
\tag{4}
$$

*Table 1.* **Comparison of UniMedVL with Other LVLMs and Unified Multi-Modal Models on Medical Visual Understanding Tasks.** Best in **bold**, second-best underlined (same convention applies to all tables).

| Model | Params | Medical | VQA-RAD | SLAKE | PathVQA | OmniMedVQA | GMAI-MMBench | Avg |
|---|---|---|---|---|---|---|---|---|
| **Understanding Only** | | | | | | | | |
| LLaVA-v1.5 | 7B | ✗ | 42.8 | 37.7 | 31.4 | 44.7 | 38.23 | 38.97 |
| InternVL2 | 8B | ✗ | 49.0 | 50.1 | 31.9 | 54.5 | 43.47 | 45.79 |
| Med-Flamingo | 8.3B | ✓ | 43.0 | 25.5 | 31.3 | 34.9 | 12.74 | 29.49 |
| LLaVA-Med | 7B | ✓ | 48.1 | 44.8 | 35.7 | 41.3 | 20.54 | 38.09 |
| RadFM | 14B | ✓ | 50.6 | 34.6 | 14.33 | 23.5 | 22.34 | 29.07 |
| HuatuoGPT-Vision-7B | 7B | ✓ | 53.0 | 49.1 | 32.0 | 50.0 | 50.22 | 46.86 |
| MedGemma-4B | 4B | ✓ | 67.6 | 71.2 | 33.7 | 68.4 | 44.0 | 56.98 |
| Lingshu-7B | 7B | ✓ | 62.7 | 77.0 | 59.6 | 82.0 | 52.3 | 66.72 |
| Lingshu-32B | 32B | ✓ | **71.4** | **84.7** | **61.3** | 80.4 | 52.7 | **70.10** |
| GMAI-VL | 7B | ✓ | 66.3 | 72.9 | 39.8 | **88.5** | **61.74** | 65.85 |
| GPT-4o | Closed | ✗ | 64.08 | 72.11 | 54.50 | 71.06 | 57.27 | 63.80 |
| Claude Sonnet 4 | Closed | ✗ | 67.60 | 70.60 | 54.20 | 63.22 | 44.26 | 59.98 |
| Gemini-2.5-Flash | Closed | ✗ | 67.41 | 76.03 | 59.70 | 71.33 | 59.05 | 66.70 |
| **Unified Understanding and Generation** | | | | | | | | |
| Janus | 1.3B | ✗ | 52.8 | 26.9 | 27.9 | 45.7 | 39.30 | 38.52 |
| Bagel | 14B | ✗ | 60.09 | 58.91 | 39.05 | 71.13 | 48.11 | 55.46 |
| HealthGPT-M3 | 3.8B | ✓ | 55.9 | 56.4 | 39.7 | 68.5 | 42.08 | 52.52 |
| HealthGPT-L14 | 14B | ✓ | 58.3 | 64.5 | 44.4 | 74.4 | 43.1 | 56.94 |
| **UniMedVL (Ours)** | 14B | ✓ | 61.9 | 75.4 | 53.5 | 85.8 | 60.75 | 67.47 |

where the individual loss terms are

$$\mathcal{L}_{\text{NTP}} = -\sum_{i=1}^{n} \log p_\theta(t_{i+1} \mid t_{\leq i}, z_{\text{VIT}}),$$

$$\mathcal{L}_{\text{flow}} = \mathbb{E}_{t,\epsilon}\left[\|v_\theta(\tilde{z}_{\text{VAE}}, t, C) - v\|_2^2\right], \tag{5}$$

with $\tilde{z}_{\text{VAE}}$ and $v$ following the standard rectified-flow construction.

## 4. Experiments

### 4.1. Benchmarks and Baselines

**Evaluation Benchmarks.** We evaluate UniMedVL across medical visual understanding and generation benchmarks. For **image understanding tasks**, we employ VQA-RAD (Lau et al., 2018), SLAKE (Liu et al., 2021), PathVQA (He et al., 2020), OmniMedVQA (Hu et al., 2024), and GMAI-MMBench (Chen et al., 2024b), which cover diverse medical scenarios. For **image generation tasks**, we split the image–caption pairs in the proposed dataset into 80% for training and 20% for testing. We use the test set to evaluate UniMedVL's text-to-image generation performance. For **interleaved tasks**, we utilize the BCI dataset (Liu et al., 2022b) for the virtual immunohistochemistry staining task. The IXI dataset (Chen et al., 2022) is leveraged to evaluate the super-resolution task, and the BraTS 2023 dataset (Adewole et al., 2023) is used for evaluating the cross-modal synthesis task. We use the ICG-CXR dataset (Ma et al., 2025b) to evaluate the counterfactual generation task.

**Baseline Methods.** We consider two categories: **specialized models** and **unified multimodal models**. For spe-

cialized models, we include understanding-only models such as Med-Flamingo (Moor et al., 2023), LLaVA-Med (Li et al., 2023), HuatuoGPT-Vision (Chen et al., 2024a), RadFM (Wu et al., 2025c), GMAI-VL(Li et al., 2026), Lingshu(Xu et al., 2025b), LLaVA-v1.5 (Liu et al., 2024), and InternVL2 (Team, 2024b). We compare against modality-specialized medical generative models, including RetinaLogos (Ning et al., 2025b) for color fundus photographs, MedSyn (Xu et al., 2024) for CT, CheXGen (Ji et al., 2026) for chest X-ray, and PathLDM (Yellapragada et al., 2024) for histopathology. We compare with image translation models including CycleGAN (Zhu et al., 2017), pix2pix (Isola et al., 2017), pix2pixHD (Wang et al., 2018), pyramid pix2pix (Liu et al., 2022b), SRCNN (Dong et al., 2015), VDSR (Kim et al., 2016), SwinIR (Liang et al., 2021), Restormer (Zamir et al., 2022), AMIR (Yang et al., 2024), ResViT (Dalmaz et al., 2022), and TransUNet (Chen et al., 2021). Additionally, to determine the model performance of medical imaging generation capability, we finetuned LlamaGen-MediTok (Ma et al., 2025a). For unified multimodal models, we include general models like Janus (Wu et al., 2025a) and Bagel (Deng et al., 2025), as well as a medical model HealthGPT (Lin et al., 2025).

**Evaluation Metrics.** We employ task-specific metrics aligned with medical relevance. For **image understanding tasks**, we utilize accuracy as the evaluation metric. For **image generation tasks**, we employ generation FID (gFID) and BioMedCLIP score (CS) to evaluate the quality of synthesized images. For **interleaved tasks**, we leverage PSNR and SSIM as evaluation metrics for virtual immunohistochemistry staining, super-resolution, and cross-modal syn-

*Table 2.* **Multi-modality generation quality: FID scores across eight medical imaging modalities.**

| Method | CFP | CXR | CT | HIS | MRI | OCT | Ultra. | Endo. | Avg |
|---|---|---|---|---|---|---|---|---|---|
| LlamaGen-MediTok | 89.14 | **68.16** | - | 198.63 | - | - | 358.11 | - | 171.85 |
| Bagel | 217.19 | 182.80 | 163.78 | 206.18 | 175.74 | 307.80 | 255.78 | 214.61 | 215.49 |
| UniMedVL-Gen | 77.35 | 190.38 | 79.84 | **107.20** | 82.99 | 107.06 | 100.44 | 121.89 | 108.40 |
| **UniMedVL (Ours)** | **53.20** | 73.04 | **73.04** | 149.01 | **90.36** | **99.27** | **95.38** | 133.11 | **96.29** |

thesis tasks. For counterfactual generation, we follow the experimental setup of ProgEmu (Ma et al., 2025b), using gFID, AUC-ROC, and F1 to evaluate the quality of synthesized images, and BLEU-3, METEOR, and ROUGE-L to assess the quality of the explanatory text.

## 4.2. Performance of UniMedVL

**Medical Visual Understanding Performance.** We evaluate the understanding capabilities of UniMedVL across diverse medical VQA and image comprehension benchmarks. Table 1 compares UniMedVL with existing medical VLLMs and unified multimodal models. Despite supporting both understanding and generation within a single architecture, UniMedVL achieves competitive or superior performance across medical domains. Notably, compared with HealthGPT, which requires separate model checkpoints for different tasks, UniMedVL achieves 85.8% on OmniMed-VQA versus HealthGPT-L14's 74.4% while enabling seamless task switching.

**Medical Image Generation Performance.** Table 2 and Figure 4 evaluate generation quality across eight medical imaging modalities. Comparing UniMedVL-Gen (generation-only training) with UniMedVL reveals that incorporating understanding training substantially improves generation fidelity, reducing the average gFID from 108.40 to 96.29. UniMedVL also achieves BioMedCLIP scores of 0.706 on

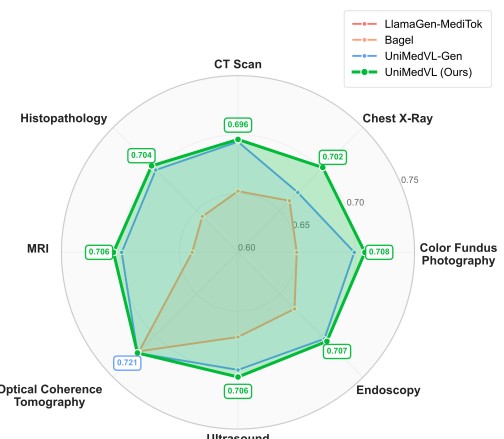

*Figure 4.* **Multi-Modality Performance: BioMedCLIP Score radar chart.** UniMedVL (green region) consistently achieves the highest BioMedCLIP Scores across all 8 medical imaging modalities.

*Table 3.* **External evaluation on held-out image-generation datasets excluded from UniMedVL-5M training**. CS denotes BioMedCLIP Score.

| Model | Metric | CFP | CXR | CT | HIS | MRI | OCT | Ultra. | Endo. |
|---|---|---|---|---|---|---|---|---|---|
| LlamaGen-MediTok | FID ↓ | **57.96** | 140.56 | 104.56 | 211.40 | 113.27 | 114.99 | 65.86 | 188.36 |
| | CS ↑ | 0.71 | 0.62 | 0.63 | 0.61 | 0.65 | 0.71 | 0.69 | 0.65 |
| Bagel | FID ↓ | 231.92 | 126.53 | 167.70 | 185.27 | 134.80 | 245.51 | 280.50 | 158.24 |
| | CS ↑ | 0.64 | 0.65 | 0.61 | 0.62 | 0.63 | 0.63 | 0.65 | 0.68 |
| UniMedVL-Gen | FID ↓ | 134.41 | 130.41 | **74.25** | 107.34 | **61.87** | 69.68 | **56.81** | 126.89 |
| | CS ↑ | 0.71 | 0.67 | 0.68 | 0.67 | 0.71 | **0.72** | **0.71** | 0.70 |
| **UniMedVL (Ours)** | FID ↓ | 60.00 | **63.54** | 100.09 | **102.31** | 87.54 | **54.20** | 86.63 | **94.04** |
| | CS ↑ | **0.72** | **0.71** | **0.70** | **0.72** | **0.73** | 0.70 | 0.70 | **0.70** |

*Table 4.* **Comparison with modality-specialized generators.** Cells report FID ↓ / BioMedCLIP Score ↑.

| Model | CFP | CXR | CT | HIS |
|---|---|---|---|---|
| RetinaLogos | 60.45/**0.711** | - | - | - |
| CheXGen | - | 90.70/0.689 | - | - |
| MedSyn | - | - | 174.42/0.626 | - |
| PathLDM | - | - | - | 241.53/0.617 |
| **UniMedVL (Ours)** | **53.20**/0.708 | **73.04**/0.702 | **73.04**/0.696 | **149.01**/0.704 |

*Table 5.* **Performance on Histological Staining and MRI Super-Resolution.** (Left) H&E to IHC staining transformation; (Right) MRI 4× super-resolution. † indicates fine-tuning variant.

| H&E→IHC Staining | | MRI Super-Resolution | |
|---|---|---|---|
| Method | PSNR/SSIM | Method | PSNR/SSIM |
| CycleGAN | 16.20/0.373 | SRCNN | 28.81/0.892 |
| Pix2Pix | 18.65/0.419 | VDSR | 30.04/0.914 |
| Pix2PixHD | 19.63/0.471 | SwinIR | 31.55/0.933 |
| Pyramid Pix2pix | **21.16/0.477** | Restormer | 31.85/0.938 |
| | | AMIR | **31.99/0.939** |
| HealthGPT-M3 | 15.81/0.242 | HealthGPT-M3 | 18.37/0.580 |
| UniMedVL † | 18.11/0.401 | UniMedVL † | 19.64/0.602 |
| **UniMedVL** | 20.27/0.456 | **UniMedVL** | 27.29/0.890 |

average across modalities in Figure 4, which indicates strong semantic alignment between generated images and medical text descriptions.

This challenges the conventional assumption that joint training compromises individual task performance and shows that medical multimodal learning benefits from task synergy. To verify generalization beyond the training distribution, we evaluate on a held-out external benchmark of text-to-image generation. As Table 3 shows, UniMedVL achieves the lowest average FID and highest average BioMedCLIP Score across eight modalities. This suggests that the generation gains extend beyond the training distribution. We further compare four modality-specialized generation models in Table 4. Despite its unified design, UniMedVL achieves lower FID than all four models, suggesting that unified training can preserve domain-specific generation quality. Across modalities, UniMedVL keeps FID below 100 on six out of eight medical imaging modalities. This suggests that, in our setting, understanding and generation can reinforce rather than compete within a single model.

**Performance on interleaved tasks.** In Table 5, for vir-

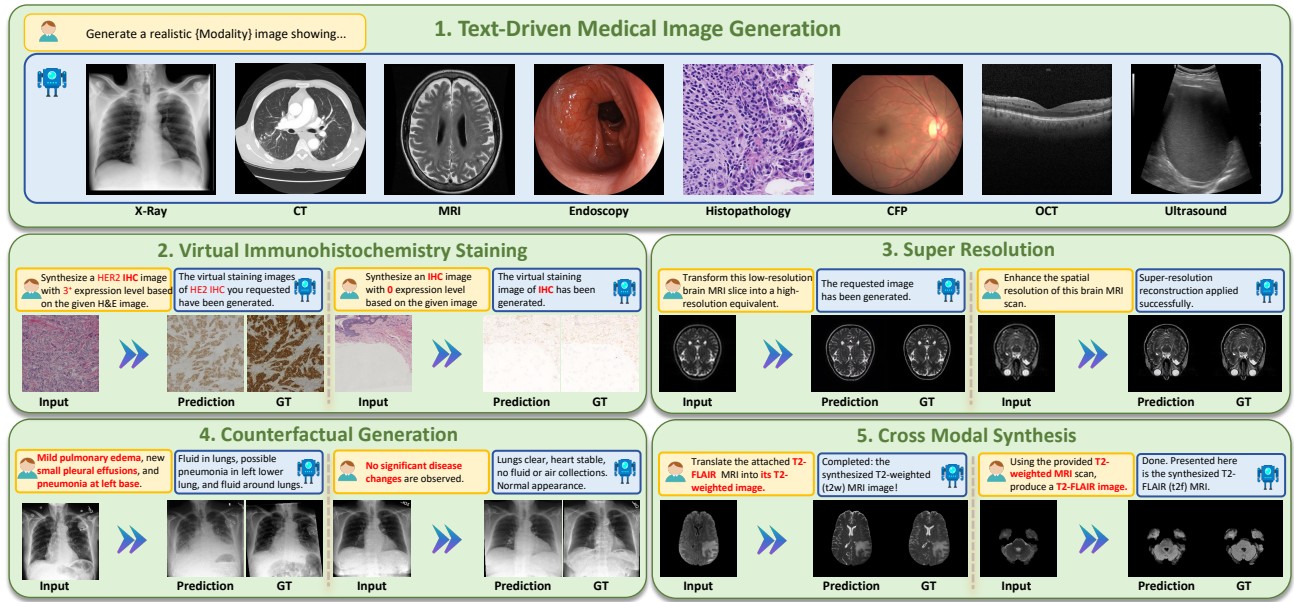

*Figure 5.* **Comprehensive visualization of UniMedVL 's multimodal capabilities.** Demonstration of diverse medical imaging tasks, including text-to-image generation, virtual staining, super resolution, counterfactual generation, and cross-modal synthesis.

*Table 6.* **Performance on Medical Image Translation.** Bidirectional translation between $T_2$ and FLAIR MRI sequences. † indicates fine-tuning variant.

| Method | $T_2\rightarrow$FLAIR | FLAIR$\rightarrow T_2$ | Avg |
|---|---|---|---|
| ResViT | **24.97**/0.870 | **25.78/0.908** | **25.38/0.889** |
| pGAN | 24.01/0.864 | 25.09/0.894 | 24.55/0.879 |
| pix2pix | 23.15/0.869 | 24.52/0.883 | 23.84/0.876 |
| A-UNet | 23.69/0.873 | 24.56/0.891 | 24.13/0.882 |
| SAGAN | 24.02/0.860 | 25.10/0.893 | 24.56/0.877 |
| HealthGPT-M3 | 18.88/0.745 | 19.30/0.750 | 19.09/0.748 |
| UniMedVL † | 23.99/0.711 | 23.49/0.732 | 23.74/0.722 |
| **UniMedVL** | 24.90/**0.881** | 25.23/0.883 | 25.07/0.882 |

*Table 7.* **Comparison of UniMedVL with baseline methods on medical counterfactual generation** using the ICG-CXR dataset. † indicates the fine-tuning variant.

| Method | Counterfactual Image | | | Explanatory Text | | |
|---|---|---|---|---|---|---|
| | gFID↓ | AUROC↑ | F1↑ | BLEU-3↑ | METEOR↑ | ROUGE-L↑ |
| CXR-IRGen | 35.39 | 0.5236 | 0.7609 | 0.0448 | 0.2115 | 0.1846 |
| ProgEmu | 29.21 | 0.7921 | **0.8914** | 0.1241 | 0.4097 | 0.2606 |
| **UniMedVL** † | **27.17** | **0.7970** | 0.8731 | **0.2641** | **0.4486** | **0.4649** |

tual immunohistochemistry staining from H&E to IHC, UniMedVL reports 20.27 PSNR, exceeding HealthGPT-M3 by 28%. On MRI super-resolution, our model obtains 27.29 PSNR and 0.890 SSIM, indicating a clear gain over the unified baseline. For T2-to-FLAIR cross-modal synthesis in Table 6, UniMedVL achieves an average PSNR of 25.07, narrowing the gap to specialized models (Liu et al., 2025) while preserving unified multimodal capabilities. Figure 5 presents qualitative examples of these results. The comparison between UniMedVL† and the full model further

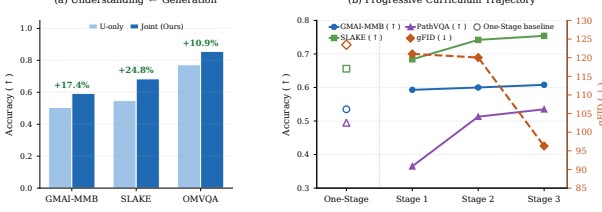

*Figure 6.* **Bidirectional Transfer between understanding and generation in UniMedVL .** (a) Adding generation training to a U-only model lifts accuracy on the reported medical understanding benchmark. (b) Progressive curriculum trajectory: relative to the One-Stage baseline.

shows consistent gains across tasks, indicating that the complete progressive training paradigm captures cross-modal relationships beyond simple fine-tuning.

Table 7 evaluates counterfactual generation with accompanying explanatory text. UniMedVL† obtains 27.17 gFID and higher text-quality metrics than specialized baselines. Its counterfactual check rate of 0.797 AUROC further indicates that unified training can generate medically plausible scenarios while producing coherent textual explanations.

### 4.3. Ablation Study

**Understanding-Generation Synergy.** Table 8 shows that joint training (H-Joint-Base) benefits both directions during the foundation stage, surpassing the single-task variants.

*Table 8.* **Ablation on Understanding-Generation Synergy (Stage 1).** Joint training outperforms single-task variants. CS: BioMedCLIP Score.

| Model | Training | GMAI-MMB↑ | SLAKE↑ | PathVQA↑ | OMVQA↑ | gFID↓ | CS↑ |
|---|---|---|---|---|---|---|---|
| F-Baseline | None | 0.481 | 0.589 | 0.390 | 0.711 | 212.73 | 0.662 |
| C-G-only | Gen | - | - | - | - | 118.60 | 0.699 |
| B-U-only | Und | 0.505 | 0.548 | 0.367 | 0.772 | - | - |
| H-Joint-Base | Joint | **0.593** | **0.684** | 0.365 | **0.856** | 121.02 | 0.683 |

*Table 9.* **Ablation on Progressive Training Stages.** Each stage brings cumulative improvements. CS: BioMedCLIP Score.

| Stage | Data | GMAI-MMB↑ | SLAKE↑ | PathVQA↑ | OMVQA↑ | gFID↓ | CS↑ |
|---|---|---|---|---|---|---|---|
| One-Stage | U+G | 0.535 | 0.656 | 0.495 | 0.778 | 123.48 | 0.695 |
| Stage 1 | U+G | 0.593 | 0.684 | 0.365 | 0.856 | 121.02 | 0.683 |
| Stage 2 | High-quality | 0.600 | 0.742 | 0.513 | **0.863** | 120.04 | 0.699 |
| Stage 3 | +Interleaved | **0.608** | **0.754** | **0.535** | 0.858 | **96.29** | **0.706** |

On understanding, adding generation training lifts GMAI-MMBench accuracy from 0.505 to 0.593, with similar relative gains of 24.8% on SLAKE and 10.9% on OMVQA. On generation, adding understanding training reduces the average gFID by 11.2% over the generation-only variant in Table 2, indicating that semantic supervision benefits generation fidelity. The lone exception is PathVQA, where a 0.5% dip is recovered through instruction tuning in Stage 2.

**Progressive Training Stages.** Table 9 shows that progressive staging further reinforces the synergy between understanding and generation. One-stage joint training obtains 0.535 on GMAI-MMBench and 123.48 gFID, whereas the three-stage progressive approach reaches 0.608 and 96.29, corresponding to a 13.5% gain on GMAI-MMBench and a 22.0% reduction in gFID, respectively. The most pronounced gain in generation appears at Stage 3, where gFID decreases from 120.04 to 96.29 after introducing interleaved tasks that require simultaneous understanding and generation, indicating that these tasks are a key driver for connecting the two capabilities. While joint training already yields synergy in the earliest stage, the largest generation gain arrives only at Stage 3, where interleaved supervision couples the two capabilities most tightly.

*Table 10.* **Ablation on Data Augmentation for Understanding Tasks.** DCOT: Distilled Chain of Thought. Higher values (↑) indicate better performance.

| Data Strategy | GMAI-MMB↑ | SLAKE↑ | PathVQA↑ | OMVQA↑ | Type |
|---|---|---|---|---|---|
| Basic instructions (U) | 0.505 | 0.548 | 0.367 | 0.772 | U-only |
| +DCOT | 0.543 | 0.603 | 0.453 | 0.817 | U-only |
| U+G | 0.593 | 0.684 | 0.365 | 0.856 | Joint |
| High-quality U+G | 0.600 | 0.742 | 0.513 | **0.863** | Joint |
| +Interleaved | **0.608** | **0.754** | **0.535** | 0.858 | Joint |

**Data Augmentation Strategies.** Tables 10 and 11 assess our data enhancement approaches across understanding and generation tasks separately. For understanding tasks, DCOT raises PathVQA from 0.367 to 0.453, with joint training further improving SLAKE to 0.754. For generation tasks, CAG lowers gFID from 118.60 to 108.40, while interleaved joint

*Table 11.* **Ablation on Data Augmentation for Generation Quality.** CAG: Caption Augmented Generation. CS: BioMedCLIP Score.

| Data Strategy | gFID↓ | CS↑ | Δ (gFID/CS) | Type |
|---|---|---|---|---|
| Basic captions (G) | 118.60 | 0.699 | – | G-only |
| +CAG | 108.40 | 0.698 | *-10.20/-0.001* | G-only |
| U+G | 121.02 | 0.683 | – | Joint |
| High-quality U+G | 120.04 | 0.699 | *-0.98/+0.016* | Joint |
| +Interleaved | **96.29** | **0.706** | *-24.73/+0.023* | Joint |

training delivers the strongest generation quality. These results indicate that data-quality and training-strategy improvements are complementary: DCOT and CAG enhance understanding and generation data respectively, while progressive staging with interleaved tasks provides the optimization curriculum.

# 5. Conclusion

We presented UniMedVL, a unified framework for medical image understanding and generation within a single model, built on two medical-specific design choices: the UniMedVL-5M corpus of over 5.6M multimodal medical samples and a three-stage Progressive Curriculum Learning pipeline that progressively activates bidirectional transfer between understanding and generation. Ablation studies further indicate that joint training and progressive staging benefit both understanding and generation. These findings highlight the value of aligning data construction, training curriculum, and model objectives in a unified medical multimodal framework. UniMedVL achieves competitive understanding performance on five medical benchmarks while maintaining competitive generation quality across eight imaging modalities. We position this work as a step toward unified medical multimodal modeling rather than a deployable clinical solution. More broadly, we hope that the released dataset, UniMedVL-5M, will offer a convenient starting point for further study of how understanding and generation can be trained in unified medical settings.

**Limitations.** The current study is restricted to 2D medical imaging and does not yet extend to volumetric modalities such as 3D CT or MRI volumes. Evaluation relies on standard automatic metrics for reproducibility; however, these do not substitute for clinical validation, and deployment in real-world clinical settings would therefore require further clinical investigation. In addition, as a single model trained across many tasks, UniMedVL does not yet match the strongest task-specific systems on every individual benchmark; we regard narrowing this gap while preserving unification as an important direction for future work.

## Acknowledgements

This work was supported by Shanghai Artificial Intelligence Laboratory.

## Impact Statement

This paper presents work whose goal is to advance the field of Machine Learning in AI4Health. The potential societal consequences of our work include both positive impacts (improved diagnostic accuracy, reduced clinical workload) and considerations that warrant attention (model reliability in clinical settings, potential biases in medical data).

This work does not involve experiments on human subjects, patient interventions, or the collection of private medical data. All datasets used in this study are publicly available, including CheXpertPlus, SLAKE, PathVQA, OmniMedVQA, IXI, BraTS 2023, and other open-access medical datasets. Data sources were used in accordance with their respective licenses, and all materials were de-identified prior to use. The purpose of this research is to advance the scientific understanding of unified multimodal modelling for healthcare data rather than to deploy clinical decision-support systems. No patient-level decisions or clinical predictions were made based on model outputs. Expert audits were limited to quality control of publicly available samples and did not involve identifiable patient information.

To ensure reproducibility, all implementation details, model configurations, and training hyperparameters are provided in the Appendix. The full code and configuration files are available at `https://huggingface.co/datasets/General-Medical-AI/UniMedVL-5M`. The UniMedVL-5M dataset construction process, including data sources, quality control criteria, and interleaved task synthesis, is fully documented in the Appendix and illustrated in Figure 2. All benchmarks used for evaluation, VQA-RAD, SLAKE, PathVQA, OmniMedVQA, BraTS 2023, and IXI, are publicly available or derived from publicly available sources.

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

# A. Appendix

**Appendix Contents**

## A.1. Implementation Details

### A.1.1. TRAINING HYPERPARAMETERS

*Table 12.* Training hyperparameters and configurations for the three-stage curriculum learning strategy in UniMedVL.

| | Stage 1 (Foundation) | Stage 2 (Instruction Tuning) | Stage 3 (Unified Multimodal) |
|---|---|---|---|
| **Hyperparameters** | | | |
| Learning rate | $5 \times 10^{-5}$ | $2.5 \times 10^{-5}$ | $1.0 \times 10^{-5}$ |
| Optimizer | | AdamW | |
| Training steps | 85K | 120K | 70K |
| EMA ratio | | 0.995 | |
| Image Resolution (VAE) | 512-1024 | 512-1024 | 32-1024 |
| Image Resolution (ViT) | 378-980 | 224-518 | 378-980 |
| Max tokens per sample | 18.5K | 20K | 27K |
| Dropout | | Text: 0.3, ViT/VAE: 0.05 | |
| ViT training | Trainable | Frozen | Frozen |
| VAE training | | Frozen | |
| Understanding branch | | Trainable | |
| LLM training | | Trainable | |
| **Data Sampling Ratio (%)** | | | |
| Text-Only | 5 | 5 | 3 |
| Text-to-Image (T2I) | 25 | 45 | 35 |
| Image-to-Text (I2T) | 75 | 40 | 37 |
| Interleaved | - | 10 | 25 |

**Detailed Training Strategy Implementation.** Our training employs a three-stage curriculum learning approach that implements the Knowledge component within the OKA framework. We use the AdamW optimizer throughout all stages:

- Stage 1 (Foundation Training) establishes basic medical understanding over 85K steps with a learning rate of $5 \times 10^{-5}$. The data composition prioritizes image-to-text tasks (75%), complemented by text-to-image generation (25%) and pure text data (5%). This stage trains both ViT and LLM components end-to-end while keeping the VAE frozen. The image resolution is restricted with the range from 512-1024 pixels for the generation branch and 378-980 pixels for the understanding branch.

- Stage 2 (Instruction Tuning) extends training to 120K steps with a reduced learning rate of $2.5 \times 10^{-5}$. The data mixture evolves to balance text-to-image (45%) and image-to-text (40%) tasks, while introducing interleaved multimodal datasets (10%). The ViT encoder is frozen at this stage to preserve learned visual features. Token capacity increases to 20K per sample.

- Stage 3, Unified Multimodal Training, focuses on interleaved generation capabilities over 70K steps with a learning rate of $1.0 \times 10^{-5}$. This stage significantly increases interleaved dataset usage to 25% while maintaining balanced generation at 35% and understanding at 37% tasks. The expanded token budget, 27K, and broader image resolution range, 32 to 1024 pixels for generation, support interleaved tasks, including medical image super-resolution, modality translation, and counterfactual generation.

**Hardware Requirements and Training Infrastructure.** Our model training was conducted using $8\times$ A800 GPUs, 80GB memory each, for experimental validation. However, for optimal training efficiency and to fully exploit the model's capacity, we recommend a minimum configuration of $16\times$ A800 GPUs or equivalent hardware.

**Technical Implementation Details.** The training employs a unified loss function that balances understanding and generation objectives with a CE:MSE weight ratio of 0.25:1.0. We apply consistent dropout rates across all stages (Text: 0.3, ViT/VAE: 0.05) to prevent overfitting. The EMA coefficient is set to 0.995 for stable model convergence. Throughout training, the VAE remains frozen to maintain stable latent representations.

A.1.2. PRETRAINED VAE DIAGNOSTIC FOR MEDICAL IMAGING

**Rationale for Using Pretrained VAE without Fine-tuning.** Our approach leverages a general-purpose pretrained VAE model from FLUX (Black Forest Labs, 2024) without medical domain-specific fine-tuning. This design choice addresses two core questions: (1) the reconstruction capability of pretrained VAE on medical imaging modalities, and (2) the cost-benefit trade-off of fine-tuning versus preserving existing capabilities. Regarding the first question, we conducted comprehensive reconstruction experiments across eight medical imaging modalities to evaluate performance. For the second question, considering that our training data is not specifically designed for reconstruction optimization, we did not pursue domain-specific fine-tuning to avoid potential degradation of the model's general-purpose capabilities while maintaining stable latent representations throughout our progressive training stages.

*Table 13.* Reconstruction quality evaluation of pretrained VAE models on medical imaging modalities.

| Metric | Model | $f_d$ | CFP | CT | CXR | Endoscopy | HIS | MRI | OCT | Ultrasound |
|---|---|---|---|---|---|---|---|---|---|---|
| **rFID (Lower is Better)** | | | | | | | | | | |
| | VAE (FLUX) | 8 | 13.22 | 5.81 | 5.42 | 11.77 | 10.00 | 10.58 | 13.23 | 9.64 |
| | Direct End-to-end VAE (FLUX) | 8 | 14.05 | 30.59 | 23.28 | 39.56 | 44.64 | 37.95 | 17.33 | 31.58 |
| | VQGAN | 8 | 27.22 | 15.97 | 18.66 | 27.33 | 67.68 | 21.33 | - | 29.48 |
| | Emu3-VQ | 8 | 16.27 | 11.83 | 11.99 | 20.83 | 69.89 | 13.52 | - | 25.43 |
| | MedITok | 16 | 14.39 | 7.88 | 6.55 | 10.66 | 46.54 | 6.32 | - | 17.64 |
| **PSNR (Higher is Better)** | | | | | | | | | | |
| | VAE (FLUX) | 8 | 34.58 | 37.34 | 37.09 | 35.33 | 34.50 | 34.30 | 34.58 | 33.59 |
| | Direct End-to-end VAE (FLUX) | 8 | 35.11 | 34.43 | 31.28 | 31.98 | 29.69 | 34.82 | 30.83 | 35.17 |
| | VQGAN | 8 | 35.40 | 31.13 | 31.68 | 25.60 | 20.42 | 29.54 | - | 24.79 |
| | Emu3-VQ | 8 | 39.64 | 36.11 | 35.81 | 28.96 | 22.08 | 34.32 | - | 27.57 |
| | MedITok | 16 | 37.72 | 36.32 | 34.42 | 29.19 | 23.54 | 33.55 | - | 28.49 |
| **SSIM (Higher is Better)** | | | | | | | | | | |
| | VAE (FLUX) | 8 | 0.892 | 0.951 | 0.973 | 0.934 | 0.922 | 0.921 | 0.892 | 0.938 |
| | Direct End-to-end VAE (FLUX) | 8 | 0.842 | 0.848 | 0.904 | 0.900 | 0.938 | 0.934 | 0.867 | 0.816 |
| | VQGAN | 8 | 0.923 | 0.885 | 0.911 | 0.768 | 0.484 | 0.844 | - | 0.682 |
| | Emu3-VQ | 8 | 0.943 | 0.928 | 0.955 | 0.847 | 0.547 | 0.957 | - | 0.751 |
| | MedITok | 16 | 0.953 | 0.937 | 0.954 | 0.890 | 0.660 | 0.972 | - | 0.839 |

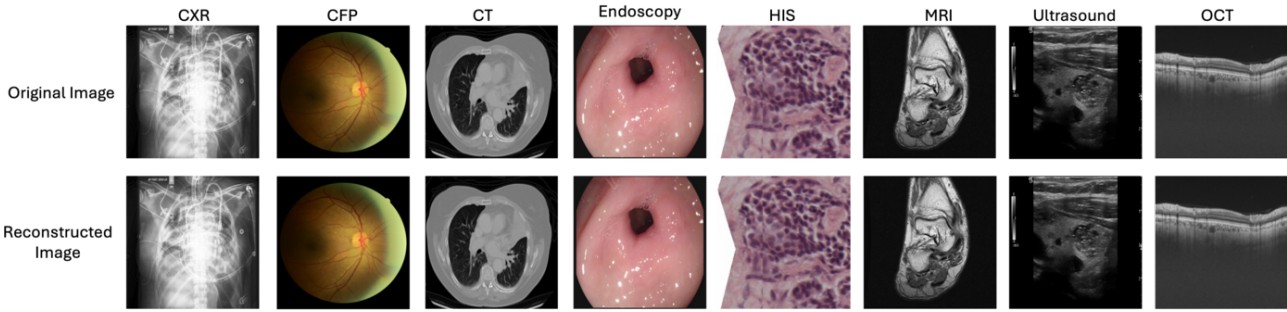

*Figure 7.* **Qualitative comparison of VAE reconstruction quality across diverse medical imaging modalities.** Visual examples demonstrating reconstruction fidelity across eight medical imaging modalities (CFP, CT, CXR, Endoscopy, HIS, MRI, OCT, Ultrasound) using the pretrained FLUX VAE without domain-specific fine-tuning.

The empirical evaluation demonstrates that the VAE (FLUX) achieves competitive reconstruction performance across eight distinct medical imaging modalities without requiring domain-specific fine-tuning. With a compression factor of $f_d = 8$, the model consistently delivers low rFID scores, competitive PSNR values, and robust SSIM scores. To evaluate the necessity of domain-specific adaptation, we performed direct end-to-end fine-tuning of the FLUX VAE on medical imaging data (highlighted in red in Table 13). The empirical results demonstrate that domain-specific fine-tuning yields negligible performance improvements for generation tasks across medical modalities, exhibiting inconsistent variations in rFID scores with marginal changes in corresponding metrics. Consequently, these observations validate the deployment of pretrained

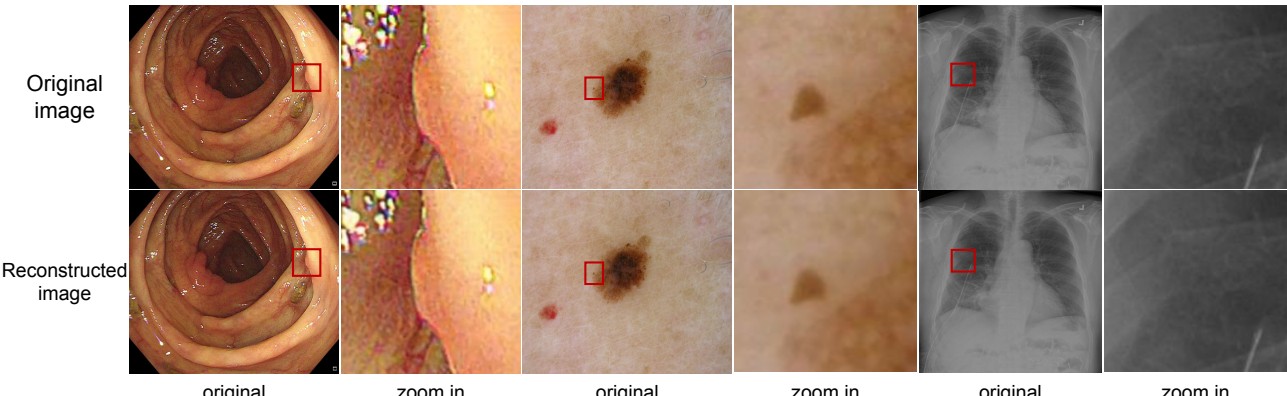

*Figure 8.* **Preservation of medically important small lesions in VAE reconstruction.** Comparison of original images (top row) and VAE-reconstructed images (bottom row) across three medical imaging scenarios. **Left:** Endoscopy image showing a polyp (highlighted in red box). **Middle:** Dermoscopy image displaying skin lesion with globules (highlighted in red box). **Right:** X-ray image with fractures (highlighted in red box). **Original** denotes the original image, and **Zoom-in** denotes the zoomed-in view of the lesion within the red boxes. The magnified views demonstrate that the FLUX VAE preserves fine structural details essential for medical interpretation, despite being a general-purpose encoder not specifically fine-tuned for medical imaging.

VAE models without domain-specific fine-tuning for 2D medical imaging generation applications within our experimental framework.

### A.1.3. RECONSTRUCTION FIDELITY FOR MEDICALLY IMPORTANT SMALL LESIONS

While Table 13 demonstrates competitive aggregate reconstruction metrics (rFID, PSNR, SSIM) across diverse medical modalities, these metrics may not fully capture the preservation of small but medically critical structures such as polyps in endoscopy, dermatoscopic features in skin lesions, or fractures in radiographs. To address this concern, we conducted targeted qualitative analysis focusing on the reconstruction fidelity of fine anatomical details and pathological findings that are essential for medical interpretation.

We selected representative cases from three imaging modalities where small lesion detection is medically critical: (1) an endoscopy image containing a polyp, (2) a dermoscopy image with skin lesion with globules, and (3) an X-ray image. For each case, we compared the original image with its VAE-reconstructed counterpart, examining both full-field views and magnified regions of interest (ROIs) centered on the lesions.

Figure 8 illustrates that the pretrained FLUX VAE maintains visually discernible fidelity for small pathological features. In the endoscopy image, the polyp's morphology and surface texture remain well preserved in the reconstruction, with boundary definition comparable to the original. For the dermoscopy image, the characteristic globular patterns, key diagnostic features for distinguishing benign nevi from melanoma—are clearly visible in both the original and reconstructed versions. In the X-ray image, the highlighted region and surrounding anatomical structures maintain structural coherence post-reconstruction. These qualitative observations suggest that the general-purpose VAE also preserves medically relevant fine-grained details that are useful for downstream tasks within our unified framework.

## A.2. Dataset Statistics

### A.2.1. DATASET COMPOSITION DETAILS

*Table 14.* Overview of training stage data distribution, showing data composition, task types, and scale statistics across different stages. Stage 2 utilized the high-quality subset of stage 1 datasets.

| Training Stage | Total Entries | Task Categories |
|---|---|---|
| **Stage 1: Foundation Training** | | |
| Understanding Tasks | 4.0M | Image comprehension, VQA |
| Generation Tasks | 1.6M | Text-to-image, controllable generation |
| *Stage 1 Subtotal* | *5.6M* | *Foundation capabilities* |
| **Stage 2: Instruction Tuning** | | |
| Understanding Tasks | 698K | Image CoT, clinical reasoning |
| Generation Tasks | 668K | Enhanced T2I, medical translation |
| CoT Understanding | 317K | Chain-of-thought reasoning |
| Text-only Tasks | 230K | Medical QA, clinical dialogue |
| *Stage 2 Subtotal* | *1.9M* | *Knowledge integration* |
| **Stage 3: Unified Multimodal Training.** | | |
| Interleaved Tasks | 330K | 5 interleaved tasks |
| *Stage 3 Subtotal* | *0.33M* | *Unified capabilities* |
| **Total Dataset** | **5.6M** | **All medical tasks** |

### A.2.2. MEDICAL DOMAIN AND MODALITY DISTRIBUTION

*Table 15.* Major datasets detailed information, showing key dataset contributions sorted by data volume. For open-source datasets, the reported numbers indicate the actual subset sizes used in our training pipeline after filtering. Part of this curation effort contributes to the (Deng et al., 2026).

| Dataset Name | Total Entries | Primary Tasks |
|---|---|---|
| PMC-OA (Lin et al., 2023) | 1.0M | Image Captioning |
| Quilt-1m (Ikezogwo et al., 2023) | 644K | Histopathology Understanding |
| Healthgpt (Lin et al., 2025) | 638K | Clinical Reasoning, Image Caption |
| PubMedVision (Chen et al., 2024a) | 385K | Controllable T2I Generation |
| Gmai-vl (Li et al., 2026) | 288K | Enhanced T2I Generation |
| Bigbio (Fries et al., 2022) | 262K | Clinical Reasoning with CoT |
| CheXpertPlus (Chambon et al., 2024) | 223K | Medical Report Understanding |
| PMC VQA (Zhang et al., 2024) | 204K | Image Caption |
| Internvl (Chen et al., 2024c) | 188K | Disease Classification, Clinical Reasoning |
| Medicat (Subramanian et al., 2020) | 132K | Controllable T2I Generation |
| Medical-diff-vqa (Hu et al., 2023) | 129K | Image Caption, Entity Recognition |
| PMC-Inline (Wu et al., 2025b) | 121K | Multi-image Understanding |
| IXI T2/T1 SR 4x (Chen et al., 2022) | 161K | Super resolution |
| BraTS23 Modality Tran (Baltruschat et al., 2023) | 52K | Cross modal synthesis |
| SynthRAD Brain (MR to CT/CT to MR) (Thummerer et al., 2025) | 66K | Cross modal synthesis |
| SynthRAD Pelvis (MR to CT/CT to MR) (Thummerer et al., 2025) | 42K | Cross modal synthesis |
| ICG-CXR dataset (Ma et al., 2025b) | 10K | Counterfactual generation |
| BCI dataset (Liu et al., 2022a) | 5K | Virtual immunohistochemistry staining |
| **Total (Selected Datasets)** | **4.55M** | – |
| **Other Datasets** | **1.05M** | – |
| **Grand Total** | **5.6M** | **All Tasks** |

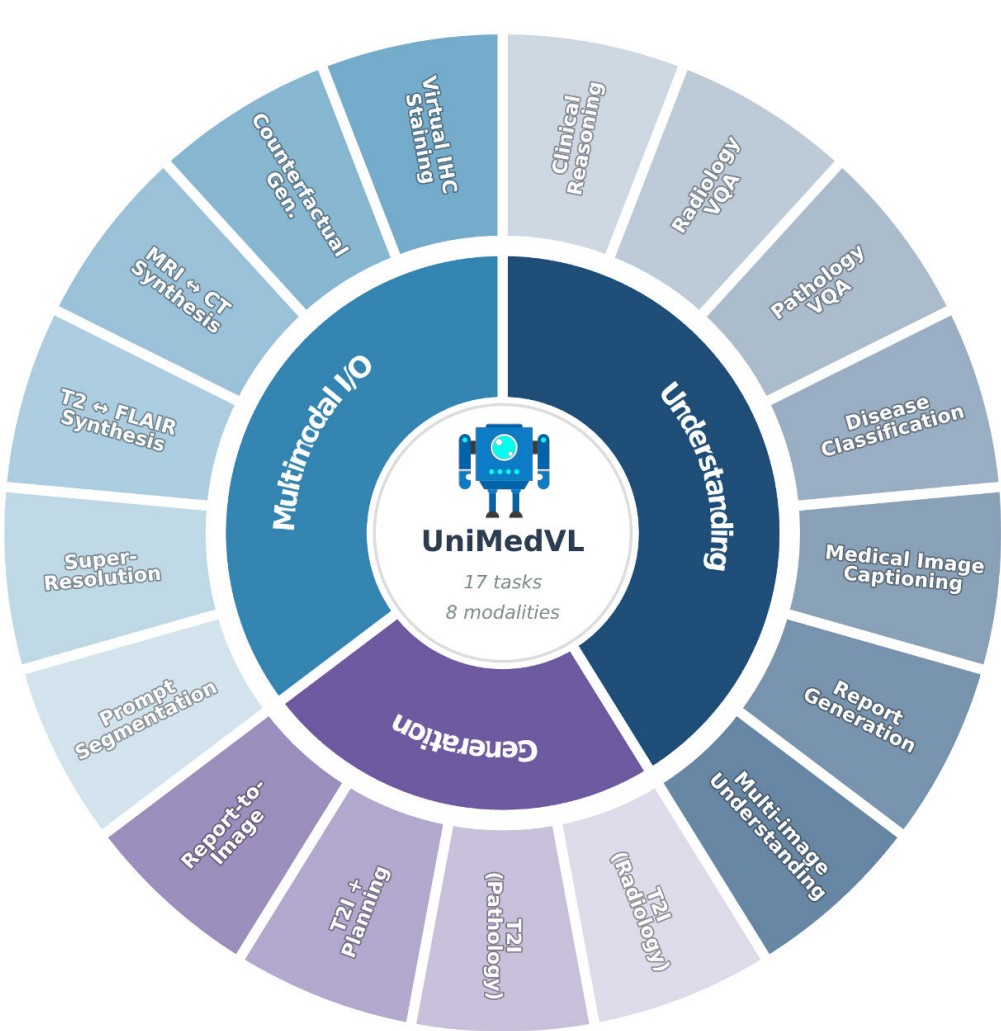

*Figure 9.* **Task taxonomy of UniMedVL .** Diverse medical tasks unified under one model, grouped into Understanding, Generation, and Multimodal input and output.

### A.3. Data Enhancement Pipeline: CAG Implementation

This section presents the complete prompt templates used in our Caption Augmented Generation (CAG) pipeline for image generation tasks, as described in Section 3. The CAG pipeline consists of two main stages: (1) structured medical description generation for quality control, and (2) caption fusion that combines original captions with generated descriptions.

#### A.3.1. STAGE 1: STRUCTURED DESCRIPTION GENERATION

---

**Stage 1: Structured Description Generation Prompt**

**Purpose:** Generate four-level structured medical image descriptions for quality control and similarity computation

```
You are a universally expert medical image analyst, proficient in all
imaging modalities and anatomical systems.
Your input is a single medical image, with no supplementary information.
Your only task is to provide a comprehensive, objective, and structured
description at four distinct levels, from the highest overview down to
the most specific and exceptional findings.
You must not offer any diagnostic, interpretive, or clinical advice.

---

Output Structure (Four-Level, Top-to-Bottom -- definitions for your
internal guidance; do NOT reproduce these headings in your answer)

LEVEL 1: IMAGE TYPE & GLOBAL CONTEXT
• In one sentence, state the presumed imaging modality (if visually
  clear), main body region(s), and overall image category (e.g.,
  cross-sectional, projectional, histological).
• Example: "This is an axial CT image of the abdomen and pelvis,
  showing cross-sectional anatomy at the level of the lower kidneys."

LEVEL 2: MACRO-ANATOMICAL OVERVIEW
• In 2-4 concise lines, summarize the global distribution and layout
  of major anatomical regions, dominant structures, and any clearly
  visible large-scale abnormalities, masses, or disease patterns.
• Describe anatomical orientation, symmetry, major organ relationships,
  and other visually prominent features.

LEVEL 3: ORGAN / SUBREGION DETAILS -- must be the most detailed section
• In 6-12 lines (use complete sentences), describe the visual
  appearance of individual organs, vessels, bones, or other relevant
  subregions.
• Provide precise, granular, reproducible details so that all main
  features can be reconstructed.
• Maintain strict objectivity; do not include diagnostic language.

LEVEL 4: SPECIAL OR INCIDENTAL FINDINGS
• List any unusual devices, postsurgical changes, image artifacts,
  rare morphologic features, or observations not already mentioned above.
• If none are visible, explicitly state: "No distinct pathological
  or incidental findings are visible."

Writing Instructions
1. Write the entire description as one continuous paragraph that
   implicitly follows the LEVEL 1 → LEVEL 4 order--do not include
   level headings, bullet points, or numbered lists in the paragraph.
2. Do not use bullet points elsewhere (except within the examples).
3. For more complex images, the portion corresponding to LEVEL 3 should
   naturally be longer; for simpler cases, keep it proportionally concise.
4. Avoid any clinical judgement or speculation--describe only what is
   directly visible.
```

---

A.3.2. STAGE 2: CAPTION FUSION ENHANCEMENT

This stage fuses original captions with Stage 1 generated structured descriptions to create enhanced descriptions for image generation tasks.

---

**Stage 2: Caption Fusion Enhancement Prompt**

**Purpose:** Fuse original captions with structured descriptions for enhanced image generation prompts

```
You are a universally expert medical image analyst, proficient in all
imaging modalities and anatomical systems.

CRITICAL CONSTRAINT: You must maintain absolute anatomical consistency.
NEVER change, assume, or modify the anatomical location described in the
original caption. Do not make assumptions about different anatomical locations or
transfer descriptions between different body parts.

Your input consists of:
1. A structured, objective, four-level description derived from a locally
   deployed AI model (following a strict hierarchy from global overview
   to specific findings).
2. An original, data-derived textual description containing high-density,
   potentially diagnostic or interpretative information, which may lack
   structured clarity.

Your task is to:
• First, critically review and confirm the completeness of the structured
  description generated by the local model.
• Then, systematically extract and objectively incorporate relevant,
  visually verifiable details from the original data-derived description,
  enhancing information density without including diagnostic, interpretive,
  or clinical judgement.
• Clearly indicate and explicitly include visually evident anatomical
  abnormalities, structural deviations, or incidental observations present
  in the original data but omitted in the structured description.

Output Structure (Four-Level, Top-to-Bottom)
LEVEL 1: IMAGE TYPE & GLOBAL CONTEXT
• In one sentence, state the presumed imaging modality, main body
  region(s), and overall image category.

LEVEL 2: MACRO-ANATOMICAL OVERVIEW
• In 2--4 concise lines, summarize global anatomical distribution,
  dominant structures, anatomical symmetry or deviations, and clearly
  visible large-scale abnormalities.

LEVEL 3: ORGAN / SUBREGION DETAILS -- must be the most detailed section
• In 6--12 complete sentences, describe individual organs, bones,
  vessels, and other relevant anatomical subregions in precise,
  reproducible detail.
• Objectively highlight visually confirmed abnormalities or structural
  deviations derived from the original data description.

LEVEL 4: SPECIAL OR INCIDENTAL FINDINGS
• Explicitly mention unusual devices, postsurgical changes, rare
  morphological features, or visually detectable anomalies present in
  the original description yet absent in the structured description.
• Clearly state the absence of commonly expected baseline anatomical
  or pathological features if definitively not observed in the image.

Writing Instructions
1. Write the final enhanced description as a single, continuous paragraph
   implicitly following LEVEL 1 → LEVEL 4 order--do not include explicit
   level headings, bullet points, or numbered lists.
2. Avoid any clinical judgement, diagnostic language, or speculative
```

---

```
    interpretation--include only details directly verifiable from visual
    inspection.
3. Start your output with "Please generate a realistic [modality] image
    showing" to make it a proper generation instruction.
```

### A.3.3. STAGE 3: THINKING-ENHANCED RESPONSE GENERATION

This stage aims to elicit the reasoning process from the medical foundation model (MedGemma-27B-IT) by prompting it to explicitly generate its internal thinking steps. We leverage this specialized medical model to simulate detailed reasoning processes through the structured prompt format. The resulting data, which includes both the explicit thinking traces and the final responses, is then used to train our model.

---

**Stage 3: Thinking-Enhanced Response Generation Prompt (Revised v2)**

**Purpose:** Generate medical image responses with thinking tags for enhanced reasoning and quality control

```
System: You are a medical image generator. You create [modality] images based
on clinical descriptions. Your responses should describe what features you
have generated in the image from the creator's perspective. Use bullet points
to organize the anatomical structures and clinical features you have included
in your generated image.

User: Based on this clinical description: "[clinical_description]"

You have been given the corresponding medical image. Please provide a response
following this format:

Required format:
<think>Analyzing the clinical description, I need to generate an image that
captures: 1) The key pathological process described, 2) The anatomical
structures involved, 3) The specific imaging characteristics for [modality].
Based on the clinical presentation, I should include [key features reasoning].
[structured_caption if available]</think>

Here/This is the generated [modality] image that displays:
• [anatomical structure or clinical finding 1]
• [anatomical structure or clinical finding 2]
• [anatomical structure or clinical finding 3]

IMPORTANT:
1. In the <think> tag, reason through WHAT you need to generate and WHY based
    on medical knowledge
2. Respond from the GENERATOR perspective - describe what features you have
    CREATED/GENERATED in the image
3. Use the exact format above with bullet points (•) to list features
4. Start with 'Here is the generated [modality] image that displays:'
5. Each bullet point should describe a specific anatomical structure,
    clinical finding, or visual feature that you have included
6. Do NOT use observational language like 'shows', 'visible', 'can be seen'
    - instead use generative language like 'displays', 'includes',
    'features', 'contains'

Note: The thinking tag should reflect your decision-making process: "I need
to generate X because Y", "The clinical description indicates I should
include Z", etc.
```

---

The enhanced captions from Stage 2 (if the process "generating" is not generated successfully) and Stage 3 (if the process "thinking" is generated successfully) are sampled and then submitted to the Expert Review system (Section A.4) for final validation.

## A.4. Expert Review Validation System

This section presents an expert review validation system that evaluates the quality of our UniMedVL-5M dataset construction and two caption generation approaches described in the Data Enhancement Pipeline (Section A.3):

**Simple approach:** Caption fusion that combines structured descriptions from Stage 1 with original captions (Stage 2 of CAG pipeline).

**Thinking-enhanced approach:** Incorporates an additional planning process with <think> tags that integrates reasoning steps before medical image generation (Stage 3 of CAG pipeline). The validation system evaluates both data quality and methodological effectiveness.

### A.4.1. EXPERT REVIEW FRAMEWORK OVERVIEW

Our expert review validation system is designed around a seven-dimensional medical evaluation framework that assesses medical AI performance.

Our evaluation framework encompasses seven dimensions that assess the synthetic quality of medical image captions. All dimensions are scored on a 0 to 5 scale, except Modality Match which uses a binary 0 to 1 scale. The framework begins with **Modality Match**, which measures consistency between images and declared medical imaging modalities, followed by **Factual Accuracy** that evaluates the precision of anatomical structure and pathological finding descriptions. **Information Completeness** assesses coverage of diagnostically relevant key information, while **Position/Quantity Accuracy** measures precision in anatomical localization and quantitative assessments. The framework also incorporates **Professionalism** to evaluate adherence to medical reporting standards, **Planning Coherence** to assess systematic thinking and logical organization quality, and finally **Clinical Reasoning**, a Turing-test style judgment, to measure approximation to human expert-level performance.

**Expert Validation Protocol:** Experts conducted audits of 200 samples across all seven dimensions, with mean quality score $\bar{Q} \geq 0.85$ consistently achieved across all dimensions.

### A.4.2. EVALUATION DIMENSION ANALYSIS

Figure 10 presents the correlation analysis and comparative results. Figure 10a shows inter-dimensional correlations, while Figure 10b compares the two generation approaches.

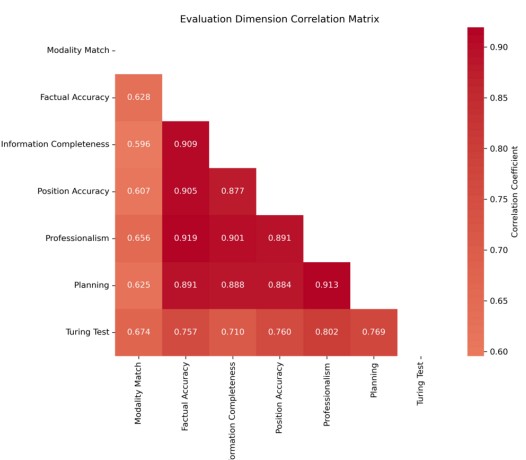

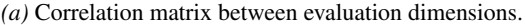

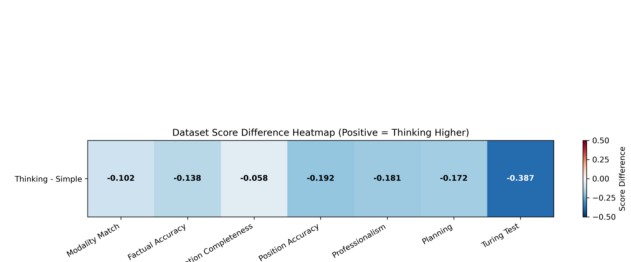

*(a)* Correlation matrix between evaluation dimensions.

*(b)* Score difference heatmap comparing thinking and simple approaches.

*Figure 10.* **Expert evaluation analysis.** (a) Correlation matrix revealing inter-dimensional relationships, with Pearson correlation coefficients ranging from 0.60 to 0.92. (b) Score difference heatmap comparing thinking and simple approaches, where negative values indicate the simple approach scores higher; all dimensions are scored on a 0 to 5 scale except Modality Match, which uses a 0 to 1 scale.

### A.4.3. DATASET QUALITY COMPARISON ANALYSIS

Figure 11 compares the two generation approaches across all evaluation dimensions. The radar chart (Figure 11a) shows closely aligned performance profiles.

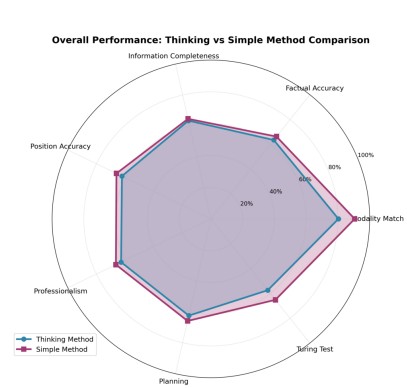

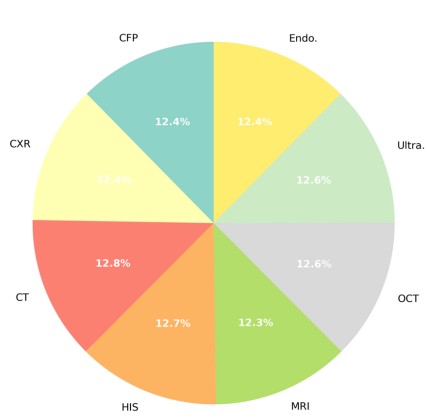

*(a)* Performance comparison: Thinking vs Simple approaches across evaluation dimensions.

*(b)* Medical imaging modalities distribution

*Figure 11.* **Expert validation overview.** (a) Radar chart comparing performance profiles of thinking and simple approaches across all seven evaluation dimensions. (b) Pie chart showing balanced representation across medical imaging modalities, ensuring comprehensive coverage.

### A.4.4. MEDICAL MODALITY-SPECIFIC ANALYSIS

Figure 12 presents modality-specific performance across eight medical imaging modalities. Figure 12a shows statistical comparisons, and Figure 12b displays detailed performance metrics.

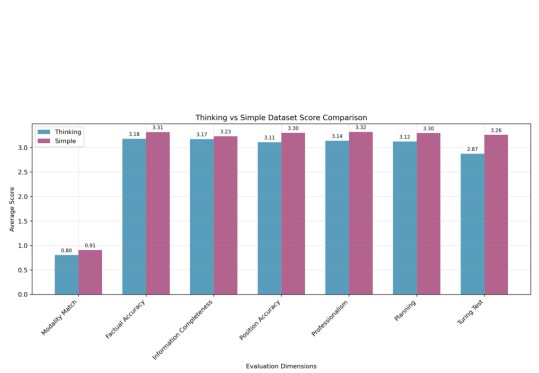

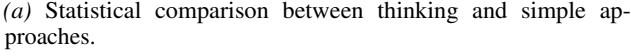

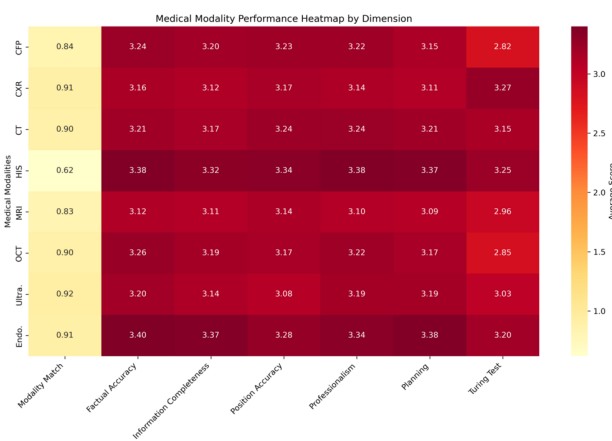

*(a)* Statistical comparison between thinking and simple approaches.

*(b)* Modality-specific performance analysis.

*Figure 12.* **Comprehensive performance analysis.** (a) Bar chart showing mean scores with confidence intervals. (b) Heatmap displaying modality-specific performance scores.

Across the evaluation figures above, the simple approach yields marginally higher scores on caption-fidelity dimensions. The simple approach stays close to the original human-validated captions with minimal extrapolation, whereas the thinking-enhanced approach introduces explicit reasoning traces that target downstream planning and counterfactual generation rather than caption fidelity.

## A.5. Downstream Task Results

### A.5.1. MEDICAL REPORT GENERATION

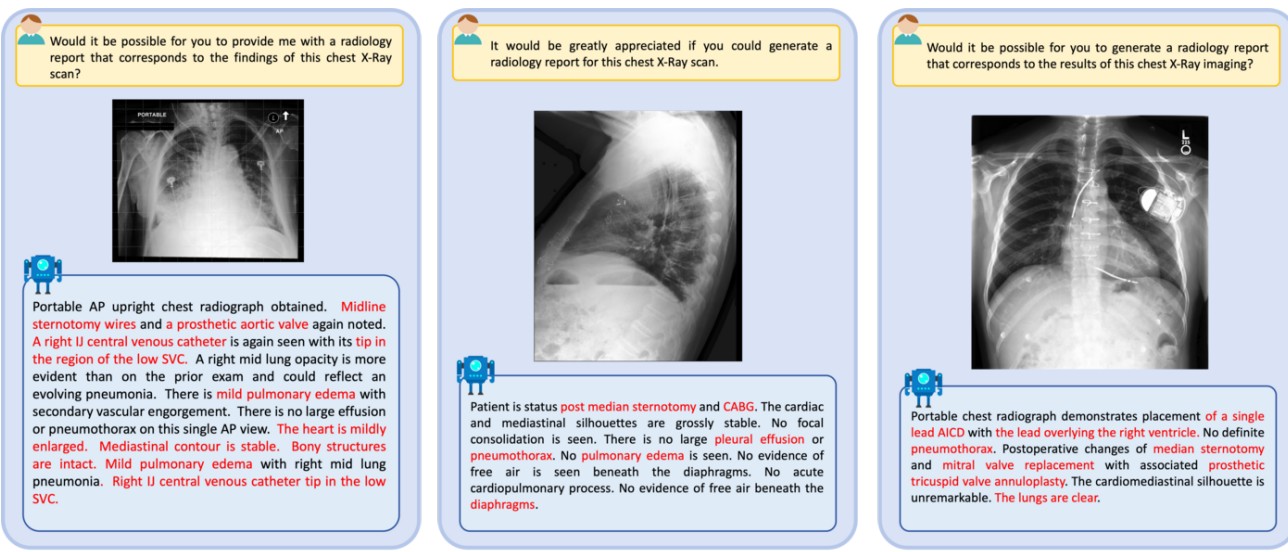

*Figure 13.* Medical report generation examples showing input images and generated reports.

### A.5.2. MULTI-ROUND MEDICAL VQAS

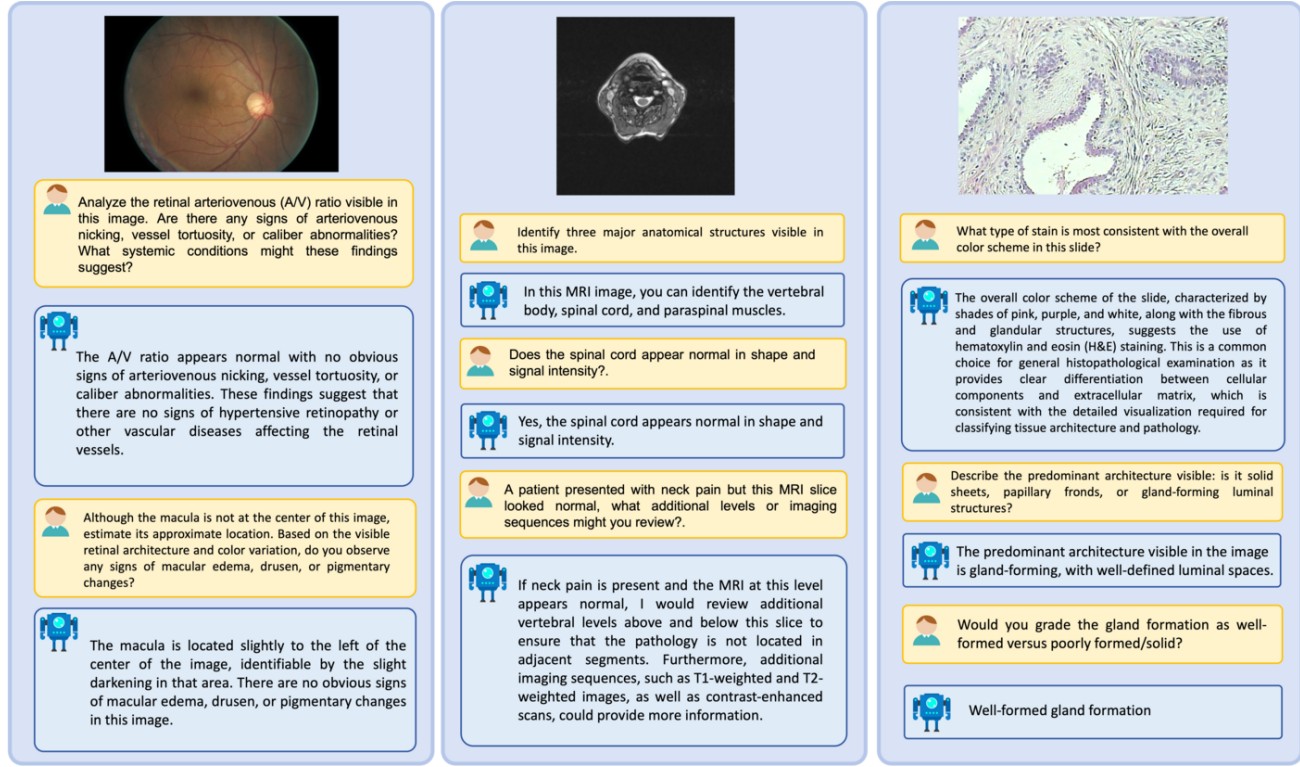

*Figure 14.* Visual question answering examples across different medical imaging modalities.

### A.5.3. MEDICAL IMAGE GENERATION

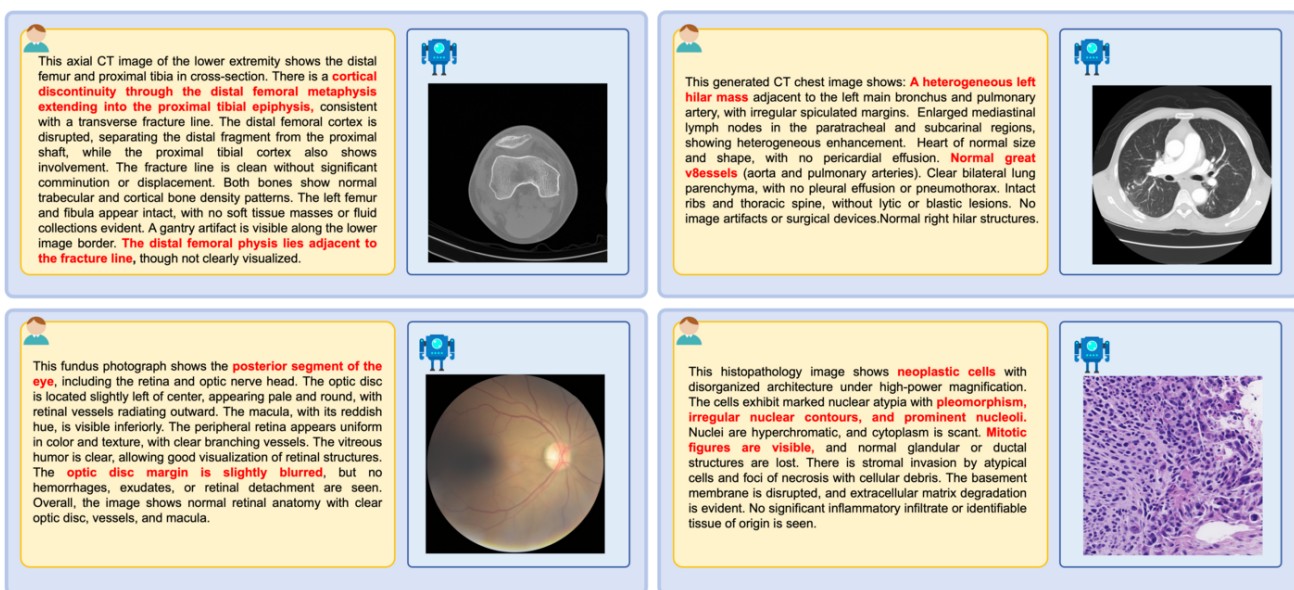

*Figure 15.* Medical image generation examples with text prompts.

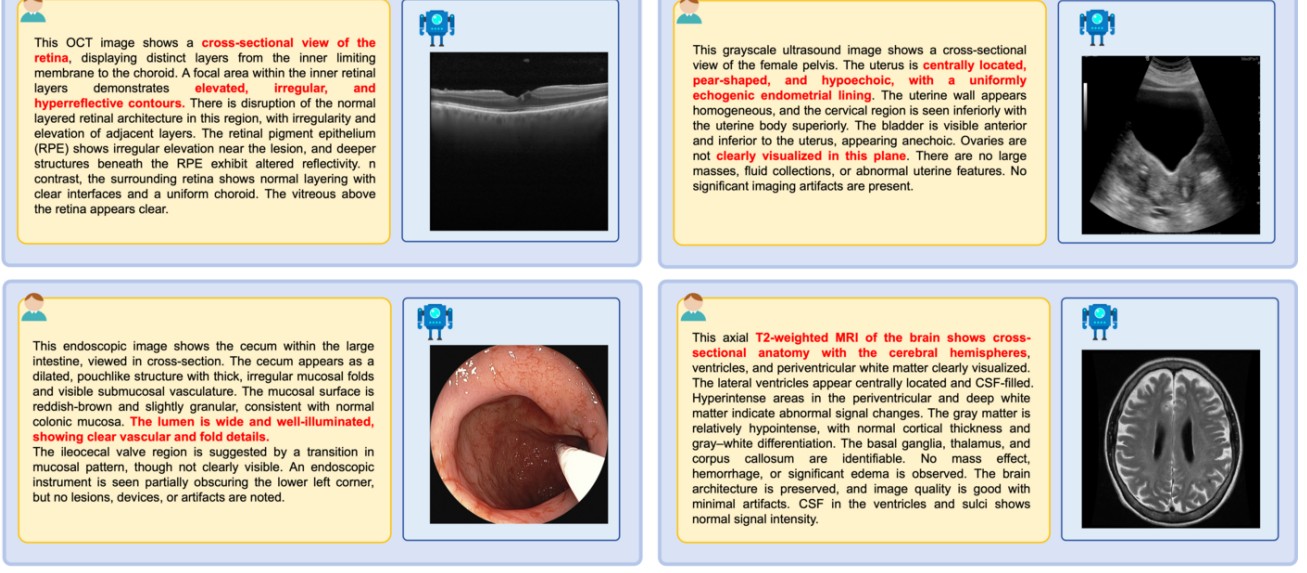

*Figure 16.* Medical image generation examples with text prompts.

A.5.4. INTERLEAVED TASKS

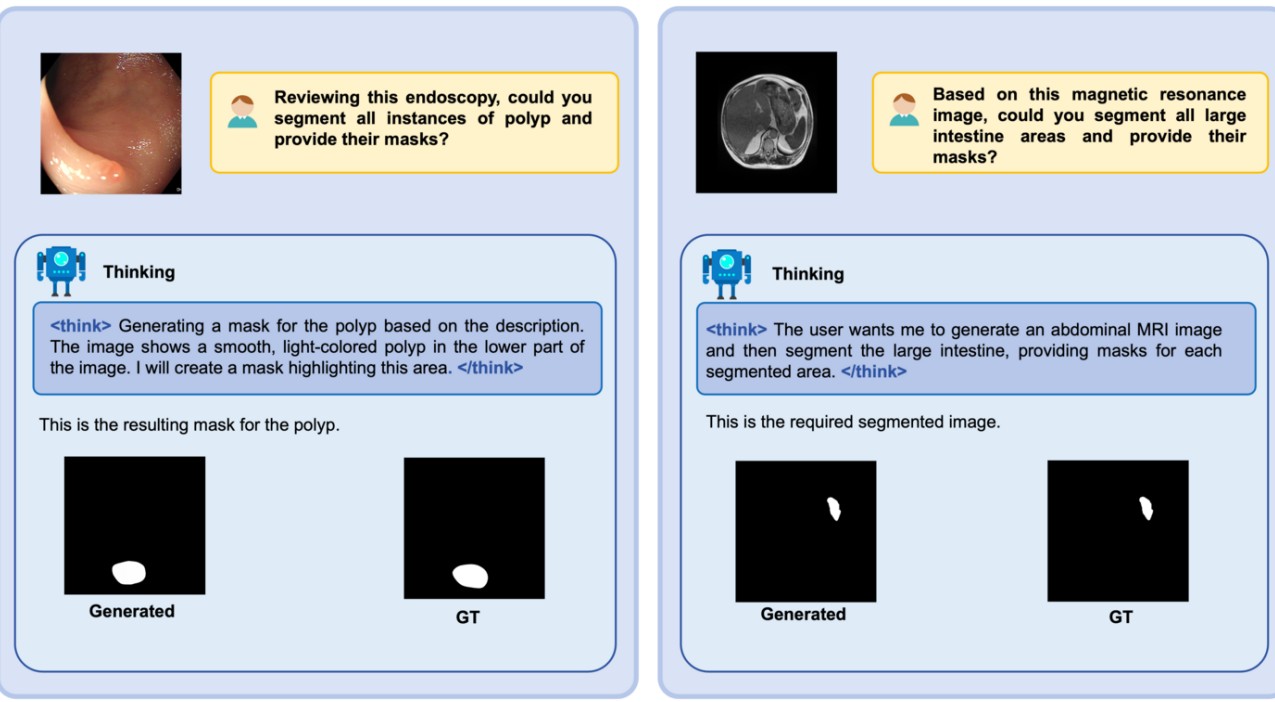

*Figure 17.* Medical Image Prompt Segmentation.

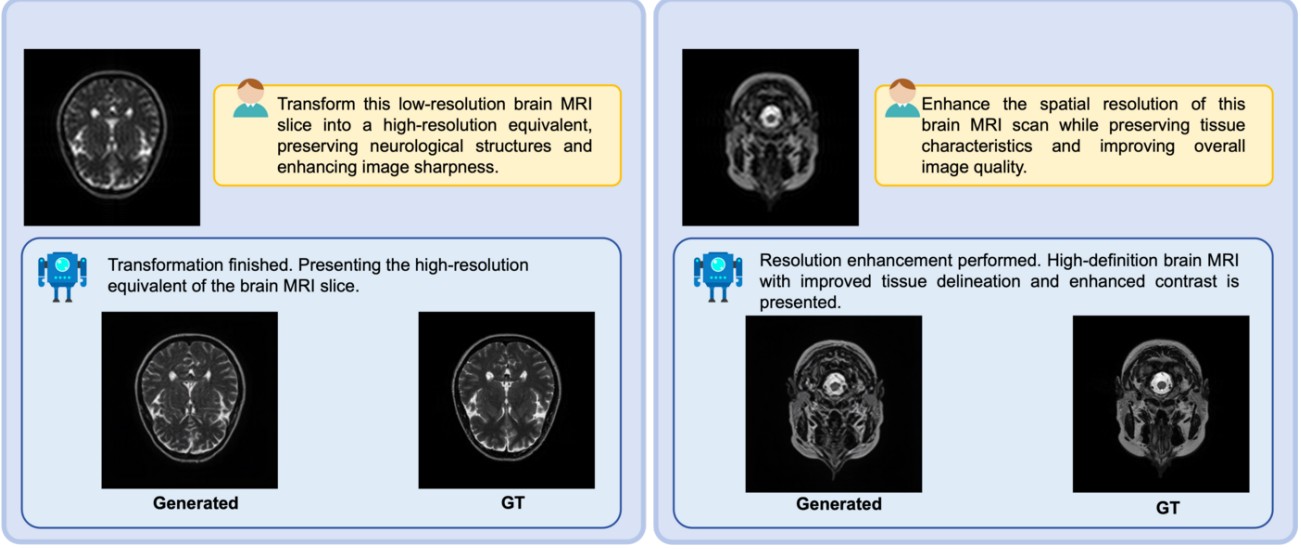

*Figure 18.* Medical Imaging Super-Resolution.

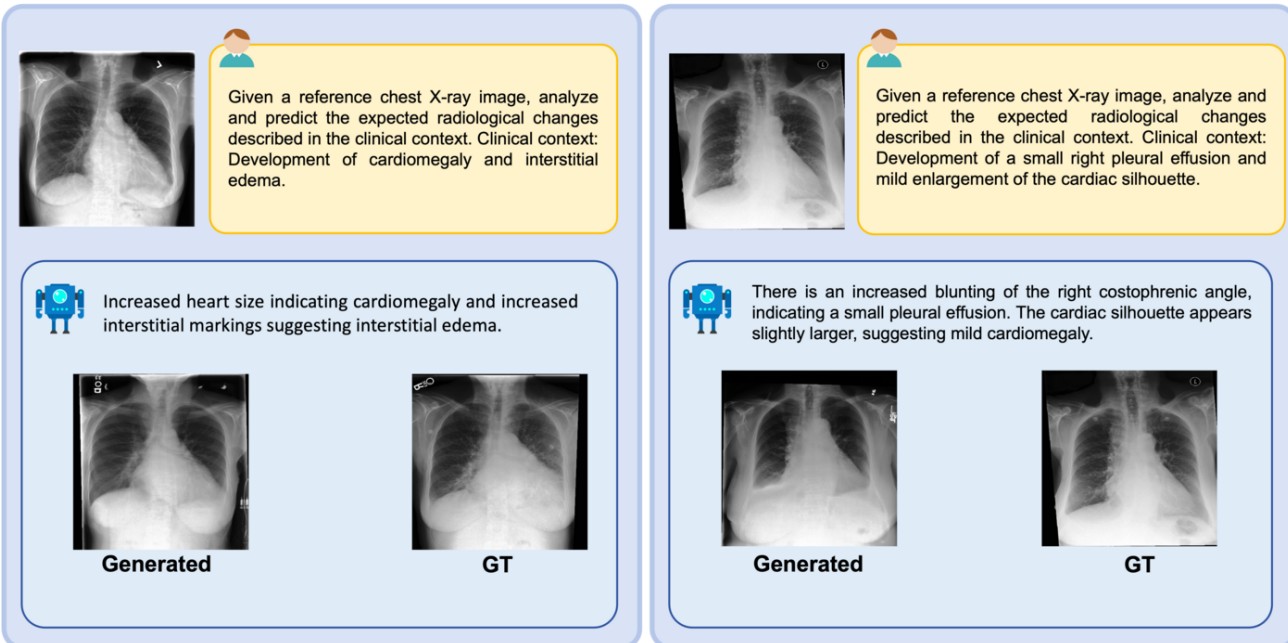

*Figure 19.* CXR Counterfactual Generation.

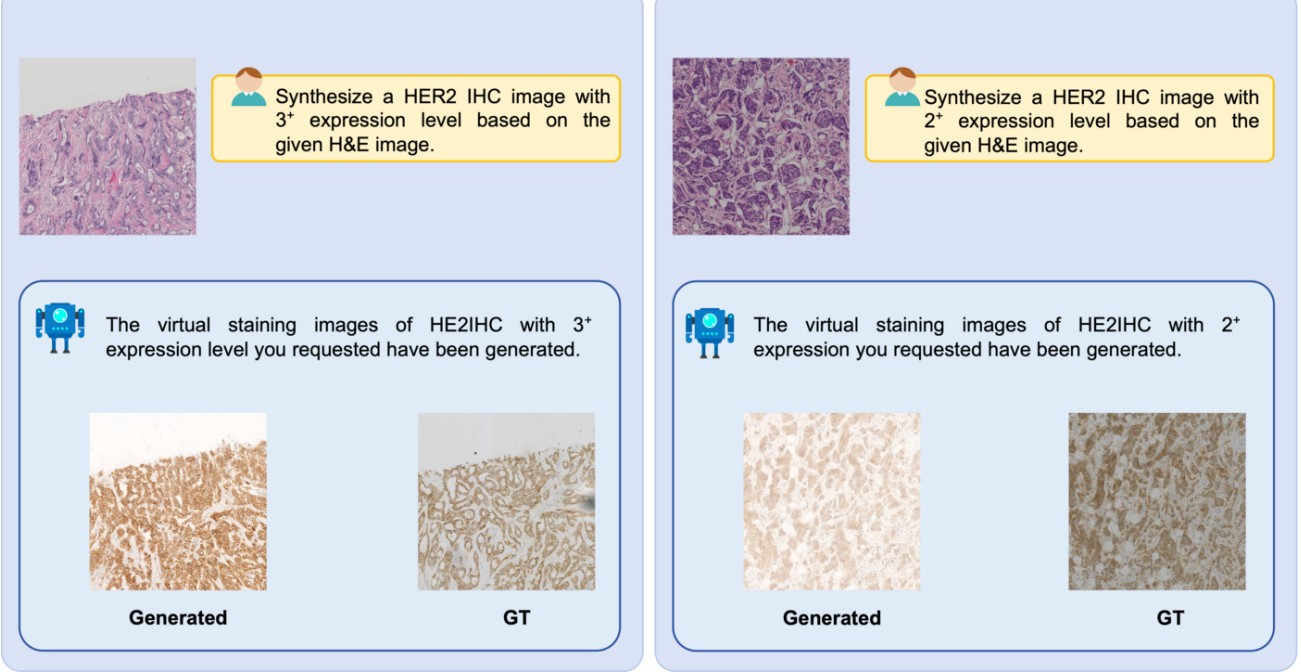

*Figure 20.* Virtual Immunohistochemistry Staining.

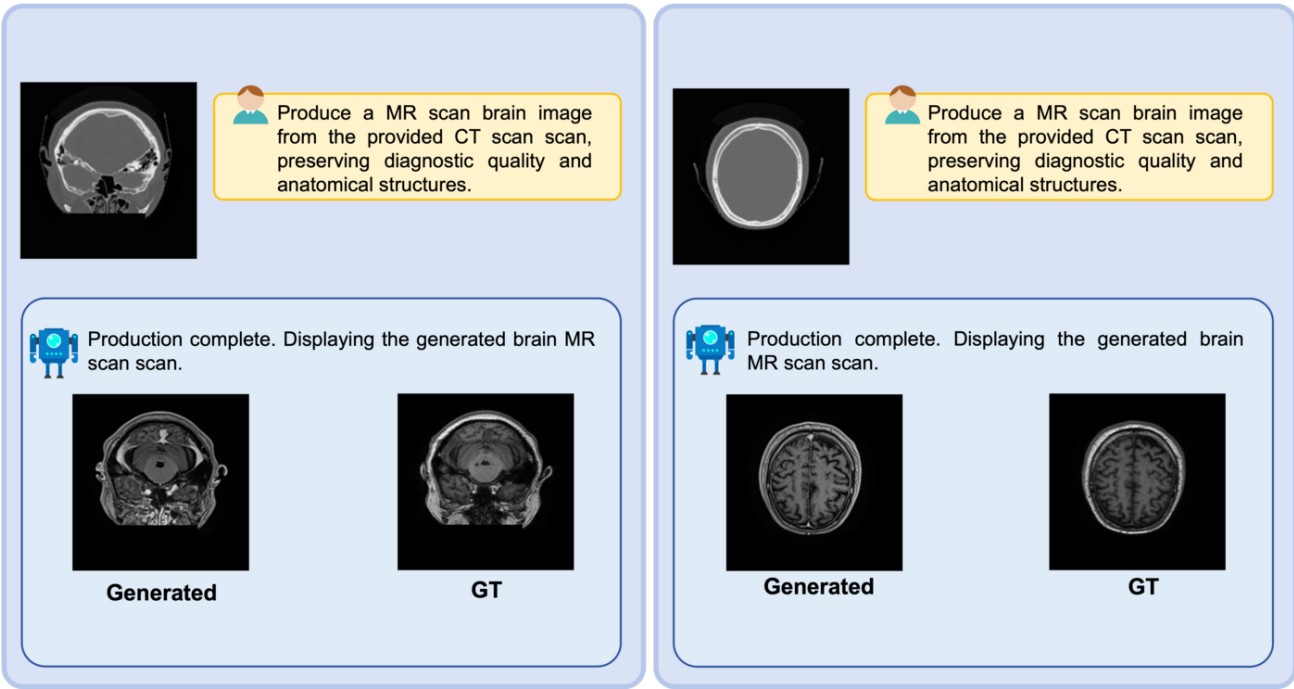

*Figure 21.* CT-MRI cross-modal synthesis.

## A.6. Additional Experimental Results on Downstream Tasks

### A.6.1. BLINDED EXPERT STUDY ON GENERATION REALISM

We conducted a blinded expert study to assess the realism of generated images. We randomly selected 100 images (50 real, 50 generated) and asked two experts to distinguish real from generated samples. The true positive rate for generated images was 64%, the true negative rate for real images was 80%, and the balanced accuracy was 72%, providing evidence that UniMedVL generates visually realistic medical images beyond what automatic metrics capture.

### A.6.2. FINE-TUNING PERFORMANCE ON MEDICAL VQA

While the main paper emphasizes unified zero-shot capability, UniMedVL can also adapt effectively to downstream medical VQA settings through task-specific fine-tuning.

*Table 16.* **Fine-tuning results on medical VQA benchmarks.** Accuracy is reported using the VLMEvalKit protocol, with accuracy for both open-set and closed-set questions.

| Model | VQA-RAD | SLAKE | PathVQA |
|---|---|---|---|
| Lingshu-7B (Xu et al., 2025b) | 62.7 | 77.0 | 59.6 |
| GMAI-VL (Li et al., 2026) | 66.3 | 72.9 | 39.8 |
| HuatuoGPT-Vision-7B (Chen et al., 2024a) | 53.0 | 49.1 | 32.0 |
| **UniMedVL (zero-shot)** | **61.9** | **75.4** | **53.5** |
| **UniMedVL-finetune** | **66.96** | **91.61** | **62.61** |

Fine-tuning improves performance by +5.06 on VQA-RAD, +16.21 on SLAKE, and +9.11 on PathVQA over zero-shot UniMedVL.

**Aligned protocol comparison.** To enable direct comparison with prior work using recall for open-set and accuracy for closed-set questions, we additionally evaluated under this protocol:

*Table 17.* **Fine-tuning comparison under aligned evaluation protocol**, with recall for open-set and accuracy for closed-set questions.

| Model | VQA-RAD (avg) | SLAKE (avg) | PathVQA (avg) | Overall |
|---|---|---|---|---|
| LLaVA-Med (Li et al., 2023) | 72.64 | 83.43 | 64.06 | 73.37 |
| ExGra-Med (MH Nguyen et al., 2026) | 74.91 | 85.46 | 63.87 | 74.75 |
| **UniMedVL (zero-shot)** | **57.63** | **75.26** | **50.06** | **60.98** |
| **UniMedVL-finetune** | **64.78** | **92.58** | **62.59** | **73.32** |

Under the aligned protocol, UniMedVL-finetune achieves an overall score of 73.32, comparable to LLaVA-Med at 73.37 and ExGra-Med at 74.75. UniMedVL-finetune shows particular strength on SLAKE with a score of 92.58, outperforming both baselines by a notable margin, while remaining competitive on VQA-RAD and PathVQA. The original gap was largely attributable to protocol differences.

### A.6.3. SEGMENTATION TASK COMPARISON

We compare UniMedVL on the segmentation task with existing text-guided segmentation models, including SAM3 (Carion et al., 2025) in the general domain and Medical SAM3 in the medical field.

*Table 18.* **Comparison with segmentation models on 8 segmentation tasks.** Dice scores are reported.

| Model | KiTS | QaTa | BUSI | CVC | Glas | ISIC | Kvasir | REFUGE | Overall |
|---|---|---|---|---|---|---|---|---|---|
| SAM3 (Carion et al., 2025) | 39.28 | 0 | 0 | 0 | 0 | 44.39 | 0 | 60.04 | 17.96 |
| Medical SAM3 (Jiang et al., 2026) | 25.61 | 38.96 | 49.04 | **83.76** | 22.28 | **65.59** | **79.16** | 44.71 | **51.14** |
| **UniMedVL (Ours)** | **13.54** | **24.35** | **14.87** | **35.84** | **52.86** | **48.62** | **35.55** | **55.86** | **33.51** |

Compared to Medical SAM3, UniMedVL still exhibits a performance gap overall, 33.51 vs. 51.14, while it outperforms

general-domain SAM3 at 17.96. This is expected, as Medical SAM3 is a specialized segmentation model, whereas UniMedVL is a unified model that simultaneously supports understanding, generation, and segmentation without task-specific checkpoints.

### A.6.4. CXR LUNG OPACITY IMAGE TRANSLATION

*Table 19.* **Unpaired chest X-ray zero-shot opacity removal translation performance on the RSNA dataset (Pan et al., 2019).** Evaluation metrics: FID and KID, where lower values indicate better performance. **Bold** indicates best performance and underlined indicates second-best performance.

| Model | FID ↓ | KID ↓ |
|---|---|---|
| **Baselines** | | |
| Original CXRs | 81.80 | 0.043 |
| Munit (Huang et al., 2018) | 109.4 | 0.073 |
| Unit (Liu et al., 2017) | 103.2 | 0.061 |
| CycleGAN (Zhu et al., 2017) | 208.3 | 0.216 |
| Uvcgan (Torbunov et al., 2023) | 210.4 | 0.225 |
| Drit (Lee et al., 2018) | 117.6 | 0.087 |
| AAMA-CDA (Ning et al., 2025a) | 67.18 | 0.016 |
| **Unified Models** | | |
| HealthGPT-M3 | 62.19 | 0.031 |
| **UniMedVL[†]** | **35.1** | **0.008** |

(a) Quantitative results

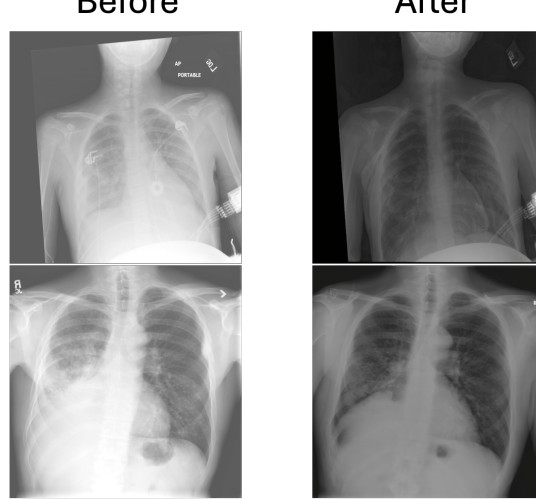

(b) Qualitative examples

A.6.5. MEDICAL REPORT GENERATION

*Table 20.* **Report Generation Performance on MIMIC-CXR Dataset.** NLG metrics (BLEU, METEOR, ROUGE-L) and clinical efficacy metrics (CE) are reported.

| Model | Medical | meteor | bleu4 | rouge1 | rougeL | RaTE |
|---|---|---|---|---|---|---|
| **Medical LVLMs < 10B** | | | | | | |
| MedGemma-1.5-4B-IT | ✓ | 0.2001 | 0.0325 | 0.3062 | 0.2949 | **0.5486** |
| Hulu-Med-7B | ✓ | 0.2241 | 0.0470 | 0.2905 | 0.2800 | 0.5382 |
| Lingshu-7B (Xu et al., 2025b) | ✓ | 0.1995 | 0.0218 | 0.3039 | 0.2923 | 0.5037 |
| MedGemma-4B-IT | ✓ | 0.2021 | 0.0175 | 0.2705 | 0.2556 | 0.5229 |
| HuatuoGPT-V-7B (Chen et al., 2024a) | ✓ | 0.1926 | 0.0072 | 0.2312 | 0.2184 | 0.4769 |
| QoQ-Med-VL-7B | ✓ | 0.1771 | 0.0049 | 0.1724 | 0.1630 | 0.4820 |
| MedVLM-R1-2B | ✓ | 0.1489 | 0.0016 | 0.2111 | 0.1980 | 0.4161 |
| BioMediX2-8B | ✓ | 0.1251 | 0.0005 | 0.2077 | 0.1959 | 0.4432 |
| GMAI-VL (Li et al., 2026) | ✓ | 0.0838 | 0.0153 | 0.1450 | 0.1415 | 0.5088 |
| LLaVA-Med-7B (Li et al., 2023) | ✓ | 0.0827 | 0.0001 | 0.1670 | 0.1559 | 0.4206 |
| **Medical LVLMs > 10B** | | | | | | |
| Hulu-Med-32B | ✓ | **0.2418** | 0.0486 | **0.3218** | **0.3106** | 0.5405 |
| Hulu-Med-14B | ✓ | 0.2304 | **0.0509** | 0.2864 | 0.2726 | 0.5295 |
| Lingshu-32B (Xu et al., 2025b) | ✓ | 0.1839 | 0.0183 | 0.2894 | 0.2778 | 0.5012 |
| MedDr-40B | ✓ | 0.1741 | 0.0184 | 0.2757 | 0.2617 | 0.4845 |
| MedGemma-27B-IT | ✓ | 0.2052 | 0.0147 | 0.2413 | 0.2294 | 0.5015 |
| HuatuoGPT-V-34B (Chen et al., 2024a) | ✓ | 0.1933 | 0.0078 | 0.2314 | 0.2194 | 0.4770 |
| QoQ-Med-VL-32B | ✓ | 0.1396 | 0.0011 | 0.1446 | 0.1369 | 0.4111 |
| **Medical Comp. & Gen. LVLMs** | | | | | | |
| **UniMedVL (Ours)** | ✓ | **0.2193** | **0.0235** | **0.2843** | **0.2727** | **0.5138** |
| HealthGPT-M3 | ✓ | 0.2080 | 0.0081 | 0.2264 | 0.2137 | 0.4588 |
| HealthGPT-L14 | ✓ | 0.1800 | 0.0051 | 0.2261 | 0.2140 | 0.4647 |
| **General Comp. & Gen. LVLMs** | | | | | | |
| Show-o2 | ✗ | 0.1767 | 0.0066 | 0.2254 | 0.2109 | 0.4418 |
| Bagel | ✗ | 0.1863 | 0.0056 | 0.2119 | 0.2000 | 0.4525 |
| Janus-pro-7b | ✗ | 0.1521 | 0.0025 | 0.2180 | 0.2069 | 0.4483 |

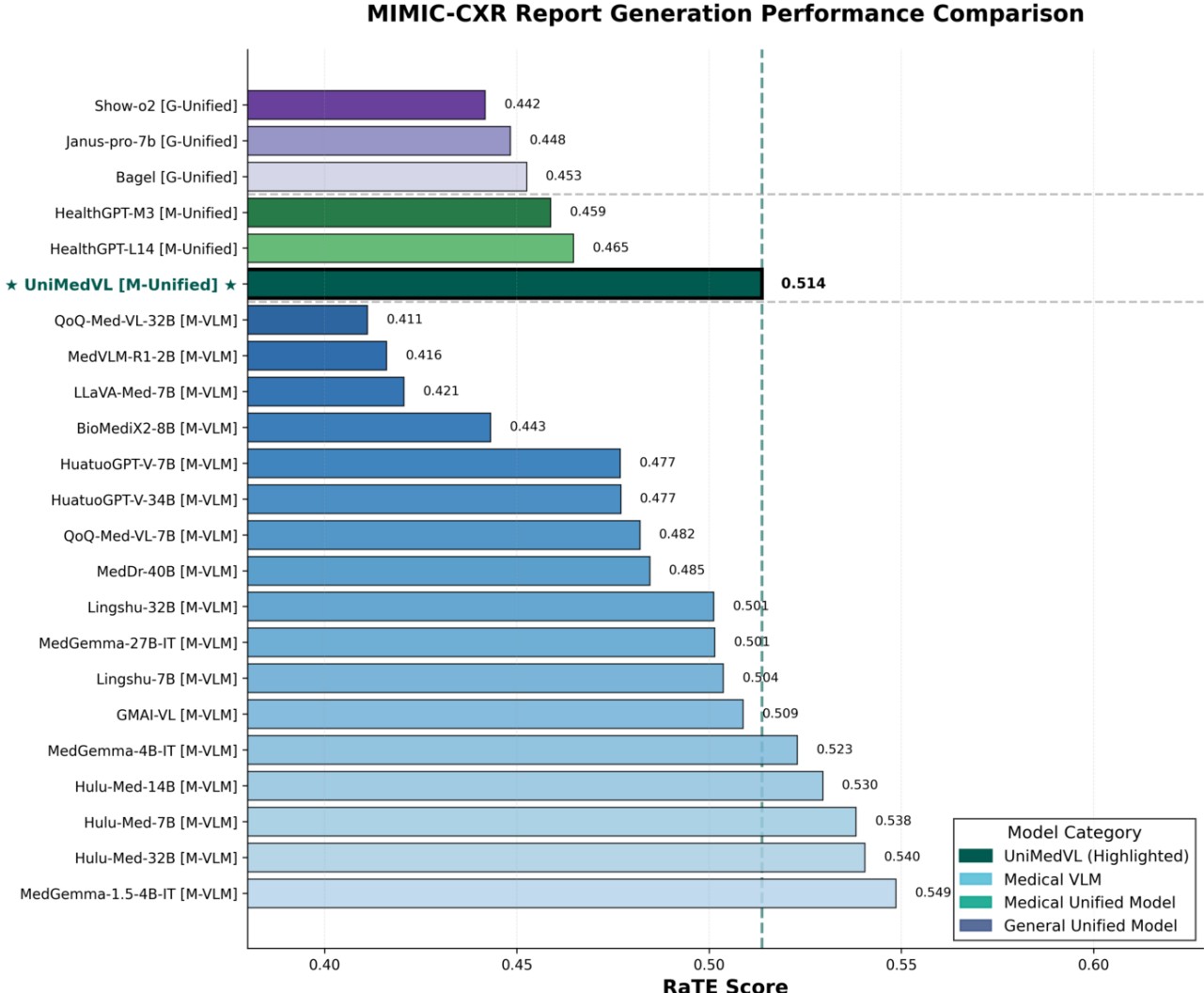

*Figure 22.* **Per-category RaTE score on MIMIC-CXR report generation.** [M-Unified]: Medical Unified models; [M-VLM]: specialized Medical VLMs. Higher is better.

## A.7. Efficiency Analysis

We measure inference efficiency with single expert activation for a controlled comparison. Comparison with BLIP3-o, Janus, and HealthGPT is provided in Table 21.

**Image generation (Table 21(a)).** UniMedVL (14B) requires 40.03 TFLOPs per image and 28.39 GB peak memory, whereas BLIP3-o 8B requires 142.59 TFLOPs per image and 25.55 GB. This corresponds to approximately 3.6x lower compute per image for UniMedVL, while peak memory rises only about 11%, from 25.55 to 28.39 GB, even though UniMedVL uses a dual-encoder and a transformer backbone with distinct FFN layers for understanding and generation tasks and shared self-attention layers, and has almost twice the parameters. UniMedVL's compute cost is also close to that of Janus 7.42B (35.56 TFLOPs per image) while providing a larger unified model.

**VQA on GMAI-MMBench (Table 21(b)).** UniMedVL achieves 25.86 tokens/s with 2.256 TFLOPs per sample and 28.25 GB peak memory, compared with BLIP3-o's 30.40 tokens/s, 9.307 TFLOPs per sample, and 18.21 GB. Thus, UniMedVL attains an approximately 4.1x reduction in FLOPs per sample with comparable throughput (about 85% of BLIP3-o's tokens/s), at the cost of higher peak memory due to the dual-encoder design. Compared to another 14B unified model, HealthGPT-L14 (12.57 tokens/s, 3.009 TFLOPs, 29.22 GB), UniMedVL is roughly twice as fast in throughput and more compute-efficient.

*Table 21.* **Efficiency Evaluation.** Comparison of throughput and computational costs across unified medical multimodal models with batch size 1.

**(a) Image Generation Throughput**

| | | | |
|---|---|---|---|
| *Warm up with 10 images and measure efficiency over 20 images* | | | |
| **Model** | **Parameters** | **FLOPs/Image (TFLOPs)** | **Peak Mem (GB)** |
| Janus | 1B | 10.01 | 5.19 |
| HealthGPT-M3 | 3.8B | 15.22 | 10.23 |
| Janus | 7.42B | 35.56 | 17.10 |
| BLIP3-o | 8B | 142.59 | 25.55 |
| **UniMedVL** | **14B** | **40.03** | **28.39** |

**(b) VQA Understanding Throughput (GMAI-MMBench validation set)**

| | | | | |
|---|---|---|---|---|
| *Warm up with 50 questions and measure efficiency over 150 VQA questions* | | | | |
| **Model** | **Parameters** | **Tokens/s** | **FLOPs/Sample (TFLOPs)** | **Peak Mem (GB)** |
| Janus | 1B | 70.11 | 0.498 | 4.46 |
| HealthGPT-M3 | 3.8B | 22.13 | 1.304 | 8.79 |
| Janus | 7.42B | 52.94 | 1.894 | 14.59 |
| BLIP3-o | 8B | 30.40 | 9.307 | 18.21 |
| **UniMedVL** | **14B** | **25.86** | **2.256** | **28.25** |
| HealthGPT-L14 | 14B | 12.57 | 3.009 | 29.22 |

## A.8. Failure Cases and Analysis

We present representative failure cases of UniMedVL organized by task type: medical image generation, medical image editing, and medical image understanding, with illustrative examples shown in Figures 24 and 25.

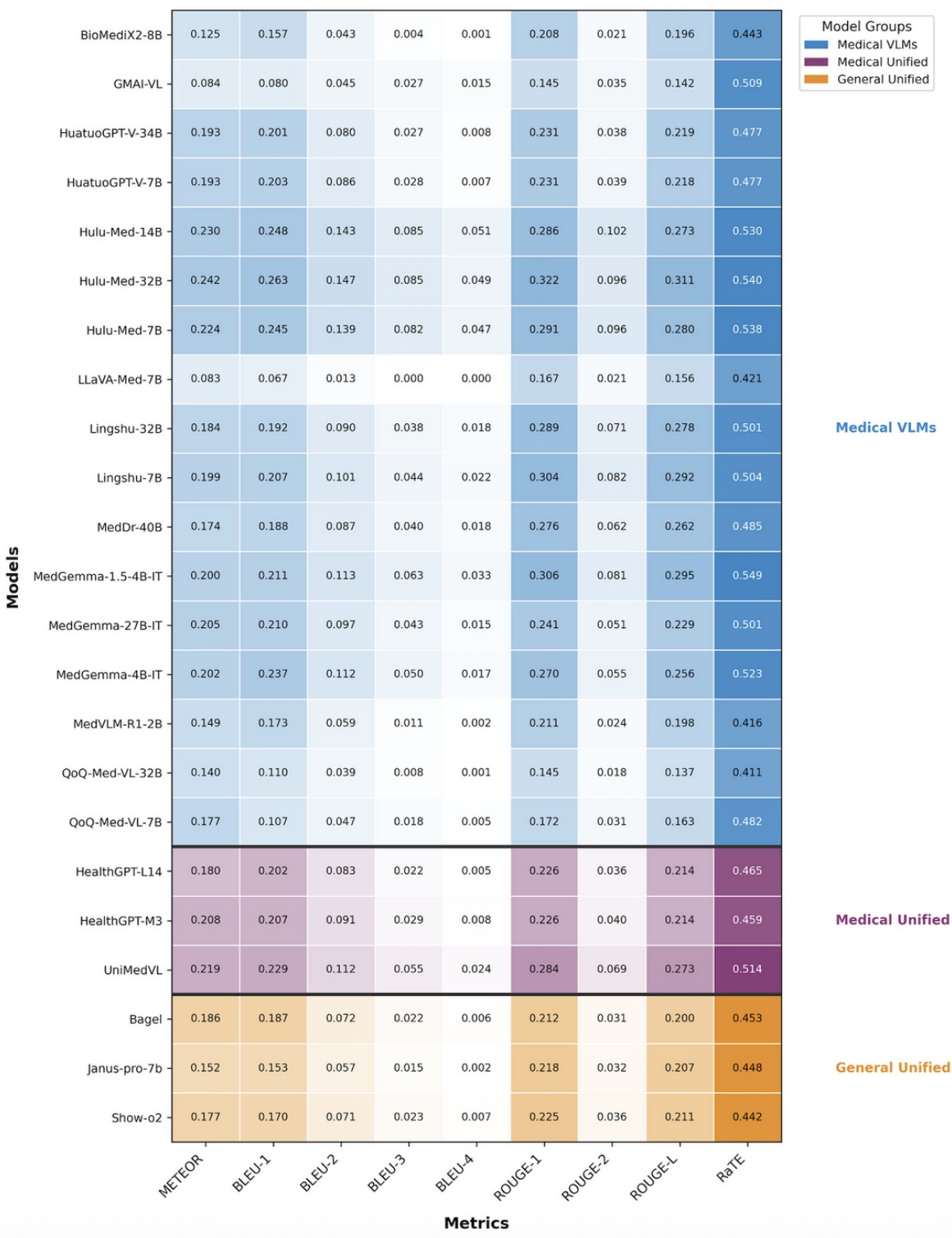

*Figure 23.* **Heatmap breakdown of MIMIC-CXR report-generation performance across multiple metrics.** [M-Unified]: Medical Unified models; [M-VLM]: specialized Medical VLMs. Cell shading encodes the metric value; higher is better unless noted otherwise.

A.8.1. MEDICAL IMAGE GENERATION

**Global appearance and background artefacts.** Although UniMedVL generally produces realistic images across modalities, a characteristic failure mode in text-to-image generation concerns embedded text and annotations (Figure 24). In some synthesised samples, the model hallucinates spurious on-image text or renders partially legible words, labels, or font styles that do not appear in the corresponding real medical images or do not match typical acquisition overlays. These artefacts do not alter the main anatomical content but introduce visually unnatural patterns. As shown in Figure 24, the generated chest X-ray exhibits spurious text overlays (red boxes highlight the artefacts), the ultrasound image contains hallucinated text labels that deviate from standard medical annotations, and the CT scan shows partially corrupted metadata text at the bottom that does not match typical DICOM overlay formatting.

A.8.2. MEDICAL IMAGE EDITING

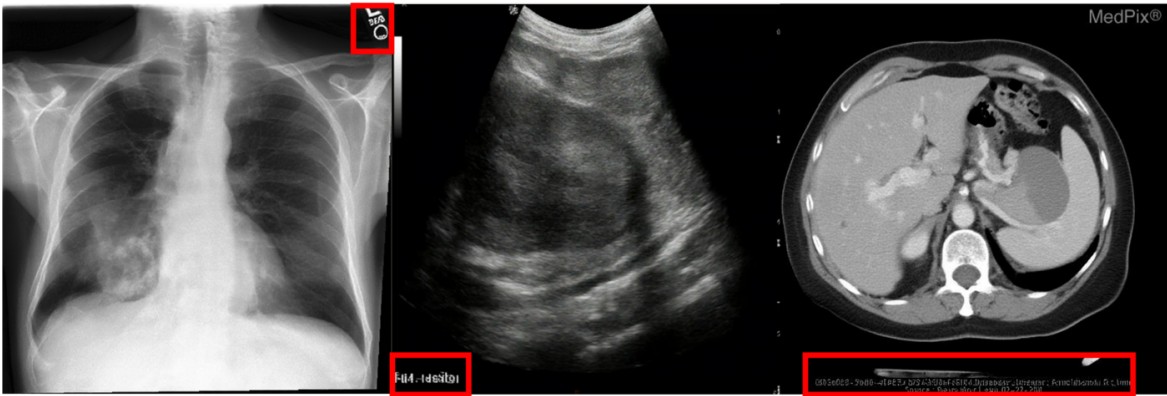

*Figure 24.* **Text and annotation artefacts in medical image generation.** Representative examples in the medical image generation task showing hallucinated or corrupted text elements across different imaging modalities. **Left:** Chest X-ray with spurious text overlay in the upper region (red box). **Center:** Ultrasound image displaying hallucinated text labels that do not conform to standard medical annotation conventions (red box). **Right:** CT scan showing partially corrupted metadata text at the bottom edge that deviates from typical DICOM overlay formatting (red box).

**Structure preservation in generation and editing.** For interleaved editing-style tasks (e.g., virtual staining, super-resolution, cross-modal synthesis, counterfactual generation), UniMedVL does not always perfectly preserve all spatial structures outside the region being semantically edited (Figure 25). For instance, in some counterfactual CXR generations, small devices or lines (e.g., catheters) can become slightly blurred or shifted, even when the main pathological change is correctly applied. Figure 25 illustrates these challenges across three representative cases: in the brain MRI cross-modal synthesis tasks (top two rows), the generated images show subtle structural discrepancies in the cerebellar and temporal regions compared to ground truth, and in the chest X-ray counterfactual generation (bottom row), while the model successfully modifies the target pathological region, minor shifts and blurring are occasionally in the cardiac silhouette boundaries.

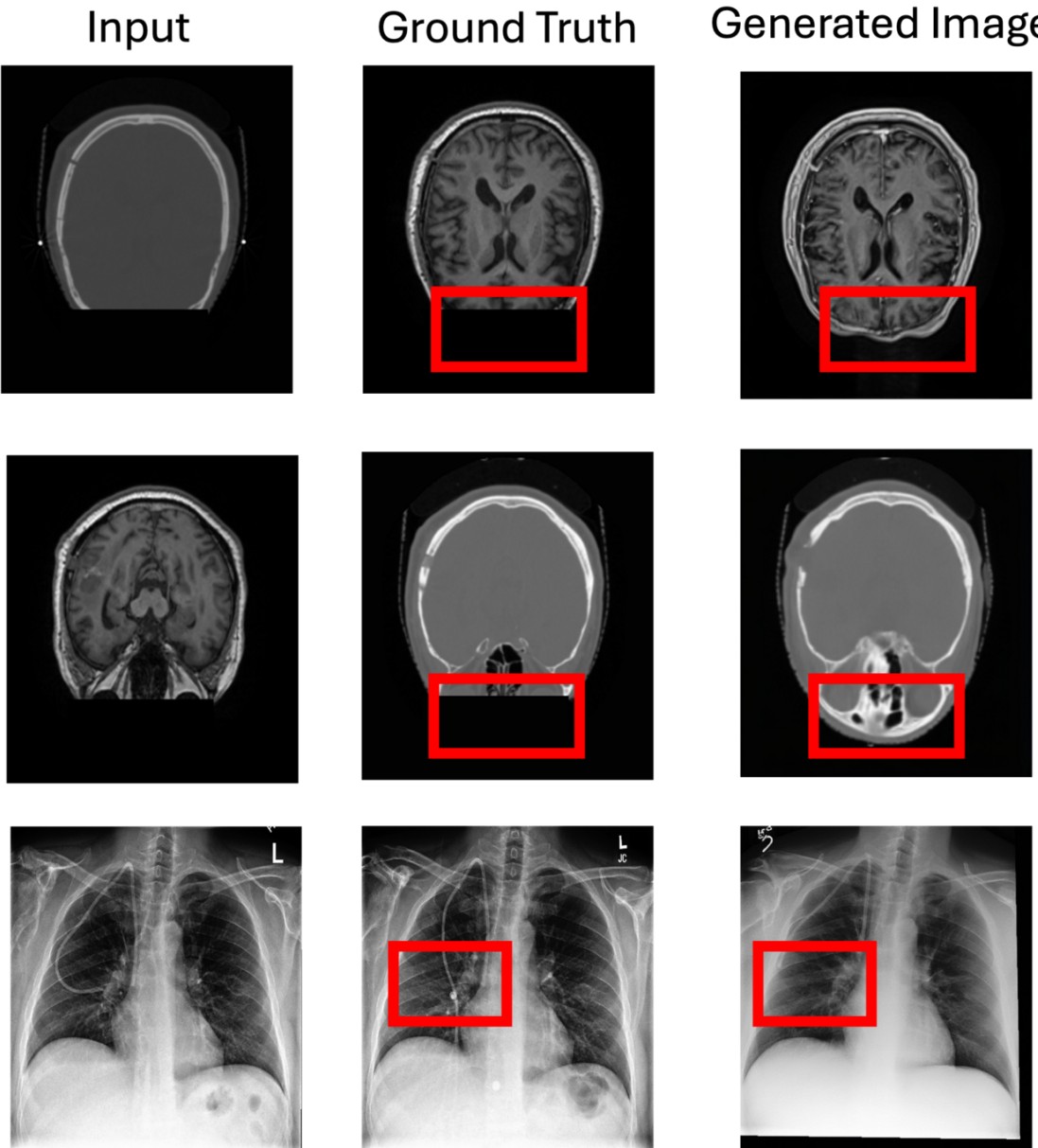

*Figure 25.* **Structural preservation challenges in interleaved generation and editing tasks.** Comparative analysis in the medical image editing task across different medical image translation scenarios. Each row presents a triplet of **Input**, **Ground Truth**, and **Generated Image**. **Top row:** Brain MRI cross-modal synthesis, where the generated image exhibits subtle structural distortions in the cerebellar region compared to the ground truth. **Middle row:** Reverse brain MRI synthesis shows minor misalignment in the temporal lobe structures. **Bottom row:** Chest X-ray counterfactual generation task demonstrating reduction of pleural effusion; while the target pathological modification is applied, the generated image shows slight blurring and positional shifts in the cardiac silhouette and mediastinal borders.

A.8.3. MEDICAL IMAGE UNDERSTANDING

*Table 22.* **UniMedVL performance on GMAI-MMBench validation set.** Accuracy across 18 medical VQA sub-categories.

| Task Category | Accuracy |
| --- | --- |
| Overall | 0.607 |
| Attribute Recognition | 0.659 |
| Blood Vessels Recognition | 0.593 |
| Bone | 0.623 |
| Cell Recognition | 0.513 |
| **Counting** | **0.457** |
| Disease Diagnosis | 0.669 |
| **Image Quality Grading** | **0.440** |
| Microorganism Recognition | 0.793 |
| Muscle | 0.580 |
| Nervous Tissue | 0.925 |
| Organ Recognition – Abdomen | 0.657 |
| Organ Recognition – Head and Neck | 0.845 |
| Organ Recognition – Pelvic | 0.560 |
| Organ Recognition – Thorax | 0.747 |
| **Severity Grading** | **0.372** |
| **Surgeon Action Recognition** | **0.287** |
| **Surgical Instrument Recognition** | **0.310** |
| **Surgical Workflow Recognition** | **0.471** |

**Medical demanding quantitative and fine-grained recognition scenarios.**   In medical demanding tasks that require precise quantitative reasoning or fine-grained visual recognition, the unified model shows clear limitations across multiple task categories. Table 22 presents UniMedVL's performance on GMAI-MMBench validation set, revealing particularly low accuracy on surgical video recognition tasks: Surgeon Action Recognition, Surgical Instrument Recognition, and Surgical Workflow Recognition.

## Multimodal Medical Quantitative Assessment

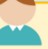 From the image input, please count and tell me the number of lymphocytes shown.

A: "the number of lymphocytes:30"
B: "the number of lymphocytes:7"
C: "the number of lymphocytes:24"
D: "the number of lymphocytes:25"

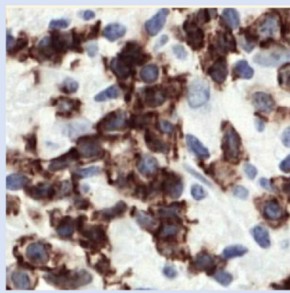

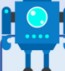 On this immunohistochemical slide, lymphocytes correspond to the brown-stained cells against blue nuclei. Visually estimating, I count ~30 such brown cells across the field. Therefore, among the options provided, I select A: "the number of lymphocytes:30" as the best match to the observed staining pattern.

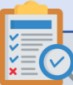 The error arises from coarse visual estimation: clusters of overlapping brown staining are treated as separate lymphocytes, inflating the count. The model fails to delineate individual nuclei and to exclude weak or nonspecific staining, leading to overcounting versus the true value of 24 lymphocytes (option C).

*Figure 26.* **Counting failure in lymphocyte quantification. Green indicates the correct answer, and red indicates our model's prediction.**

## Surgeon Action Recognition

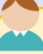 Focusing on the boxed section of this endoscopy video, what surgical action is in progress?
A: "bladder neck dissection"
B: "pulling seminal vesicle"
C: "bagging prostate"
D: "sucking smoke"

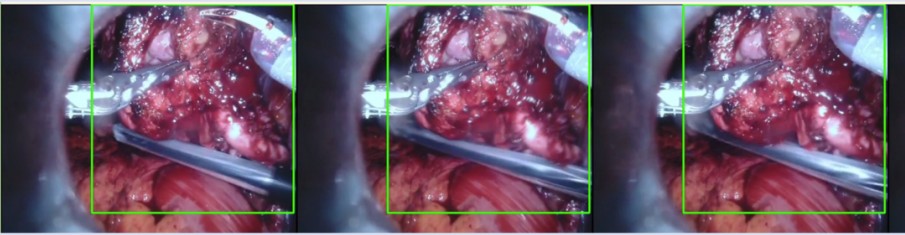

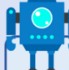 Based on the boxed endoscopic view, I compare visual cues with typical urologic steps. Bladder-neck dissection usually dominates such frames, whereas seminal-vesicle traction, prostate bagging, or smoke suction require more specific devices or anatomy not clearly visible. Given the limited context, I infer the action is bladder-neck dissection **(A)**.

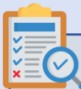 This answer reflects procedural misclassification. The model relies on generic priors ("typical" frames of bladder-neck dissection) instead of carefully inspecting the boxed region for the retrieval bag and specimen, which are diagnostic of prostate bagging. It underutilizes the explicit prompt to recognize the specific surgical step in progress.

*Figure 27.* **Surgeon Action Recognition failure. Green indicates the correct answer, and red indicates our model's prediction.**

## Surgical Instrument Recognition

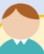 Looking at the region delineated by a rectangle in the provided endoscopy image, which option would be the most target-like?
    A: "bipolar"
    B: "scissors"
    C: "hook"
    D: "irrigator"

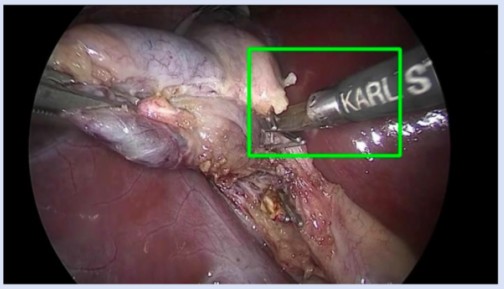

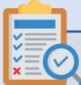 Within the boxed endoscopic region, I compare the visible instrument with typical laparoscopic tools. The elongated tubular appearance suggests a fluid-delivery device rather than a cutting tool or curved hook. Thus, among the options "bipolar," "scissors," "hook," and "irrigator," I judge the instrument most consistent with an irrigator **(D)**

The mistake stems from coarse shape-based reasoning. The model focuses on the shaft's cylindrical form and ignores the distal jaws, which clearly resemble scissor blades rather than an irrigation nozzle. By not attending to fine-grained tip morphology and articulation, it misclassifies the laparoscopic scissors (correct answer B) as an irrigator.

*Figure 28.* **Surgical Instrument Recognition failure. Green indicates the correct answer, and red indicates our model's prediction.**

## Surgical Workflow Recognition

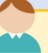 Please identify which of the following options the surgical procedure shown in the image belongs to.
- A: "durotomy"
- B: "nasal corridor creation"
- C: "haemostasis"
- D: "anterior sphenoidotomy"

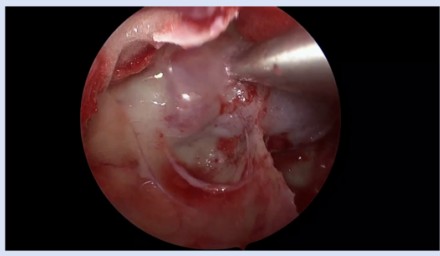

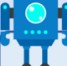 Endoscopic view shows nasal/sinus cavity with instrument working on bony/soft tissue surface. This appearance is more consistent with an endonasal approach to the sphenoid sinus than with durotomy or generic nasal corridor creation or simple haemostatic manoeuvres. Thus I classify it as anterior sphenoidotomy **(D)**.

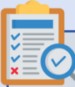 The model overemphasizes coarse anatomical context (endonasal access, apparent sinus wall) and underutilizes dynamic cues that the instrument is primarily controlling bleeding rather than creating a bony opening. Consequently, a supportive step (haemostasis, C) is misinterpreted as a definitive sinus procedure (anterior sphenoidotomy, D)

*Figure 29.* **Surgical Workflow Recognition failure. Green indicates the correct answer, and red indicates our model's prediction.**

# Image Quality Grading

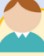

Assessing the boxed area in this endoscopy image, which artifact stands out?
A: "instrument artifacts"
B: "saturation"
C: "blur"
D: "low contrast"

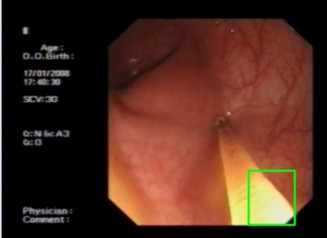

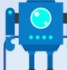

In the boxed region, a sharply bright, featureless patch is visible on the mucosal surface. I interpret this intense highlight as glare caused by the endoscope or another tool, i.e., an instrument-related artifact, rather than blur, low contrast, or general image-quality degradation. Therefore I select option **A**, "instrument artifacts."

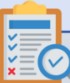

The model conflates extreme brightness with instrument glare, ignoring that no tool contour is visible and that the signal is locally clipped with complete texture loss. These properties are typical of sensor saturation rather than reflection from an instrument surface, so the correct dominant artifact is saturation (option B), not instrument artifacts.

*Figure 30.* **Image Quality Grading failure. Green indicates the correct answer, and red indicates our model's prediction.**

## Severity Grading

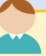 Please determine the severity of the provided image according to the core-needle biopsy grading protocol.
A: "axillary lymph node (aln) metastasis - n0)"
B: "axillary lymph node (aln) metastasis - n+(1-2)"
C: "axillary lymph node (aln) metastasis - n+(>2)"

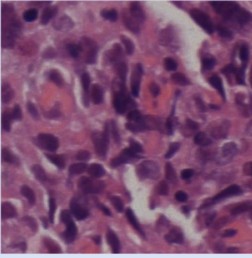

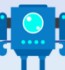 In this core-needle biopsy image, I see numerous metastatic tumor cells filling much of the field. According to the aln grading scheme (n0: none; n+(1–2): limited; n+(>2): extensive), this density appears compatible with involvement beyond two high-power fields, so I classify it as n+(>2) **(C)**.

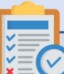 The error arises from misusing the HPF-based protocol: a single densely infiltrated patch is overinterpreted as implying involvement of >2 fields. The model extrapolates tumor extent beyond the visible image instead of restricting grading to sampled HPFs, so the burden actually corresponds to n+(1–2) metastasis (correct answer B).

*Figure 31.* **Severity Grading failure. Green indicates the correct answer, and red indicates our model's prediction.**

## A.9. Information-Theoretic Advantages of Unified Understanding and Generation

**Setup.** Let $(X, T) \sim p_{\text{data}}$ denote paired training data, where $X$ is an image and $T = (T_{\text{in}}, T_{\text{out}})$ is the text sequence split into an input prompt $T_{\text{in}}$ and a target continuation $T_{\text{out}}$. We form two deterministic latent views of $X$: $z_{\text{VIT}} = E_{\text{VIT}}(X)$ (for understanding) and $z_{\text{VAE}} = E_{\text{VAE}}(X)$ (for generation). The unified condition is $C := (T_{\text{in}}, z_{\text{VIT}})$ with embedding $c := g(C)$. All entropies and mutual informations below are computed under the joint distribution induced by $p_{\text{data}}$ and these encoders.

**Training objective (context).** We optimize understanding tasks via next-token prediction (NTP) and generation tasks via rectified flow matching:

$$
\begin{aligned}
\mathcal{L}(\theta) &= \mathcal{L}_{\text{NTP}} + \mathcal{L}_{\text{flow}}, \\
\mathcal{L}_{\text{NTP}} &= -\sum_i \log p_\theta(t_{i+1} \mid t_{\leq i}, z_{\text{VIT}}), \\
\mathcal{L}_{\text{flow}} &= \mathbb{E}_{t, \varepsilon}\left[\left\| v_\theta(\tilde{z}_{\text{VAE}}, t, c) - v \right\|_2^2\right],
\end{aligned}
\tag{6}
$$

where $\tilde{z}_{\text{VAE}}$ and $v$ follow the standard rectified-flow construction.

**Information-theoretic guarantees.** The following statements are distributional and hold independent of the particular optimizer.

**Lemma A.1** (Bayes risk under log-loss). *For any random variables $(Y, X)$, the Bayes-optimal conditional log-loss equals the conditional entropy:* $\inf_q \mathbb{E}[-\log q(Y \mid X)] = H(Y \mid X)$.

*Proof.* For any conditional predictor $q(\cdot \mid X)$,
$$
\mathbb{E}[-\log q(Y \mid X)] = H(Y \mid X) + \mathbb{E}[D_{\text{KL}}(p(\cdot \mid X) \,\|\, q(\cdot \mid X))] \geq H(Y \mid X).
$$
Equality holds if and only if $q(\cdot \mid X) = p(\cdot \mid X)$ almost surely. $\square$

**Proposition A.2** (Benefit of joint modeling). *Let $C = (T_{\text{in}}, z_{\text{VIT}})$ denote the multimodal context. Let $\mathcal{R}_{\text{indep}}$ and $\mathcal{R}_{\text{joint}}$ denote the Bayes-optimal risks for independent and joint models, respectively. Then:*
$$
\Delta\mathcal{R} = \mathcal{R}_{\text{indep}} - \mathcal{R}_{\text{joint}} = I(T_{\text{out}}; z_{\text{VAE}} \mid C) \geq 0.
\tag{7}
$$

*Proof.* By Lemma A.1, the independent risk decomposes as:
$$
\mathcal{R}_{\text{indep}} - \mathcal{R}_{\text{joint}} = \Big[H(T_{\text{out}} \mid C) + H(z_{\text{VAE}} \mid C)\Big] - H(T_{\text{out}}, z_{\text{VAE}} \mid C).
\tag{8}
$$
Applying the chain rule for conditional entropy:
$$
H(T_{\text{out}}, z_{\text{VAE}} \mid C) = H(T_{\text{out}} \mid C) + H(z_{\text{VAE}} \mid T_{\text{out}}, C).
\tag{9}
$$
Substituting equation 9 into equation 8:
$$
\begin{aligned}
\Delta\mathcal{R} &= H(z_{\text{VAE}} \mid C) - H(z_{\text{VAE}} \mid T_{\text{out}}, C) \\
&= I(T_{\text{out}}; z_{\text{VAE}} \mid C) \geq 0.
\end{aligned}
\tag{10}
$$
$\square$

Proposition A.2 quantifies the information-theoretic gain of unified modeling. While factorized modeling implicitly assumes $p(T, z \mid C) = p(T \mid C) p(z \mid C)$, the joint objective captures the cross-modal mutual information $I(T_{\text{out}}; z_{\text{VAE}} \mid C)$. Unified joint modeling is therefore never worse than separate task-specific models, and yields strict improvements whenever the tasks share mutual information—i.e., whenever understanding informs generation or vice versa.

