# A. Appendix

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

| original | zoom in | original | zoom in | original | zoom in |

*Figure 7.* **Preservation of clinically important small lesions in VAE reconstruction.** Side-by-side comparison of original images (top row) and VAE-reconstructed images (bottom row) across three medical imaging scenarios. **Left:** Endoscopy image showing a polyp (highlighted in red box). **Middle:** Dermoscopy image displaying skin lesion with globules (highlighted in red box). **Right:** X-ray image with fractures (highlighted in red box). **Original** denotes the original image, and **Zoom-in** denotes the zoomed-in view of the lesion within the red boxes. The magnified views demonstrate that the FLUX VAE preserves fine structural details essential for clinical interpretation, despite being a general-purpose encoder not specifically fine-tuned for medical imaging.

Figure 7 illustrates that the pretrained FLUX VAE maintains visually discernible fidelity for small pathological features. In the endoscopy image, the polyp's morphology and surface texture remain well preserved in the reconstruction, with boundary definition comparable to the original. For the dermoscopy image, the characteristic globular patterns, key diagnostic features for distinguishing benign nevi from melanoma—are clearly visible in both the original and reconstructed versions. In the X-ray image, the highlighted region and surrounding anatomical structures maintain structural coherence post-reconstruction. These qualitative observations suggest that general-purpose VAE also preserves clinically relevant fine-grained details that are critical for downstream diagnostic tasks within our unified framework.

## A.2. Information-Theoretic Advantages of Unified Understanding and Generation

**Setup.** Let $(X, T) \sim p_{\text{data}}$ denote paired training data, where $X$ is an image and $T = (T_{\text{in}}, T_{\text{out}})$ is the text sequence split into an input prompt $T_{\text{in}}$ and a target continuation $T_{\text{out}}$. We form two deterministic latent views of $X$: $z_{\text{VIT}} = E_{\text{VIT}}(X)$ (for understanding) and $z_{\text{VAE}} = E_{\text{VAE}}(X)$ (for generation). The unified condition is $C := (T_{\text{in}}, z_{\text{VIT}})$ with embedding $c := g(C)$. All entropies and mutual informations below are computed under the joint distribution induced by $p_{\text{data}}$ and these encoders.

**Training objective (context).** We optimize understanding tasks via next-token prediction (NTP) and generation tasks via rectified flow matching:

$$
\begin{aligned}
\mathcal{L}(\theta) &= \mathcal{L}_{\text{NTP}} + \mathcal{L}_{\text{flow}}, \\
\mathcal{L}_{\text{NTP}} &= -\sum_i \log p_\theta(t_{i+1} \mid t_{\leq i}, z_{\text{VIT}}), \\
\mathcal{L}_{\text{flow}} &= \mathbb{E}_{t, \varepsilon}\left[\left\|v_\theta(\tilde{z}_{\text{VAE}}, t, c) - v\right\|_2^2\right],
\end{aligned}
\tag{6}
$$

where $\tilde{z}_{\text{VAE}}$ and $v$ follow the standard rectified-flow construction.

**Information-theoretic guarantees.** The following statements are distributional and hold independent of the particular optimizer.

**Lemma 1 (Bayes risk under log-loss).** For any random variables $(Y, X)$, the Bayes-optimal conditional log-loss equals the conditional entropy: $\inf_q \mathbb{E}[-\log q(Y \mid X)] = H(Y \mid X)$.

Proof. For any conditional predictor $q(\cdot \mid X)$,
$$
\mathbb{E}[-\log q(Y \mid X)] = H(Y \mid X) + \mathbb{E}[D_{\text{KL}}(p(\cdot \mid X) \,\|\, q(\cdot \mid X))] \geq H(Y \mid X).
$$
Equality holds if and only if $q(\cdot \mid X) = p(\cdot \mid X)$ almost surely.

**Theoretical benefits of joint modeling.** We demonstrate that jointly modeling understanding ($T_{\text{out}}$) and generation ($z_{\text{VAE}}$) is strictly superior to treating them as independent tasks (conditionally factorized), given the multimodal context $C = (T_{\text{in}}, z_{\text{VIT}})$.

Let $\mathcal{R}_{\text{indep}}$ denote the sum of Bayes-optimal risks for independent models, and $\mathcal{R}_{\text{joint}}$ denote the risk of the unified joint model. By applying Lemma 1, the reduction in uncertainty is given by:
$$
\begin{aligned}
\Delta \mathcal{R} &= \mathcal{R}_{\text{indep}} - \mathcal{R}_{\text{joint}} \\
&= \left[H(T_{\text{out}} \mid C) + H(z_{\text{VAE}} \mid C)\right] - H(T_{\text{out}}, z_{\text{VAE}} \mid C).
\end{aligned}
\tag{7}
$$
Using the chain rule for conditional entropy, we can expand the joint entropy term as:
$$
H(T_{\text{out}}, z_{\text{VAE}} \mid C) = H(T_{\text{out}} \mid C) + H(z_{\text{VAE}} \mid T_{\text{out}}, C).
\tag{8}
$$
Substituting equation 8 back into equation 7, the redundancy reduction becomes explicit:
$$
\begin{aligned}
\Delta \mathcal{R} &= H(T_{\text{out}} \mid C) + H(z_{\text{VAE}} \mid C) - \left[H(T_{\text{out}} \mid C) + H(z_{\text{VAE}} \mid T_{\text{out}}, C)\right] \\
&= H(z_{\text{VAE}} \mid C) - H(z_{\text{VAE}} \mid T_{\text{out}}, C) \\
&= I(T_{\text{out}}; z_{\text{VAE}} \mid C) \geq 0.
\end{aligned}
\tag{9}
$$

Equation equation 9 quantifies the strict information-theoretic gain of the unified approach. While factorized modeling assumes task independence (implicitly setting $p(T, z \mid C) = p(T \mid C)p(z \mid C)$), our joint objective captures the cross-modal mutual information $I(T_{\text{out}}; z_{\text{VAE}} \mid C)$. Consequently, in the Bayes-optimal sense, unified joint modeling is never worse than separate understanding and generation. It yields strict improvements whenever the tasks are correlated (i.e., when understanding helps generation or vice-versa), converting cross-modal dependence into reduced predictive uncertainty.

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

 (negative values indicate simple approach scores higher; all dimensions scored on 0-5 scale except Modality Match on 0-1 scale).

### A.6.3. DATASET QUALITY COMPARISON ANALYSIS

Figure 18 compares the two generation approaches across all evaluation dimensions. The radar chart (Figure 18a) shows closely aligned performance profiles.

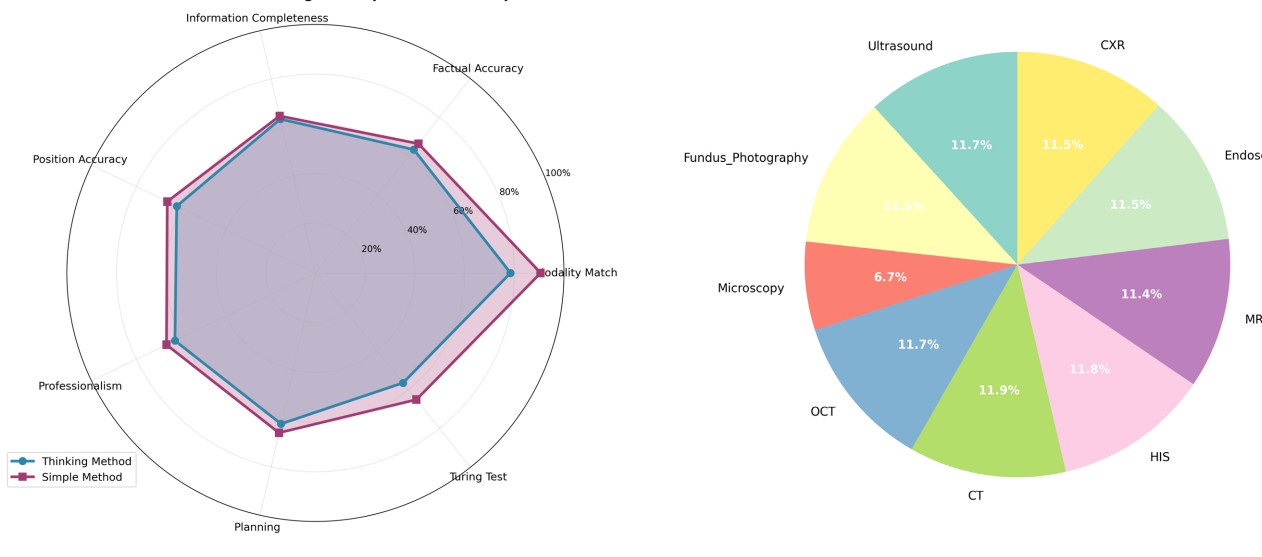

*(a)* Performance comparison: Thinking vs Simple approaches across evaluation dimensions.

*(b)* Medical imaging modalities distribution

*Figure 18.* **Expert validation overview.** (a) Radar chart comparing performance profiles of thinking and simple approaches across all seven evaluation dimensions. (b) Pie chart showing balanced representation across medical imaging modalities, ensuring comprehensive coverage.

### A.6.4. MEDICAL MODALITY-SPECIFIC ANALYSIS

Figure 19 presents modality-specific performance across nine medical imaging modalities. Figure 19a shows statistical comparisons, and Figure 19b displays detailed performance metrics.

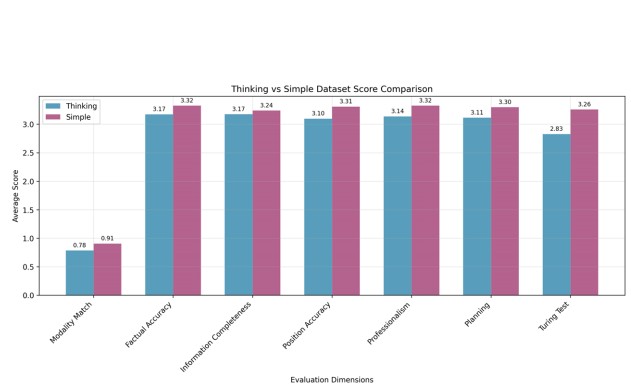

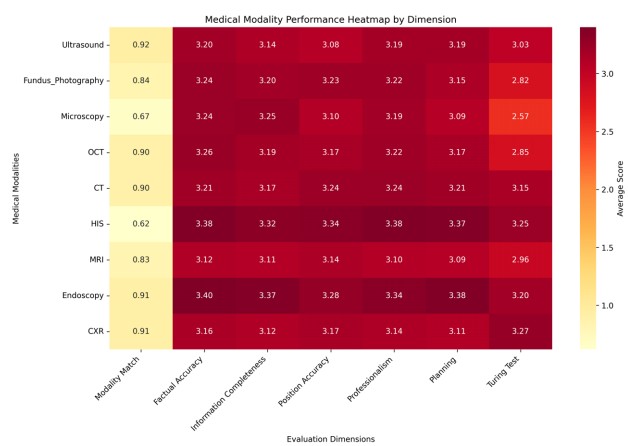

*(a)* Statistical comparison between thinking and simple approaches.

*(b)* Modality-specific performance analysis.

*Figure 19.* **Comprehensive performance analysis.** (a) Bar chart showing mean scores with confidence intervals. (b) Heatmap displaying modality-specific performance scores.

**A.7. Downstream Task Performance**

A.7.1. MEDICAL REPORT GENERATION

*Table 13.* **Report Generation Performance on MIMIC-CXR Dataset. Bold** and underlined text indicate the best performance and second-best performance, respectively.

| Model | Medical | meteor | bleu4 | rouge1 | rougeL | RaTE |
|---|---|---|---|---|---|---|
| **Medical LVLMs < 10B** | | | | | | |
| MedGemma-1.5-4B-IT | ✓ | 0.2001 | 0.0325 | 0.3062 | 0.2949 | **0.5486** |
| Hulu-Med-7B | ✓ | 0.2241 | 0.0470 | 0.2905 | 0.2800 | 0.5382 |
| Lingshu-7B | ✓ | 0.1995 | 0.0218 | 0.3039 | 0.2923 | 0.5037 |
| MedGemma-4B-IT | ✓ | 0.2021 | 0.0175 | 0.2705 | 0.2556 | 0.5229 |
| HuatuoGPT-V-7B | ✓ | 0.1926 | 0.0072 | 0.2312 | 0.2184 | 0.4769 |
| QoQ-Med-VL-7B | ✓ | 0.1771 | 0.0049 | 0.1724 | 0.1630 | 0.4820 |
| MedVLM-R1-2B | ✓ | 0.1489 | 0.0016 | 0.2111 | 0.1980 | 0.4161 |
| BioMediX2-8B | ✓ | 0.1251 | 0.0005 | 0.2077 | 0.1959 | 0.4432 |
| GMAI-VL | ✓ | 0.0838 | 0.0153 | 0.1450 | 0.1415 | 0.5088 |
| LLaVA-Med-7B | ✓ | 0.0827 | 0.0001 | 0.1670 | 0.1559 | 0.4206 |
| **Medical LVLMs > 10B** | | | | | | |
| Hulu-Med-32B | ✓ | **0.2418** | 0.0486 | **0.3218** | **0.3106** | 0.5405 |
| Hulu-Med-14B | ✓ | 0.2304 | **0.0509** | 0.2864 | 0.2726 | 0.5295 |
| Lingshu-32B | ✓ | 0.1839 | 0.0183 | 0.2894 | 0.2778 | 0.5012 |
| MedDr-40B | ✓ | 0.1741 | 0.0184 | 0.2757 | 0.2617 | 0.4845 |
| MedGemma-27B-IT | ✓ | 0.2052 | 0.0147 | 0.2413 | 0.2294 | 0.5015 |
| HuatuoGPT-V-34B | ✓ | 0.1933 | 0.0078 | 0.2314 | 0.2194 | 0.4770 |
| QoQ-Med-VL-32B | ✓ | 0.1396 | 0.0011 | 0.1446 | 0.1369 | 0.4111 |
| **Medical Comp. & Gen. LVLMs** | | | | | | |
| UniMedVL | ✓ | 0.2193 | 0.0235 | 0.2843 | 0.2727 | 0.5138 |
| HealthGPT-M3 | ✓ | 0.2080 | 0.0081 | 0.2264 | 0.2137 | 0.4588 |
| HealthGPT-L14 | ✓ | 0.1800 | 0.0051 | 0.2261 | 0.2140 | 0.4647 |
| **General Comp. & Gen. LVLMs** | | | | | | |
| Show-o2 | ✗ | 0.1767 | 0.0066 | 0.2254 | 0.2109 | 0.4418 |
| Bagel | ✗ | 0.1863 | 0.0056 | 0.2119 | 0.2000 | 0.4525 |
| Janus-pro-7b | ✗ | 0.1521 | 0.0025 | 0.2180 | 0.2069 | 0.4483 |

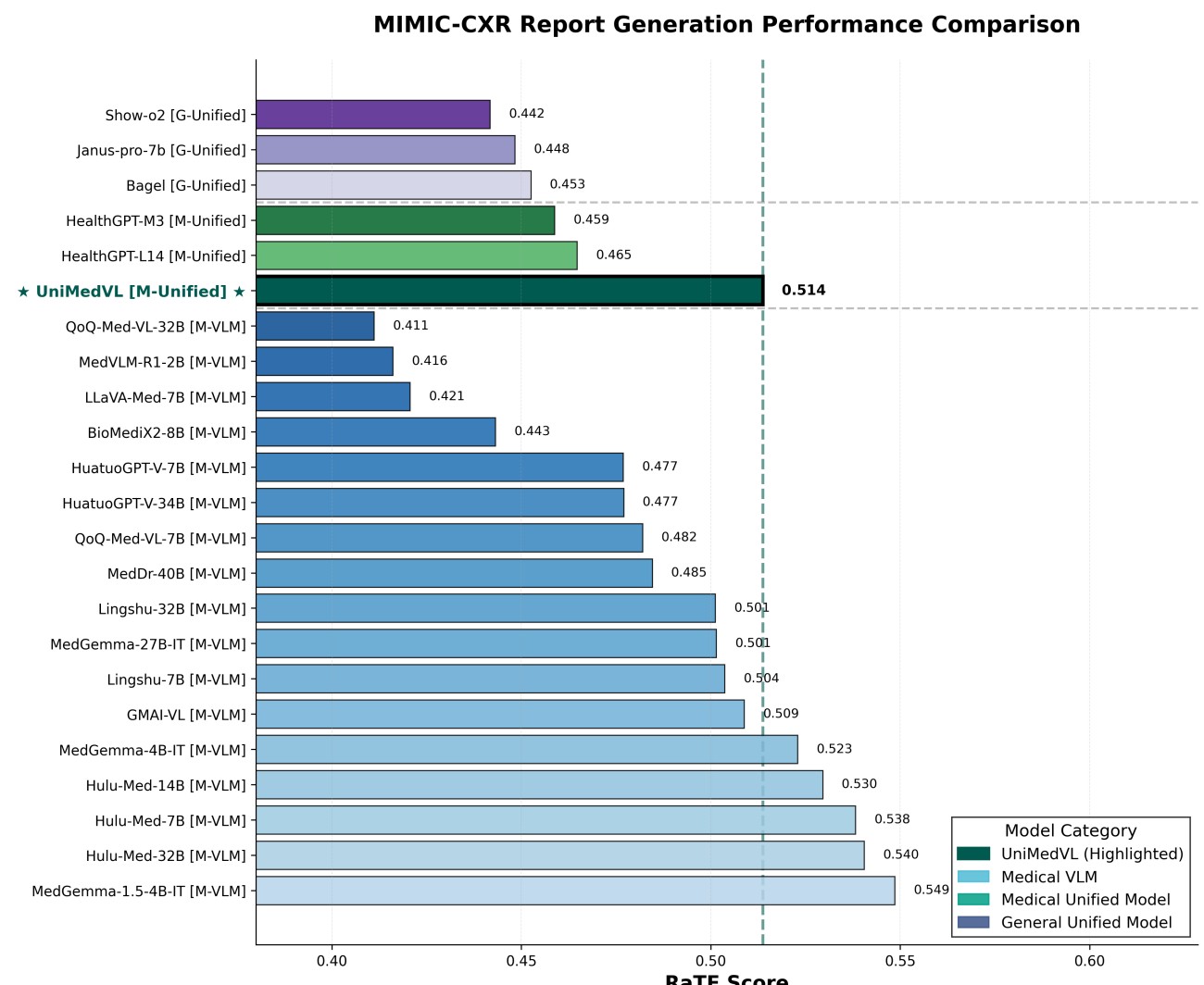

*Figure 20.* **Performance comparison on MIMIC-CXR report generation.** UniMedVL, a Medical Unified ([M-Unified]) model, achieves a RaTE score of **0.514**, surpassing numerous specialized Medical VLM ([M-VLM]) models.

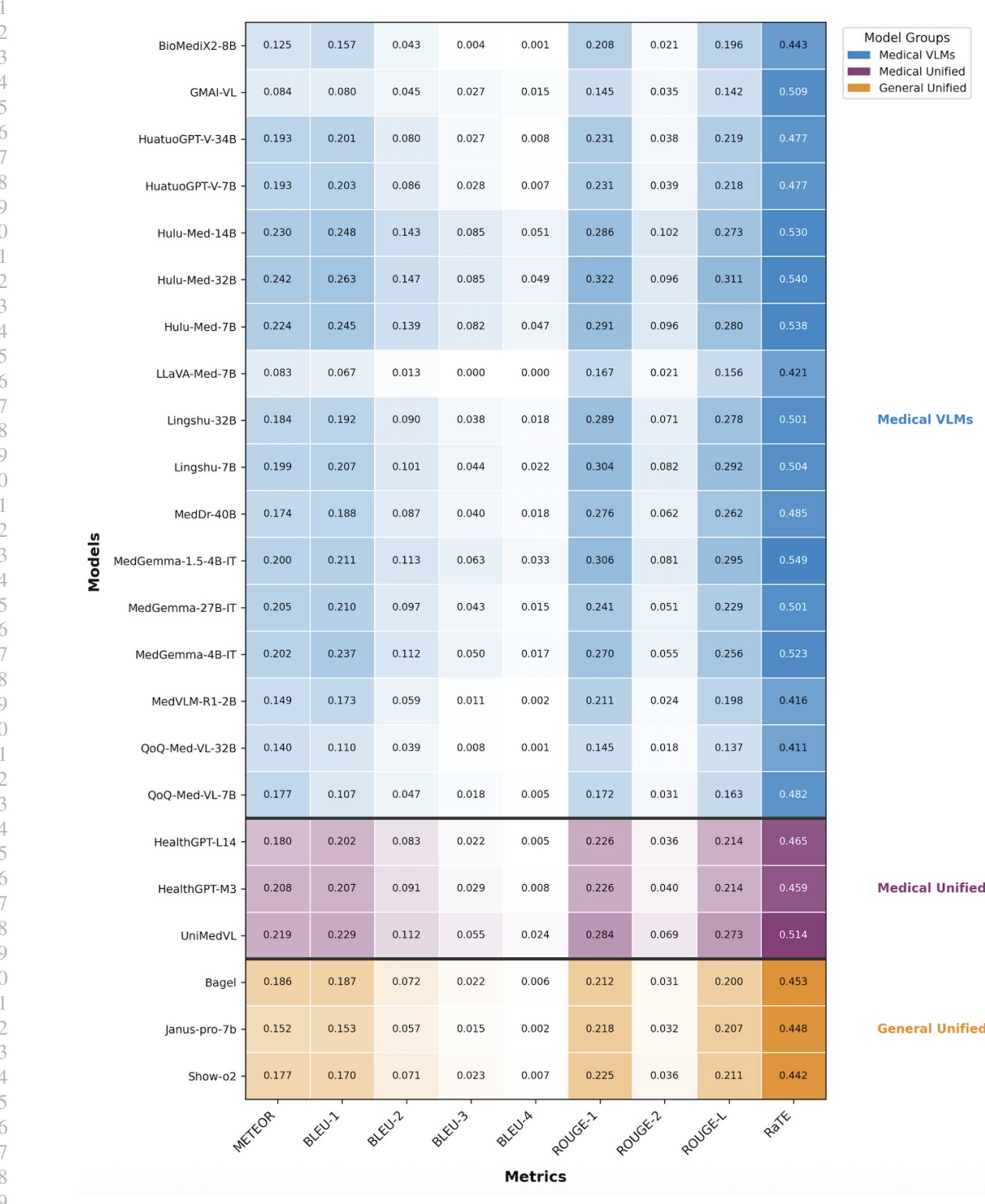

*Figure 21.* **Detailed performance comparison on MIMIC-CXR report generation.** The heatmap breaks down performance across multiple metrics. UniMedVL ([M-Unified]) achieves a RaTE score of **0.514**, a result that is competitive with top-performing specialized Medical VLMs and surpasses other unified models.

## A.8. Failure Cases And Analysis