# OpenReview forum: "UniMedVL: Unifying Medical Multimodal Understanding and Generation through Observation-Knowledge-Analysis"
_ICML.cc/2026/Conference — ICML 2026 regular_

### Official Review · Reviewer_VATK · 2026-03-01

**Soundness:** 3
**Presentation:** 2
**Significance:** 3
**Originality:** 3
**Overall Recommendation:** 4
**Confidence:** 4

**Summary:**

This paper proposes UniMedVL, a unified framework for medical understanding and generation that supports multi-modal image-text inputs and outputs. UniMedVL develops an Observation-Knowledge-Analysis framework to address the conflict between understanding and generation tasks. Extensive experiments show that UniMedVL achieves performance comparable or superior to specialized models across tasks and modalities, highlighting its strong potential as a unified clinical solution.

**Compliance With Llm Reviewing Policy:**

Affirmed.

**Final Justification:**

I am satisfied with the rebuttal, and I'm gonna keep my score as weak accept.

**Key Questions For Authors:**

Overall, I regard the work to be substantial. However, a few issues require improvement.

1. The authors should revise the motivation and method description, particularly the formulation of progressive training as knowledge integration. This framing makes the paper harder to understand and does not appear well justified.
2. Unifying understanding and generation for medical images is an ambitious goal, so the paper should include more detailed comparisons to clarify the capability boundary of the model.

I view the minor issues as suggestions on presentation and completeness.  It is understandable if the authors cannot address all minor issues due to time constraints.

**Limitations:**

1. Although UniMedVL proposes a unified architecture to address understanding and generation, it mainly handles these two tasks separately. Whether it can support a truly unified setting that requires both capabilities simultaneously remains to be determined.
1. The inference speed of UniMedVL is not reported, so its practical usability remains unclear.

**Strengths And Weaknesses:**

Strength:
1. This paper introduces a new dataset, UniMed-5M that covers diverse understanding and generation tasks and can facilitate future research.
2. The paper proposes the UniMedVL architecture to solve medical understanding and generation within a unified framework, without requiring separate checkpoints for different task types.
3. UniMedVL shows strong performance across a range of tasks, achieving results that are comparable to, and sometimes better than specialized models on both understanding and generation tasks.

Major Weakness:
1. The motivation appears slightly over-claimed. In many clinical scenarios, the required visual outputs are segmentation or grounding of tissues or lesions. If specialized models perform better, initializing multiple checkpoints or models is not a major challenge for clinicians. Moreover, lightweight vision models may offer both faster inference and better performance in practice.
2. In Section 3.2, the proposed *Knowledge Level* seems to be progressive curriculum training. Calling this stage knowledge may be inappropriate, since the process lacks explicit medical knowledge injection. In the mentioned instruction tuning stage, the input knowledge context $k$ is not explicitly defined or described, which makes the procedure hard to follow.
3. Regarding the interleaved tasks, although both inputs and outputs are multi-modal image-text pairs, in most cases the output text is only a direct response to the instruction, rather than requiring textual understanding and image generation at the same time. This may raise concerns about whether UniMedVL truly supports a understanding and generation at the same time, or whether it performs only one task at a time. I suggest constructing a subset where, given an image-text input pair, the output simultaneously requires understanding, such as an MCQ, and image generation, in order to evaluate the integrated capability.
4. As discussed in point 1, a unified model may not be the most urgent need in clinical practice. To better assess applicability, including runtime comparisons would help clarify the trade-off between performance and efficiency.

Minor Issues
1. In Section 3.1, the quality control part introduces extra mathematical notation and formula explanations, which makes the process harder to follow. A textual description appears sufficient.
2. The comparisons focus on open-source models only. Including some closed-source models, such as GPT-5 and Gemini-3, can provide a more comprehensive view of UniMedVL's capability.
3. For some interleaved tasks, such as medical image prompt segmentation, it is helpful to compare with specialized referring or reasoning segmentation methods, for example a simple comparison with SAM3. UniMedVL may underperform specialized models, but such comparisons can better clarify its capability boundary.
4. In the current manuscript, some tables, such as Table 2, can be replaced by plots, which may make UniMedVL's performance across tasks and domains easier to interpret.

---

> ### Author Rebuttal · Authors · 2026-03-31
>
> Thank you for the careful and constructive review.
>
> > **Weakness 1: Motivation for unifying medical image understanding and generation.**
>
> **A1.** We agree that our original motivation can be read as over-claimed, and we will revise the manuscript accordingly. UniMedVL is not intended to replace specialist models in narrow clinical workflows where a dedicated segmentation, grounding, or classification model may be more accurate or efficient. Rather, our goal is to study whether a unifieid model paradigm can support mixed medical interactions requiring both textual and visual outputs within one unified interface.
>
> We therefore position this work as a study of the feasibility and value of unified medical multimodal modeling, not as a claim that unified models are already the best solution for every clinical setting. Specialist and lightweight models may remain preferable for some individual tasks, and we will revise the introduction to make this scope clearer.
>
> > **Weakness 2 \& Q1: Clarification of the Knowledge Level and the variable $k$.**
>
> **A2.**
> Progressive Curriculum Learning. At the knowledge level, we apply a three-stage Progressive Curriculum Learning strategy to transfer the medical multimodal knowledge contained in UniMed-5M to the pretrained base model (e.g., Bagel), with Stage 1 - Foundation Training and Stage 2 - Instruction Tuning injecting image understanding and generation knowledge through supervised finetuning, and Stage 3 - Unified Multimodal Training enhancing the synergy between understanding and generation by finetuning on interleaved task data.
>
> Meaning of the knowledge context k. The knowledge context k refers to contextual information related to the current query, such as the patient’s medical history or medical image modality information.
>
> > **Weakness 3: Evaluation on Interleaved tasks.**
>
> **A3.** We agree that the current interleaved benchmark mainly evaluates tightly coupled understanding-generation interactions, rather than a fully general setting in which every sample jointly requires both capabilities.
> At the same time, the current paper already includes one concrete interleaved task that genuinely requires both capabilities: explainable counterfactual generation. In this setting, the model must understand the textual query, synthesize a hypothetical medical image, and provide textual explanation of the radiological changes, results are shown in the table 5 in the paper.
>
> > **Weakness 4 \& Limitations: Inference cost.**
>
> **A4.** Thanks for your suggestion. We added a comparison of the inference cost of UniMedVL with task-specific models such as Lingshu-7B and Lingshu-32B for image-understanding-specific models, and Stable Diffusion 2.1 (SD), finetuned on our proposed datasets, for image-generation-specific models.
> As shown in Table 15, for the image generation task, UniMedVL is slower than the SD model due to UniMedVL's larger number of parameters, but UniMedVL achieves better performance when using the same training dataset.
>  For the medical image understanding task, UniMedVL achieves comparable performance with comparable inference cost to Lingshu-7B, while Lingshu-7B lacks the capability to generate visual outputs.
>
> Table 15. Efficiency evaluation.
> **(a) Image Generation Throughput**
>
>
> | Model | Parameters (B) | FLOPs/Image (TFLOPs) ↓ | Peak Mem (GB) ↓ |
> |---|---:|---:|---:|
> | Stable Diffusion 2.1 | 1.28 | 0.42 | 3.86 |
> | UniMedVL | 14.0 | 40.03 | 28.39 |
>
> **(b) VQA Understanding Throughput (GMAI-MMBench)**
>
> | Model | Parameters (B) | Tokens/s ↑ | FLOPs/Sample (TFLOPs) ↓ | Peak Mem (GB) ↓ |
> |---|---:|---:|---:|---:|
> | Lingshu-7B | 7.6 | 43.891 | 1.758 | 15.547 |
> | UniMedVL | 14.0 | 25.86 | 2.256 | 28.25 |
>
>
> > **MI2: Comparison with Closed-source Models**
>
> **A6.**  We compare three closed-source models. Please refer to  [link](https://anonymous.4open.science/r/icml_2026_unimedvl_rebuttal_materials/).
>
> > **MI3: Comparison on the Segmentation Task**
>
> **A5.** We compare the performance of UniMedVL on the segmentation task with existing text-guided segmentation models, such as SAM3 [1] in the general domain and Medical SAM3 [2] in the medical field.
>  The results are shown in the following table. Compared to Medical SAM3, UniMedVL still exhibits a performance gap, while it outperforms SAM3.
>
> Table 1. Comparison of UniMedVL with segmentation models on 8 segmentation tasks. Dice scores are used to evaluate the performance.
>
> | Model | KiTS | QaTa-COV19 | BUSI | Cvc clinicdb | Glas | Isic2018| Kvasir seg  | REFUGE  | overall |
> |---|---:|---:|---:|---:|---:|---:|---:|---:|---:|
> | SAM3 [1] | 39.28 | 0 | 0 | 0 | 0 | 44.39 | 0 | 60.04 | 17.96 |
> | Medical SAM3 [2] | 25.61 | 38.96 | 49.04 | 83.76 | 22.28 | 65.59 | 79.16 | 44.71 | 51.14 |
> | UniMedVL | 13.54 | 24.35 | 14.87 | 35.84 | 52.86 | 48.62 | 35.55 | 55.86 | 33.51 |
>
> [1] Sam 3: Segment anything with concepts, arxiv, 2025.
>
> [2] Medical SAM3: A Foundation Model for Universal Prompt-Driven Medical Image Segmentation, arxiv, 2026.

---

> > ### Author Rebuttal · Reviewer_VATK · 2026-04-01
> >
> > I am satisfied with the rebuttal, and I'm gonna keep my score as weak accept.

---

> > > ### Author Response · Authors · 2026-04-01
> > >
> > > Dear Reviewer VATK,
> > >
> > > We are glad that our responses have addressed your concerns. Thank you for the constructive suggestions, and we will incorporate the revisions into the next version.
> > >
> > > Best regards,
> > >
> > > The Authors

---

### Official Review · Reviewer_nf6C · 2026-03-08

**Soundness:** 2
**Presentation:** 3
**Significance:** 3
**Originality:** 2
**Overall Recommendation:** 4
**Confidence:** 3

**Summary:**

The paper tackles the challenge of isolated single-capability models in medical AI by proposing a unified framework for both medical multimodal understanding and generation. The authors introduce the Observation-Knowledge-Analysis (OKA) paradigm, curating a 5.6M sample dataset (UniMed-5M) to convert unimodal data into multimodal pairs, and employing a three-stage progressive curriculum learning strategy. The resulting model, UniMedVL, utilizes a dual visual encoder architecture ($E_{ViT}$ and $E_{VAE}$) alongside a Transformer backbone, trained with a joint objective of next-token prediction and rectified flow matching to perform diverse tasks without reloading checkpoints. Empirical results across various benchmarks indicate performance competitive with specialized, single-task models spanning 5 understanding and 8 generation modalities.

**Compliance With Llm Reviewing Policy:**

Affirmed.

**Final Justification:**

The authors’ follow-up provides the kind of direct, medically meaningful subset-level error analysis I was asking for, including modality-wise estimates of pathology-related mismatch and abnormality omission. However, I will maintain my current score, because while this follow-up resolves my main remaining question, it does not substantially change my overall judgment of the paper’s contribution and limitations.

**Key Questions For Authors:**

- How do you justify the architectural novelty of UniMedVL compared to concurrent unified general-domain models like Bagel, aside from its application to the curated UniMed-5M dataset?

- Can you provide generation quality comparisons against state-of-the-art, specialized medical generative models for specific modalities (like CXR or MRI), rather than relying solely on generalist baselines like Bagel?

- Given that expert validation covered only a small fraction of the data, how did you quantify and mitigate the specific hallucination rates or pathological omissions introduced by MedGemma-27b during the automated medical alignment filtering phase?

**Limitations:**

While the authors briefly mention focusing on 2D imaging as a limitation, they fail to adequately discuss the inherent limitations of their chosen evaluation metrics. Standard NLP and vision metrics like BLEU, ROUGE, FID, and SSIM  are notoriously poorly correlated with actual clinical utility, factual completeness, and physician preference.

**Strengths And Weaknesses:**

## Strengths:

- The motivation to unify medical image understanding and generation within a single, checkpoint-free architecture is highly relevant and addresses a legitimate bottleneck in clinical AI workflows.

- The curation of the UniMed-5M dataset, which reformats extensive unimodal data into structured multimodal pairs across 8 modalities, represents a substantial engineering effort that could benefit the broader research community.

## Weaknesses:

- **Lack of Architectural Novelty:** The core architecture, which relies heavily on dual visual encoders ($E_{ViT}$ and $E_{VAE}$) connected to a Transformer backbone , appears highly derivative of existing general-domain unified models (e.g., Bagel, which is cited , and Janus ). The paper fails to articulate any domain-specific architectural innovations beyond retraining an existing paradigm on medical data.

- **Weak Baseline Comparisons for Generation:** In Table 2, the generation FID scores are compared against only two models: LlamaGen-MediTok and Bagel. Omitting comparisons against leading medical-specific generative diffusion models (e.g., MedSegDiff, RoentGen) severely undermines the claim that UniMedVL matches specialized models in generation quality.

- **Opaque Dataset Quality Control:** While a "Quality Control Pipeline" is presented , the heavy reliance on MedGemma-27b for alignment scoring  introduces a critical risk of inheriting LLM-induced biases or hallucinations. The expert validation involved only a 5% subset, which is statistically insufficient to guarantee the clinical reliability of a 5.6M sample training corpus.

---

> ### Author Rebuttal · Authors · 2026-03-31
>
> Thank you for the thoughtful review. We address these points below.
>
> > **Weakness 1 \& Q1: Architectural clarification.**
>
> **A1.** We clarify that UniMedVL adopted the Bagel architecture, and we will revise the manuscript to state this more directly. We do not claim a new domain-specific backbone for medical imaging. Rather, the contribution of this work is to show that a single unified architecture can be made effective for medical understanding and generation through medical-specific data construction and training design strategy.
>
> Concretely, this contribution lies in three aspects: **(i)** UniMed-5M, which reformulates diverse medical datasets into a unified multimodal input-output interface; **(ii)** a medical quality-control pipeline for large-scale image-text curation; and **(iii)** a three-stage Progressive Curriculum Learning strategy that gradually equips the model with basic medical multimodal capability, instruction following, and understanding-generation synergy. We will therefore revise the paper so that the originality claim is framed more precisely as a medical data and training contribution, rather than as a new architectural block.
>
> > **Weakness 2 \& Q2: Comparison with specialized medical generative models.**
>
> **A2.** We agree that stronger modality-specific generation baselines are important. Following this suggestion, we further compared UniMedVL with open, modality-specialized medical generative models on matched external subsets, including RetinaLogos for CFP, MedSyn for CT, CheXGen for CXR and PathLDM for HIS:
>
> | Modality | Model | FID ↓ | BioMedCLIP ↑ |
> |---|---|---:|---:|
> | CFP | RetinaLogos[1]| 60.45 | **0.7110** |
> | CFP | **UniMedVL** | **53.20** | 0.7080 |
> | CT | MedSyn[2] | 174.42 | 0.6259 |
> | CT | **UniMedVL** | **73.04** | **0.6960** |
> | CXR | CheXGen[3] | 90.70 | 0.6890 |
> | CXR | **UniMedVL** | **73.04** | **0.7020** |
> | HIS | PathLDM[4] | 241.53 | 0.6171 |
> | HIS | **UniMedVL** | **149.01** | **0.7040** |
>
> [1] RetinaLogos: Fine-Grained Synthesis of High-Resolution Retinal Images Through Captions, MICCAI, 2025.
>
> [2] MedSyn: Text-guided Anatomy-aware Synthesis of High-Fidelity 3D CT Images, IEEE TMI, 2024.
>
> [3] A Generative Foundation Model for Chest Radiography, arXiv, 2025.
>
> [4] Pathldm: Text conditioned latent diffusion model for histopathology, CVPR, 2024.
>
> These results show that UniMedVL is competitive not only with generalist baselines, but also with strong modality-specific medical generators on matched settings. We will include these additional comparisons in the revised manuscript.
>
> > **Weakness 3 \& Q3: Quality control, MedGemma-induced bias, and expert validation scope.**
>
> **A3.** We agree that this part should be clarified more carefully. The key point is that MedGemma-27b is used only for alignment scoring during filtering, not to create the retained training targets. The retained pairs remain the original image-text pairs from human-validated open datasets, including expert-annotated medical imaging datasets, peer-reviewed articles, and authentic radiology reports, rather than synthetic captions produced by MedGemma. Specifically, we use a multi-stage filtering pipeline: coarse filtering, a medical alignment stage combining caption-based semantic similarity with MedSigLIP image-text matching, and a 5\% stratified expert audit across modalities and anatomical groups. On the audited subset, the curated data achieved mean quality above 0.85 across seven clinically grounded criteria, with inter-rater agreement above 0.80 among five experts. However, we agree that this does not prove zero hallucination risk or provide a direct hallucination-rate estimate for the full 5.6M corpus. We will therefore revise the wording to present expert validation as a quality audit, not as a guarantee of full clinical reliability.
>
> > **Limitations: Evaluation metrics.**
>
> **A4.** We agree that standard metrics such as BLEU, ROUGE, FID, and SSIM are imperfect proxies for clinical utility, factual completeness, and physician preference. We use these quantitive metrics because they are standard in prior benchmarks and allow reproducible, task-aligned quantitative comparison across models, but they do not replace expert judgment in clinical settings. We will make this limitation more explicit in the revision.

---

> > ### Author Rebuttal · Reviewer_nf6C · 2026-04-04
> >
> > The rebuttal addresses several of my concerns well. In particular, I appreciate the clarification that UniMedVL does not claim a new medical-specific backbone and instead makes its contribution through medical data construction, quality control, and curriculum-based training. This makes the originality claim more precise. I also appreciate the newly added comparisons against modality-specialized medical generative models, which substantially strengthen the empirical case that UniMedVL is competitive beyond generalist baselines.
> >
> > My remaining concern is mainly about dataset quality control and the risk of MedGemma-induced bias or omission during filtering. The clarification that MedGemma-27b is only used for alignment scoring, rather than generating retained targets, is helpful, and the additional details about stratified expert auditing, average quality scores, and inter-rater agreement improve confidence in the curation pipeline. However, the rebuttal also acknowledges that it does not provide a direct estimate of hallucination rates or pathological omission rates for the full corpus. Since this issue is central to the reliability of a large-scale medical training set, I still view it as only partially resolved.
> >
> > My follow-up question is therefore: if possible, can the authors provide a more direct estimate of medically relevant filtering errors on the audited subset, such as omission, incorrect alignment acceptance, or pathology-related mismatch rates, ideally broken down by modality?

---

> > > ### Author Response · Authors · 2026-04-06
> > >
> > > We are encouraged by the reviewer's continued engagement and this constructive follow-up question, and we provide a targeted analysis below.
> > >
> > > As noted in A3, MedGemma-27b serves as a scoring mechanism rather than a content generator; the residual errors reported below therefore reflect imperfect filtering decisions in pair selection rather than hallucinated or fabricated content. Following the reviewer's suggestion, we released our reserved audit results on the stratified subset. To better characterize these residual errors in medically meaningful terms, we focused on two clinically oriented review criteria: **(i) pathology-related mismatch** and **(ii) omission of the abnormality**.
> > >
> > > | Modality | Pathology-related mismatch | Omission of the abnormality |
> > > |---|---:|---:|
> > > | CFP | ~10.0% | ~4.0% |
> > > | CXR | ~24.0% | ~12.0% |
> > > | CT | ~22.0% | ~6.0% |
> > > | HIS | ~2.0% | ~2.0% |
> > > | MRI | ~14.0% | ~4.0% |
> > > | OCT | ~18.0% | ~2.0% |
> > > | Ultrasound | ~4.0% | ~2.0% |
> > > | Endoscopy | ~8.0% | ~4.0% |
> > > | **Overall** | **~12.8%** | **~4.5%** |
> > >
> > >
> > > For reference, the real-time diagnostic error rate in clinical radiology practice is reported at 3–5%, with retrospective discrepancy rates averaging approximately 30% [1]. Cognitive errors, where findings are detected but incorrectly interpreted, account for 20–40% of all radiology errors [2]. Our overall abnormality omission rate of ~4.5% is comparable to the established real-time clinical error floor, and the pathology-related mismatch category is analogous to the cognitive error class described in the literature.
> > >
> > > We appreciate that the reviewer's question prompted this additional analysis. We will incorporate this subset-level error analysis into the revised manuscript.
> > >
> > > [1] Lee et al., Cognitive and System Factors Contributing to Diagnostic Errors in Radiology. AJR, 2013.
> > >
> > > [2] Brady, Error and discrepancy in radiology: inevitable or avoidable? Insights into Imaging, 2017.

---

### Official Review · Reviewer_RYVn · 2026-03-13

**Soundness:** 3
**Presentation:** 2
**Significance:** 3
**Originality:** 2
**Overall Recommendation:** 4
**Confidence:** 5

**Summary:**

This paper proposes UniMedVL, a unified medical multimodal model designed to perform both image understanding (e.g., VQA, report generation, disease classification) and image generation (e.g., text-to-image synthesis, cross-modal translation, super-resolution) within a single architecture. The work is organized around a three-level "Observation-Knowledge-Analysis" (OKA) framework. At the observation level, the authors construct UniMed-5M, a dataset of over 5.6 million samples aggregated from 30+ existing medical datasets and reformatted into multimodal input-output pairs across 8 imaging modalities. At the knowledge level, they introduce Progressive Curriculum Learning, a three-stage training pipeline consisting of foundation training, instruction tuning, and unified multimodal training with interleaved tasks. At the analysis level, the UniMedVL model itself adopts a dual-encoder architecture (ViT for understanding, VAE for generation) with a shared Transformer backbone featuring task-specific FFN layers and shared self-attention. The training objective combines next-token prediction with rectified flow matching. The authors provide an information-theoretic argument that joint modeling yields lower Bayes-optimal uncertainty than independent task modeling. Experiments are conducted on 5 medical VQA/understanding benchmarks, 8 modalities for text-to-image generation, and 4 interleaved tasks (staining, super-resolution, cross-modal synthesis, counterfactual generation).

**Compliance With Llm Reviewing Policy:**

Affirmed.

**Final Justification:**

I maintain my score, as I believe it's a good reflection of my review, which is reflected in the discussions.

**Key Questions For Authors:**

Q1. Can you precisely enumerate what architectural modifications, if any, were made to the Bagel framework for the medical domain? If none, please clarify that the contribution is primarily in data and training strategy. This would significantly change how the originality of the work is assessed

Q2. The 80/20 train/test split of UniMed-5M for evaluating generation performance raises concerns about circularity. Could you evaluate generation quality on an entirely held-out external dataset

Q3. Have you conducted or planned any expert radiologist evaluation of the text-to-image generated outputs? FID and BioMedCLIP scores capture distributional similarity and semantic alignment, but they may not capture clinically critical errors

**Limitations:**

The authors have partially addressed limitations: they note the restriction to 2D imaging and acknowledge failure cases in the appendix. However, several important limitations are insufficiently discussed. First, the clinical safety implications of a unified model that generates medical images are not adequately addressed e.g what safeguards prevent the model from generating clinically misleading images that could be mistaken for real diagnostic data?

**Strengths And Weaknesses:**

Strengths

- The paper tackles the lack of a truly unified model that can both interpret medical images and generate visual outputs within a single forward pass.

- The authors introduce UniMed-5M dataset. The quality control pipeline, combining coarse filtering, MedGemma-based alignment scoring, and expert validation with inter-rater agreement (κ > 0.80), is methodologically sound. The data spans 8 imaging modalities and covers understanding, generation, and interleaved tasks, providing a comprehensive resource\

- I appreciate the detailed failure case analysis (Section A.8) covering text artifacts in generation, structural preservation issues in editing tasks, and fine-grained recognition failures.

- The evaluation covers a wide range of tasks and baselines. The comparison includes both specialized understanding-only models

Weaknesses

- The UniMedVL architecture closely follows Bagel (Deng et al., 2025), adopting the same dual-encoder design with shared self-attention and task-specific FFNs, the same VAE decoder, and the same combined NTP + rectified flow loss. The paper acknowledges this lineage ("Following Deng et al. (2025)") but does not clearly articulate what, if any, architectural modifications were made specifically for the medical domain. The main novelty appears to reside in the data curation and training recipe rather than the model architecture itself. This should be more clearly acknowledged.

- Trivial Information-Theoretic Justification: The information-theoretic argument in Section 3.3 (Eq. 3) and Appendix A.2 establishes that joint conditional entropy is at most the sum of marginals, with the gap equal to the conditional mutual information, this is well known and provides no new insight

- Generation Evaluation Protocol: For image generation (Table 2), the test set is a 20% split of the authors' own proposed dataset. This raises circularity issues as the data curation decisions directly control the test distribution. Furthermore, the generation benchmarks are evaluated using gFID and BioMedCLIP score, but there is no human/expert evaluation of generated image quality or clinical plausibility for the text-to-image task, which is critical in medical imaging generation.

- (Minor writing issues) a) There is a typo: "primiarly" (line 76). (b) Table numbering is inconsistent: two tables are both labeled "Table 9" (one in the main text for generation ablation, one in the appendix for hyperparameters). (c) Table 14 is referenced as "Table ??" in the text (line 1853) (d) Some figures (especially Fig. 2) are extremely dense and difficult to parse at standard print size.

---

> ### Author Rebuttal · Authors · 2026-03-31
>
> Thank you for the detailed and constructive comments. We appreciate the reviewer’s concerns regarding architectural originality, the role of the information-theoretic argument, the generation evaluation protocol, and the discussion of limitations. We address these points below.
>
> > **Weakness 1 \& Q1: Model architecture clarification.**
>
> **A1.** We agree that UniMedVL largely inherits the Bagel architecture, and we will revise the manuscript to make this explicit. We do not claim domain-specific architectural modifications to the Bagel backbone. The main contribution is instead to show that a single unified architecture can support medical image understanding and generation through medical-specific data construction and training design.
>
> Concretely, our contribution lies in: **(i)** UniMed-5M, which reformulates diverse medical datasets into a unified multimodal input-output interface; **(ii)** a medical quality-control pipeline to improve large-scale image-text fidelity; and **(iii)** a three-stage Progressive Curriculum Learning strategy in which Stage 1 establishes basic medical capabilities, Stage 2 strengthens instruction following with high-quality medical instruction data, and Stage 3 enhances understanding-generation synergy through interleaved tasks. We will therefore revise the paper so that the originality claim is framed more precisely as a medical data and training contribution.
>
> > **Weakness 2: Information-theoretic justification.**
>
> **A2.** We agree that the result in **Section 3.3 (Eq. 3)** and **Appendix A.2** is a standard information-theoretic identity. Our intention was not to present it as a theoretical contribution, but only as a perspective on why a unified understanding-generation model may exploit shared information more effectively than separate understanding-only and generation-only models. We will revise the wording to make this motivational role clearer and avoid overstating the novelty of this analysis.
>
> > **Weakness 3 \& Q2/Q3: Generation evaluation protocol and clinical plausibility.**
>
> **A3.** We agree that the original split of UniMed-5M should be interpreted only as an in-domain evaluation, not as evidence of external generalization. To address this concern, we additionally prepared a held-out external benchmark with 1,000 samples for each of 8 modalities from datasets not used in training. Averaged across the eight modalities, the results are in Table 2:
>
> **Table 2.** External evaluation results for image generation on unseen datasets. Each modality contains 1,000 samples. Each cell is reported as FID ↓ / BioMedCLIP ↑.
>
> | Model | CFP | CT | CXR | Endoscopy | Histopathology | MR | OCT | Ultrasound |
> |---|---:|---:|---:|---:|---:|---:|---:|---:|
> | UniMedVL | 60.00 / **0.72** | 100.09 / **0.70** | **63.54** / **0.71** | **94.04** / **0.70** | **102.31** / **0.72** | 87.54 / **0.73** | **54.20** / 0.70 | 86.63 / 0.70 |
> | UniMedVL Gen | 134.41 / 0.71 | **74.25** / 0.68 | 130.41 / 0.67 | 126.89 / 0.70 | 107.34 / 0.67 | **61.87** / 0.71 | 69.68 / **0.72** | **56.81** / **0.71** |
> | Bagel | 231.92 / 0.64 | 167.70 / 0.61 | 126.53 / 0.65 | 158.24 / 0.68 | 185.27 / 0.62 | 134.80 / 0.63 | 245.51 / 0.63 | 280.50 / 0.65 |
> | LlamaGen-MediTok | **57.96** / 0.71 | 104.56 / 0.63 | 140.56 / 0.62 | 188.36 / 0.65 | 211.40 / 0.61 | 113.27 / 0.65 | 114.99 / 0.71 | 65.86 / 0.69 |
> | Stable Diffusion | 119.76 / 0.70 | 133.05 / 0.65 | 124.12 / 0.64 | 136.10 / 0.64 | 227.65 / 0.60 | 127.58 / 0.69 | 98.07 / 0.70 | 188.49 / 0.67 |
>
> **Metrics**.   We also agree that gFID and BioMedCLIP cannot fully capture clinically critical errors or clinical plausibility. The current submission does not include a blinded radiologist study for the text-to-image task, and we will state this explicitly as a limitation.
>
> We also conducted a blinded expert study to further assess the realism of images generated by UniMedVL. Specifically, we randomly selected 100 images, including 50 real images and 50 generated images, and asked two experts to distinguish real from generated samples in a blinded setting. On average, the true positive rate for generated images was 64%, the true negative rate for real images was 80%, and the balanced accuracy was 72%. These results provide additional evidence that UniMedVL can generate visually realistic medical images beyond automatic metrics.
>
>
> > **Weakness 4: Minor writing issues and limitations.**
>
> **A4.** We appreciate these comments and will correct the corresponding presentation issues in the revision.

---

> > ### Author Rebuttal · Reviewer_RYVn · 2026-04-04
> >
> > Thanks for agreeing in the rebuttal, I appreciate the honest acknowledgments and believe the papers contributions are interesting. I’ll maintain my score accordingly.

---

> > > ### Author Response · Authors · 2026-04-04
> > >
> > > Dear Reviewer RYVn,
> > >
> > > Thank you for your support and constructive feedback, which have significantly improved the quality of our work.
> > >
> > > We are glad that our discussion has fully resolved your concerns, and we will integrate these insights to finalize the paper.
> > >
> > > Best regards,
> > >
> > > Authors

---

### Official Review · Reviewer_79xp · 2026-03-13

**Soundness:** 2
**Presentation:** 3
**Significance:** 3
**Originality:** 2
**Overall Recommendation:** 3
**Confidence:** 4

**Summary:**

The paper proposes a framework that builds a large interleaved multimodal dataset for medical vision–language modeling and trains a unified architecture capable of handling multiple tasks such as visual question answering (VQA), segmentation, and report generation. The authors curate the dataset by prompting existing vision–language models and applying several pre-processing and post-processing steps to generate structured training pairs. Using this dataset, the method trains a vision–language model training pipeline and evaluates the resulting model across a wide range of medical tasks. Experimental results demonstrate competitive zero-shot performance across multiple benchmarks compared to several baseline models, suggesting that the curated dataset helps improve general multimodal medical understanding.

**Compliance With Llm Reviewing Policy:**

Affirmed.

**Key Questions For Authors:**

Can the author provide additional experiments on fine-tuning for VQA and compare the results with those of current models benchmarked on this task? Also, please explain and discuss the other drawbacks mentioned above. While I am quite skeptical about the methodological novelties, the paper's impact is another strong point that can be considered, given the sufficient and convincing experiments.

**Limitations:**

Please include a specific discussion of the paper's limitations.

**Strengths And Weaknesses:**

***A. Strengths***

**i. Effort in constructing an interleaved medical multimodal dataset.**
The paper presents a significant effort in curating a single interleaved dataset that integrates multiple types of supervision signals (e.g., segmentation, VQA, and report-style annotations). Such unified datasets are valuable for training general-purpose medical vision–language models.

**ii. Unified architecture for multiple medical tasks.** The work explores a single model capable of handling multiple task formats (e.g., segmentation, VQA, and textual outputs). This direction aligns with recent trends in multimodal foundation models and could help reduce fragmentation across specialized medical models.

***iii. Comprehensive evaluation across multiple tasks and datasets.***  The model is evaluated across several benchmarks and task types, with comparisons against multiple baseline models. This provides a relatively broad view of the model’s capabilities.

***iv. Strong zero-shot evaluation coverage.*** Author reports extensive zero-shot experiments, which help assess the representation quality of the pretrained model and demonstrate generalization across datasets.

***B. Weakness***

***- Limited methodological novelty.***
In my opinion, the main contribution lies primarily in curating the dataset and adapting an existing VLM training pipeline, while the model architecture and training strategy closely follow existing work in the general vision–language literature. For instance, in Figure 3, the training procedure consists of three stages: foundation training, instruction tuning, and unified multimodal training, which are commonly adopted in recent VLM/VLLM frameworks by adding a specific visual encoder to return image outputs.

***- Evaluation focuses mainly on zero-shot settings:***
***Most experiments are conducted in zero-shot settings***, which are useful for evaluating representation quality but ***do not fully reflect real-world deployment scenarios***. In practice, medical applications usually require fine-tuning on downstream datasets, especially for critical tasks such as VQA or diagnosis-related reasoning. Including fine-tuning experiments would provide a clearer understanding of the practical performance gap between zero-shot and adapted models, which is often what matters most for end users such as clinicians. Thus Reviewer expects to see fine-tuning performance of key applications such as VQA and compare with similar model sizes (7B, 14B, don't need to include all), such as the VQA benchmark in [1], and similar ones on other downstream tasks, where zero-shot lags behind fully fine-tuning with a large gap.

***- Potential issues in presentation and experimental framing.***

+ *Figure 1 appears misleading*. For instance, statements (e.g., “No diagnostic explanations for generated images”) in Figure 1 are not true because such capabilities already support such functionalities in certain tasks [2]. A more informative figure should focus on the model’s ability to support multiple query types (e.g., segmentation, VQA, report generation) within a unified framework.

+ *Methodology description is somewhat unclear.*
The distinction between Stage 2 and Stage 3 in Figure 3 is difficult to understand, as the diagrams appear very similar and the textual explanation does not clearly highlight what differs between the two stages. It is also unclear which datasets are used in each stage and which components of the model are trained or frozen, making the training pipeline harder to follow.

[1] ExGra-Med: Extended Context Graph Alignment for Medical Vision-Language Models, NeurIPS 2025.

[2] MedRG: Medical report grounding with a multi-modal large language model

---

> ### Author Rebuttal · Authors · 2026-03-31
>
> Thank you for the careful review. We appreciate your concerns regarding methodological novelty, downstream evaluation, presentation clarity, and limitations. We address these points below.
>
> > **W1: Contribution of our work.**
>
> **A1.** We agree that UniMedVL is not introduced as a new backbone, and we will revise the manuscript to state this more explicitly. The architectural basis follows prior unified multimodal models, and the main contribution of this work lies in making unified medical understanding and generation feasible in a single checkpoint through medical-specific data construction and training pipeline design, rather than through a new architecture.
> The main challenge in the medical field is that the data are highly heterogeneous across modalities, task formats, supervision granularity, and output types, while medically faithful generation requires preserving pathology and cross-modal consistency instead of only matching general semantics. To address this, we propose the Observation-Knowledge-Analysis (OKA) framework.
>   Specifically, we propose UniMed-5M as the dataset-level foundation for unified understanding and generation capabilities (Observation level), and a novel three-stage Progressive Curriculum Learning strategy (Knowledge level) to transfer medical multimodal knowledge from UniMed-5M to the pre-trained base model (e.g., Bagel), with Stages 1 and 2 injecting image understanding and generation knowledge and Stage 3 enhancing the synergy between understanding and generation.
>  Through these processes, we construct UniMedVL (Analysis level), a unified multimodal model for medical images and textual queries that yields visual and textual responses.
>
>
> > **W2: Performance after fine-tuning on VQA tasks.**
>
> **A2.** We agree that zero-shot evaluation alone is insufficient for deployment-oriented medical tasks. Following your suggestion, we fine-tuned UniMedVL on three representative medical VQA benchmarks and compared it with strong baselines already reported in the paper:
>
> | Model | VQA-RAD | SLAKE | PathVQA |
> |---|---:|---:|---:|
> | Lingshu-7B | 62.7 | 77.0 | 59.6 |
> | GMAI-VL | 66.3 | 72.9 | 39.8 |
> | HuatuoGPT-Vision-7B | 53.0 | 49.1 | 32.0 |
> | **UniMedVL** | **61.9** | **75.4** | **53.5** |
> | **UniMedVL-finetune** | **66.96** | **91.61** | **62.61** |
>
> Compared with zero-shot UniMedVL, fine-tuning improves performance by **+5.06** on VQA-RAD, **+16.21** on SLAKE, and **+9.11** on PathVQA. These results show that although the main paper emphasizes unified zero-shot capability, the model can also adapt effectively to downstream medical VQA settings. We will include these results in the revision and clarify the practical gap between zero-shot and adapted performance.
>
> > **W3: Clarification revision.**
>
> **A3.** We agree that the original presentation can be clearer. We have revised Figure 1 so that it emphasizes the unified support for multiple medical query types, including segmentation, VQA, and report generation, rather than implying that prior work cannot provide any diagnostic grounding. We have also revised Figure 3 to make the distinction between Stage 2 and Stage 3 explicit by marking trainable and frozen components and by clarifying the stage-specific data. In particular, Appendix A.3.1 now makes the stage-wise distribution more explicit, with Stage 1 using 5.6M samples, Stage 2 using 1.9M high-quality instruction data, and Stage 3 using 0.33M interleaved-task samples, while Appendix A.3.2 summarizes the major datasets used at each stage. For your review, the revised figures are available at **[link](https://anonymous.4open.science/r/icml_2026_unimedvl_rebuttal_materials/)**.
>
> > **Limitations.**
>
> **A4.** We appreciate the suggestion and will add a dedicated limitations discussion. In particular, we will state explicitly that UniMedVL does not introduce a new architecture relative to prior unified models, that the current study focuses on 2D medical imaging, and that the current evaluation still relies mainly on standard automatic metrics and benchmark protocols. These settings are useful for reproducible comparison, but they do not fully capture clinical utility or expert preference. We will therefore revise the paper to make these boundaries explicit and position the current work as a step toward unified medical multimodal modeling rather than a complete clinical solution.

---

> > ### Author Rebuttal · Reviewer_79xp · 2026-04-03
> >
> > Thank you to the authors for the detailed responses, particularly the additional results after fine-tuning on VQA tasks. These clarifications are helpful.
> >
> > However, the reported results still do not appear to be contextualized against the current state-of-the-art (SOTA) among open-source medical MLLMs. For example, prior work such as *LLaVA-Med (NeurIPS 2023)* reports performance of 72.64 on VQA-RAD, 83.43 on SLAKE, and 64.06 on PathVQA, while more recent work like *ExGra-Med (NeurIPS 2025)* achieves 74.91, 85.46, and 64.82 on the same benchmarks, respectively.
> >
> > In comparison, the performance of UniMedVL after fine-tuning still appears to lag behind these published results. Additionally, several of the baselines used for comparison in the manuscript are preprints and have not yet been peer-reviewed, which raises concerns about the strength of the empirical positioning.
> >
> > Could the authors please clarify this discrepancy? In particular, it would strengthen the paper to:
> >
> > i. Include comparisons with established SOTA models on these benchmarks, and/or
> > ii. Provide a more detailed discussion explaining why such comparisons are currently missing or not directly applicable.

---

> > > ### Author Response · Authors · 2026-04-04
> > >
> > > Thank you for the insightful follow-up and for drawing our attention to the comparison with LLaVA-Med and ExGra-Med. This is a valuable suggestion that has helped us identify an important evaluation detail.
> > >
> > > In our experimental setting, the evaluation is performed within the VLMEvalKit [1] framework, where VQA-RAD, SLAKE, and PathVQA use **accuracy** for both open-set and closed-set questions, following GMAI-VL (AAAI 2026) [4], whereas the cited works use recall for open-set questions and accuracy for closed-set questions [2,3].
> > >
> > > We apologize for not making this distinction clear in our initial rebuttal, which led to numbers that are not directly comparable. To ensure a fair and meaningful comparison, we additionally evaluated our model on the three benchmarks under the same metric protocol used by LLaVA-Med and ExGra-Med. The results are shown below.
> > >
> > > | Model | VQA-RAD (open) | VQA-RAD (close) | VQA-RAD (avg) | SLAKE (open) | SLAKE (close) | SLAKE (avg) | PathVQA (open) | PathVQA (close) | PathVQA (avg) | Overall |
> > > |---|---:|---:|---:|---:|---:|---:|---:|---:|---:|---:|
> > > | LLaVA-Med [2] | 63.65 | 81.62 | 72.64 | 83.44 | 83.41 | 83.43 | 36.78 | 91.33 | 64.06 | 73.37 |
> > > | ExGra-Med [3] | 66.35 | 83.46 | 74.91 | 85.34 | 85.58 | 85.46 | 36.82 | 90.92 | 63.87 | 74.75 |
> > > | UniMedVL | 37.57 | 77.69 | 57.63 | 68.07 | 82.45 | 75.26 | 21.95 | 78.17 | 50.06 | 60.98 |
> > > | UniMedVL-finetune | 45.50 | 84.06 | 64.78 | 89.66 | 95.49 | 92.58 | 34.91 | 90.27 | 62.59 | 73.32 |
> > >
> > > Under the aligned protocol, UniMedVL-finetune achieves an overall score of 73.32, which is in a comparable range to LLaVA-Med (73.37) and ExGra-Med (74.75). We note that ExGra-Med achieves the best overall performance. Meanwhile, UniMedVL-finetune shows particular strength on SLAKE (92.58), outperforming both baselines by a notable margin, while remaining competitive on VQA-RAD and PathVQA. We believe these results suggest that the original gap reported in our initial rebuttal was largely attributable to protocol differences, and that UniMedVL can adapt effectively to downstream VQA benchmarks after fine-tuning, despite being a unified model that simultaneously supports image generation.
> > >
> > > We will incorporate these corrected results, the aligned evaluation protocol, and the comparison with LLaVA-Med and ExGra-Med into the revised manuscript to ensure the benchmarking is transparent and complete. We are grateful to the reviewer for raising this point.
> > >
> > > [1] Duan H, et al. Vlmevalkit: An open-source toolkit for evaluating large multi-modality models, ACM MM, 2024.
> > >
> > > [2] Li et al., LLaVA-Med: Training a Large Language-and-Vision Assistant for Biomedicine in One Day. NeurIPS, 2023.
> > >
> > > [3] Nguyen et al., ExGra-Med: Extended Context Graph Alignment for Medical Vision-Language Models. NeurIPS, 2025.
> > >
> > > [4] Li et al. Gmai-vl & gmai-vl-5.5 m: A large vision-language model and a comprehensive multimodal dataset towards general medical ai. AAAI, 2026.

---

### Decision · Program_Chairs · 2026-04-30

**Decision:**

Accept (regular)

**Comment:**

Reviewers agreed that the paper makes substantial contributions, including great effort in curating the interleaved UniMed-5M dataset, proposing a unified UniMedVL architecture for handling various medical tasks without separate checkpoints, conducting comprehensive evaluations across multiple tasks and baselines, and providing extensive zero-shot experiments to demonstrate the model’s generalization ability. However, they also raised serious concerns about the methodological novelty and dataset quality: the model architecture and training strategy closely follow existing general-domain VLM frameworks with limited domain-specific architectural innovations, and the dataset’s quality control relies heavily on MedGemma-27b (introducing potential bias), lacking direct estimates of filtering errors for the full corpus. The rebuttal addressed most of the above concerns. But, there is still no unanimous agreement among reviewers. After careful deliberation, I recommend accepting this paper since its merits over-weights its limitations.